



# Short-Lived Halogen Sources and Chemistry in the Community Earth System Model v2 (CESM2-SLH)

Rafael P. Fernandez[1,2,*], Carlos A. Cuevas[1], Julián Villamayor[1], Aryeh Feinberg[1], Douglas E. Kinnison[3], Francis Vitt[3], Adriana Bossolasco[1,4], Javier A. Barrera[1,5], Amelia Reynoso[2], Orlando G. Tomazzeli[2], Qinyi Li[6] and Alfonso Saiz-Lopez[1,*]

[1] Department of Atmospheric Chemistry and Climate, Institute of Physical Chemistry Blas Cabrera, CSIC, Madrid, 28006, Spain
[2] Institute for Interdisciplinary Science, National Research Council (ICB-CONICET/UNCUYO). School of Natural Sciences, National University of Cuyo (FCEN/UNCUYO), Mendoza, 5501, Argentina
[3] Atmospheric Chemistry, Observations & Modeling Laboratory (ACOM), NSF National Center for Atmospheric Research (NCAR), Boulder, CO 80301, USA
[4] Physics Institute of Northwest Argentina (INFINOA-CONICET/UNT), Tucumán, Argentina.
[5] Research Institute for Physical Chemistry of Córdoba, National Research Council (INFIQC-CONICET/UNC). Department of Physical Chemistry, School of Chemical Sciences, National University of Córdoba (FCQ/UNC), Córdoba, Argentina
[6] Environment Research Institute, Shandong University, Qingdao, China.

*Correspondence to*: Alfonso Saiz-Lopez (a.saiz@csic.es) and Rafael P. Fernandez (rpfernandez@mendoza-conicet.gob.ar)

**Abstract.** The implementation of short-lived halogen (SLH) sources and atmospheric chemistry in the Community Earth System Model v1 (CESM1), has extended the capabilities of this state-of-the-art model to study how SLH chemistry alters the oxidative capacity of the atmosphere and, consequently, the Earth's climate. In this manuscript, we summarize 15 years of research and developments of SLH chemistry in CESM1 and present a complete revision of the porting of the original SLH implementation into the latest released version of CESM (v2), hereafter CESM2-SLH. This includes a detailed description of all offline and online sources of organic and inorganic SLH, as well as the gas-phase and heterogeneous recycling of chlorine, bromine and iodine in the troposphere and stratosphere, including their species-independent atmospheric sinks. In doing so, we provide a comprehensive evaluation of how changes in model parameters and coupled dynamics within the Community Atmosphere Model v6 (CAM6) affect SLH abundances and their implications. The new CESM2-SLH implementation offers various component sets (compsets) and resolutions, all of which result in equivalent global budgets and zonal distributions of organic and inorganic chlorine, bromine and iodine, which in turn, lead to SLH impacts on atmospheric composition that are consistent with previous CESM1 results. The released CESM2-SLH version includes specific namelist options, input files and technical notes detailing the most important SLH updates implemented over the CESM2/CAM6 routines. Our results show that the global tropospheric burden and tropical stratospheric injection of organic and inorganic chlorine, bromine and iodine species in CESM2-SLH are in agreement with observational assessments and previous modelling studies using CESM1, resulting in significant reductions in global ozone abundance (−21-28% at the surface, −17-22% in the troposphere and −2-3% in the stratosphere) as well as in OH and NO$_2$ (ranging between −2-9% and −1-10%, respectively), depending on the specific model configuration and resolution. Based on this, we encourage the wider CESM community to consider the released CESM2-SLH scheme to obtain a more realistic representation of the background influence of natural and anthropogenic short-lived halogen sources and chemistry in air quality and Earth's climate studies.

**Keywords.** CESM2-SLH, Natural and Anthropogenic Halogens, Atmospheric Chemistry, Chemistry-Climate coupling.



**1 Introduction**

Over the past few decades, both observational and modelling studies have shown that Short-Lived Halogens (SLH) are widespread and ubiquitously distributed throughout the troposphere and stratosphere (Chance, 1998; Alicke et al., 1999; Saiz-Lopez et al., 2007; Read et al., 2008; Thornton et al., 2010; Prados-Roman et al., 2015a; Koenig et al., 2020). This SLH definition (Saiz-Lopez et al., 2023) includes the contribution from organic (i.e., carbon-bonded) very short-lived (VSL) halocarbons with a chemical lifetime smaller than 6 months plus the contribution of inorganic halogen species (collectively called $Cl_y$, $Br_y$ and $I_y$ for chlorine, bromine and iodine, respectively) that, through chemical cycling, partition between reactive and reservoir fractions within the gas-phase. Due to their short lifetime, reactive halogens affect atmospheric chemical composition including the depletion of tropospheric ozone ($O_3$), and consequently, the production of the hydroxyl radical (OH) and the partitioning between both hydrogen ($HO_2/OH$) and nitrogen ($NO_2/NO$) oxides (Saiz-Lopez and von Glasow, 2012). In addition, SLH compounds contribute to stratospheric ozone depletion (Salawitch et al., 2005; Sinnhuber et al., 2009), particularly over Antarctica where reactive bromine and iodine enhance de depth and size of the springtime ozone hole (Fernandez et al., 2017; Cuevas et al., 2022). Recently, SLH has been shown to influence the Earth radiative balance through direct and indirect chemical coupling with methane ($CH_4$) (Li et al., 2022), $O_3$ (Saiz-Lopez et al., 2023) and OH (Bossolasco et al., 2025), as well as the oxidation of dimethyl sulphide (DMS) and other sulphate sources which, in turn, can alter the formation of cloud condensation nuclei (Wohl et al., 2024).

VSL consist of organic halogenated source gases (SGs) such as chloro-, bromo-, iodo-carbons which are present in the atmosphere in the form they were emitted. Bromine and iodine SGs are predominantly natural and emitted from the oceans (e.g., $CHBr_3$, $CH_2Br_2$ and $CH_3I$), whereas chlorine species such as $CH_2Cl_2$ and $C_2Cl_4$ have an important contribution from anthropogenic sources (WMO, 2022). The oceans are also the source of abiotic emission of hypoiodous acid (HOI) and molecular iodine ($I_2$) following the deposition of tropospheric ozone to the ocean surface and the reaction with iodide ions (Carpenter et al., 2013; MacDonald et al., 2014). This oceanic source of inorganic iodine is estimated to account for ~75% of the total source of atmospheric iodine, with the rest coming from VSL iodocarbons (Prados-Roman et al., 2015b). Similarly, very efficient heterogeneous reactions occurring at the surface of sea-salt aerosols (SSA) can result in the initial uptake and subsequent release of different halogen species within the Marine Boundary Layer (MBL), constituting a significant source of inorganic chlorine and bromine to the lower atmosphere, which is currently considered to be the dominant natural halogen source (WMO, 2018; Saiz-Lopez et al., 2023).

Once in the atmosphere, VSL SGs photo-decompose to release inorganic halogen product gases (PGs) that cycle back and forth between reactive ($ClO_x$, $BrO_x$ and $IO_x$) and reservoir species, the whole group of which constitute SLH species. In contrast, long-lived Ozone Depleting Substances (ODSs), such as chloro-fluoro-carbons (CFCs) and brominated halons, remain unreactive throughout the troposphere until they are photodecomposed in the stratosphere (WMO, 2018, 2022) . The photochemical cycling of SLH involves chemical reactions with other hydrogen, nitrogen, carbon and sulphur species, modifying the chemical composition of the atmosphere and therefore affecting its oxidative capacity and consequently, the Earth's climate (Li et al., 2022; Saiz-Lopez et al., 2023; Bossolasco et al., 2025). While detailed descriptions of the overall processes and reactions occurring in the atmosphere have been widely described in the literature (e-g-. see the reviews of (Saiz-Lopez and von Glasow, 2012; Simpson et al., 2015) and supplementary information of previous studies (Saiz-Lopez et al., 2014, 2023; Badia et al., 2021)), here we only provide a general description of the most relevant chemical cycles in which SLH participate, which can certainly be expanded and complemented following the comprehensive SLH literature referenced herein.

The first step in the long reactive chain of SLH is the photochemical breakdown of either organic VSL halocarbons and/or photo-labile inorganic halogen species that results in the release of chlorine, bromine and iodine atoms, typically represented by X,Y = Cl, Br, I (R1a–R1c). Once in the atmosphere, these highly reactive halogen atoms (particularly I and Br) can catalytically destroy ozone ($O_3$), which in turn, controls the tropospheric formation of the hydroxyl radical OH through



the reaction of $O^1D$ with water vapour (R2−R4). Here, note that OH is typically referred to as the "atmospheric detergent"
because it dominates the photodecomposition of different volatile organic compounds (VOCs), and therefore controls the
oxidative capacity of the troposphere (R5a). Similarly, direct reactions of halogen radicals (particularly Cl atoms) with several
VOCs also contribute to the photochemical degradation of biogenic compounds and anthropogenic pollutants emitted to the
atmosphere (R5b−R7), including $CH_4$. As a result, halogen chemistry alters the overall partitioning between nitrogen ($NO_x$ =
$NO + NO_2$) and hydrogen ($HO_x$ = $OH + HO_2$) cycling (R8−R10), which has important implications for determining the
chemical composition not only for halogens, but also for several reactive and reservoir species in the global atmosphere.
Therefore, SLH chemistry can either increase the photochemical degradation rate of VOCs as well as the formation of aerosols
(Saiz-Lopez and von Glasow, 2012; Simpson et al., 2015), while at the same time it can decrease the OH-driven chemical
destruction and/or production rates due to their direct influence on $O_3$ abundance (Iglesias-Suarez et al., 2020; Badia et al.,
2021; Saiz-Lopez et al., 2023), which in turn controls OH (Benavent et al., 2022; Bossolasco et al., 2025). Finally, it is worth
noting that depending on the background environment, reactive halogen chemistry can also result in ozone production under
polluted environments due to the formation of nitryl chloride ($ClNO_2$) that, through photolysis (R11), represents an additional
source of $NO_x$ (R12-R13).

$$CHX_3 + h\upsilon \rightarrow 3X \tag{R1a}$$
$$CH_2XY + OH \rightarrow X + Y \tag{R1b}$$
$$X_2/XY + h\upsilon/OH \rightarrow (2)X + Y \tag{R1c}$$

$$X + O_3 \rightarrow XO \tag{R2}$$
$$O_3 + h\upsilon \rightarrow O^1D + O_2 \tag{R3}$$
$$O^1D + H_2O \rightarrow 2OH \tag{R4}$$

$$OH + VOCs \rightarrow RO_2 + HO_2 \tag{R5a}$$
$$X/Y + VOCs \rightarrow RO_2 + HX \tag{R5b}$$
$$RO_2 + NO \rightarrow NO_2 + RO \tag{R6}$$
$$RO + O_2 \rightarrow HO_2 + products \tag{R7}$$

$$HO_2 + NO \rightarrow NO_2 + OH \tag{R8}$$
$$XO + HO_2 \rightarrow HOX \tag{R9}$$
$$HOX + h\upsilon \rightarrow X + OH \tag{R10}$$

$$ClNO_2 + h\upsilon \rightarrow Cl + NO_2 \tag{R11}$$
$$NO_2 + h\upsilon \rightarrow NO + O^3P \tag{R12}$$
$$O^3P + O_2 + M \rightarrow O_3 \tag{R13}$$




In this work, we describe the implementation of SLH chemistry into the Community Earth System Model, version 2
(CESM2) (Danabasoglu et al., 2020), which include version 6 of the Community Atmospheric Model (CAM6) physics
(Gettelman et al., 2019a), as well as the tropospheric and stratospheric (TS1) chemistry scheme within CAM6-Chem (Emmons
et al., 2020). The corresponding SLH implementation in the Whole Atmosphere Community Climate Model v6 (WACCM6)
troposphere, stratosphere, mesosphere, and lower thermosphere (TSMLT1) chemical scheme (Gettelman et al., 2019b) has
also been performed. The implementation is based on most of the previous SLH developments implemented in CESM1
(Ordóñez et al., 2012; Saiz-Lopez et al., 2012). Due to the changes between the base CESM1 and CESM2 models in several
atmospheric fields such as sea surface temperature (SST), SSA and the corresponding aerosols surface area density (SAD)
computation, as well as the different prescribed pollutant emissions, meteorological fields and/or atmospheric dynamics
(Gettelman et al., 2019a; Danabasoglu et al., 2020; Emmons et al., 2020; Simpson et al., 2023), all of which affect SLH
chemical cycling, transport and washout in the atmosphere, we have included a new user-defined namelist section including a
set of scaling-factors that allows CESM2-SLH performance to be consistent with the previous version of the model.
The paper is organized as follows: Section 2 reviews the historical developments of SLH chemistry in CESM1, and
describes the implementation and porting to the different CESM2-SLH compsets available. Then, in Section 3 we present a
comprehensive description of SLH global abundances and distributions obtained with the new CESM2-SLH model and how
they compare with previous CESM1 results as well as with observations. Finally, section 4 provides several remarks,
suggestions and notes of caution for new CESM2-SLH users, emphasising the need and types of model validation checks that
must be performed in case other configurations beyond the ones described here are used.
**2 Model Description**
The implementation of SLH sources and chemistry in CESM described in this work has been performed in version
2.2.0 (branch *cesm2.2-asdbranch*), whose atmospheric component is based on CAM6 (branch *cam_cesm2_2_rel_09*). To
simplify the validation of SLH chemistry amongst all the available model configurations in CESM2, we focused on the widely
used *FCnudged* and *FCHIST* compsets (https://docs.cesm.ucar.edu/models/cesm2/config/2.2/compsets.html), expanding their
default namelists and configuration files to include SLH sources and chemistry. These *FCASE* compsets are forced with
prescribed SSTs, while an additional fully-coupled model compset for WACCM has also been developed based on *BWHIST*
(Gettelman et al., 2019b). In particular, and given that the following model inputs are of relevance for the computation of SLH
sources and sinks, we highlight that the current CESM2-SLH configuration includes a climatological SST configuration from
(Huang et al., 2017) and an explicit representation of SSA lifting, transport and removal based on 3 (three) individual bins as
described in (Tilmes et al., 2023). This differs from the original SSA representation available in the CAM4-Chem version used
for the initial implementation of SLH in CESM1. In addition, note that most of the previous CAM4-Chem configurations were
performed using the old Specified Dynamics (SD) approach forced with high-frequency meteorological fields from a previous
CAM-Chem simulation (e.g., (Saiz-Lopez et al., 2023) and references therein), while the *FCnudged* and *FWnudged* compset
in CESM2-SLH are based on the new nudging approach (Davis et al., 2022) that considers air temperature (T), zonal and
meridional wind velocity (respectively, *u* and *v*) from MERRA2 (Modern-Era Retrospective analysis for Research and
Applications v2) reanalysis (Rienecker et al., 2011). Given that most of the initial SLH developments were performed on top
of a discontinued CESM1 version that is no longer available to the community, the current CESM2-SLH implementation is
presented and sustained with a double purpose: *i)* to be easily configured by users that are not initially familiarized with SLH
chemistry; and *ii)* to allow the whole scientific community to evaluate the global mean influence of SLH in atmospheric
chemistry and their implications for the Earth's climate system.



Below we provide a brief review of the historical implementation of SLH in CAM4-Chem (Section 2.1), and then
present a comprehensive description of the main sources (Section 2.2), gas-phase and multi-phase SLH chemistry (Section
2.3) and removal processes (Section 2.4) ported into CESM2-SLH. The final Section 2.5 describes the complete list of SLH
compsets available with their corresponding namelist updates and scaling factors, which are required for a proper SLH model
setup within the different resolutions and configurations existing in CESM2.

## 2.1 Historical Developments of SLH in CAM-Chem

Below we provide a brief description of the most important SLH developments originally implemented in CESM1
(CAM4-Chem). Although not strictly chronological, we begin with the initial descriptive papers that presented the main
sources, reactions and halogen species, then continue with the subsequent updates and improvements, and finalize with the
latest modelling studies that have demonstrated the importance of considering SLH chemistry to improve our understanding
of atmospheric chemistry, air quality and climate evolution from past, to present and future. All these developments, have been
led by the Atmospheric Chemistry and Climate group (AC2, IQF-CSIC, Madrid, Spain), in close cooperation with the
Atmospheric Chemistry Observations & Modeling Department (ACOM-NCAR, Boulder, Colorado) and the Institute for
Interdisciplinary Science (ICB-CONICET/UNCUYO, Mendoza, Argentina), as well as with fundamental contributions from
other research groups around the world. Although most of previous research used the term VSL or VSLs to refer to organic
halogenated very short-lived substances, in this work and following the most recent and complete representation of short-lived
halogens in CESM (Saiz-Lopez et al., 2023), we expanded the terminology to SLH, which includes, in addition to VSL, those
inorganic halogen species ($Cl_y$, $Br_y$ and $I_y$) that rapidly interconvert between each other and are responsible for the halogen
impacts in the atmosphere (Saiz-Lopez et al., 2023).
The initial description of SLH halogen chemistry in CAM-Chem was published in (Ordóñez et al., 2012), who
presented a complete set of tables with all photochemical processes considered and also developed the VSL halocarbon
emission inventory that was used in all subsequent studies. The companion work of (Saiz-Lopez et al., 2012) performed the
first estimation of the impacts of combined SLH (chlorine, bromine and iodine) on tropospheric ozone and radiative balance.
Here we acknowledge that these initial works considered offline inorganic iodine emissions. Afterwards, (Prados-Roman et
al., 2015b, 2015a) implemented the online computation of oceanic iodine emissions and determined the negative geophysical
feedback with tropospheric ozone that led to the hypothesis that atmospheric iodine levels would have tripled since the onset
of the Industrial Era, later confirmed by ice-core observations (Cuevas et al., 2018). Further updates related mostly to
heterogeneous recycling processes occurring in the MBL, Free Troposphere (FT) and Upper Troposphere (UT), relevant for
properly quantifying the contribution of SLH to the stratospheric halogen loading of bromine (Fernandez et al., 2014) and
iodine (Saiz-Lopez et al., 2014, 2015), led to quantification of the overall impact of natural bromine and iodine over the
Antarctic ozone hole (Fernandez et al., 2017; Cuevas et al., 2022), as well as the seasonal variability within the mid-latitudes
(Barrera et al., 2020). These updates included the online implementation of SSA-dehalogenation sources which are of
particular importance for improving BrO observations in the MBL (Koenig et al., 2017), as well as the inclusion of higher
order iodine oxides ($I_xO_y$), whose effective photolysis was found to be necessary to reproduce iodine observations in the upper
troposphere and lower stratosphere (Saiz-Lopez et al., 2014, 2015; Koenig et al., 2020). For the particular case of chlorine, the
contribution of anthropogenic VSL was implemented based on the emissions inventories from (Hossaini et al., 2019) and
(Claxton et al., 2020), with further updates related to the inorganic chlorine emissions resulting from the heterogeneous
recycling of nitrogen oxides and nitric acid over chloride-rich aerosols, the so called acid-displacement reactions (Li et al.,
2022). These allowed to evaluate for the first time the combined impact of halogens (chlorine, bromine and iodine) over
tropospheric ozone during the 21st century (Iglesias-Suarez et al., 2020; Badia et al., 2021) in comparison with pristine pre-
industrial conditions (Barrera et al., 2023); as well as the enhanced chemical coupling (in concert) of natural and anthropogenic



SLH on the lowermost tropical stratosphere ozone trends (Villamayor et al., 2023). Based on these works, and thanks to the
implementation of CH$_4$ emission-driven simulations (Li et al., 2022) and polar halogen emissions from the sea-ice (Fernandez
et al., 2019, 2024), SLH were demonstrated to induce an overall cooling effect on the climate system arising from the direct
and indirect (and sometimes opposite) influence of reactive halogens over ozone, methane, aerosols and stratospheric water
vapour (Saiz-Lopez et al., 2023). CESM1 simulations including SLH also enabled quantification of the radiative influence of
DMS and methanethiol (MeSH) oceanic emissions on the direct sulphate aerosol radiative effect over the Southern Ocean
(Veres et al., 2020; Wohl et al., 2024), while the overall influence of SLH on all tropospheric oxidants has recently been
described in (Bossolasco et al., 2025). The role of anthropogenic SLH emissions from biomass burning and other sources on
long-term atmospheric chemistry and mercury contamination over continental Asia has also been evaluated using CESM1-
SLH (Chang et al., 2024; Fu et al., 2024). The model has been also employed to evaluate the potential of chlorine based
interventions in climate mitigation through methane reduction (Li et al., 2023; Meidan et al., 2024), as well as to quantify for
the first time the role of the stratosphere on the global mercury cycle and surface deposition (Saiz-Lopez et al., 2025).
It is important to note that not all processes, reactions and implications summarized in previous studies have been
implemented in the current release of CESM2-SLH described in this work. Please refer to Section 2.5.2 below for further
details. Finally, it is worth noting that following the pioneering implementation of combined SLH emissions and chemistry
into CAM4-Chem, other research groups implemented SLH schemes in different models, which in many cases followed the
original implementation of SLH in CESM, particularly for iodine. These include global and regional models such as GEOS-
Chem (Sherwen et al., 2016b, 2016a), CMAQ (Sarwar et al., 2015), WRF-Chem (Badia et al., 2019), SOCOL (Karagodin-
Doyennel et al., 2021) and LMDZ-INCA (Caram et al., 2023).

### 2.2 Short-Lived Halogen Emissions

The starting point of SLH implementation in CESM was the development of an oceanic emission inventory of VSL
halocarbons (Ordóñez et al., 2012), which not only represents an important tropospheric halogen source, but was also needed
to reproduce total stratospheric bromine loading observations (WMO, 2018, 2022). To achieve this goal, many global models
-including CESM- initially assumed a constant surface abundance of the two main bromocarbons (i.e., bromoform (CHBr$_3$)
and dibromomethane (CH$_2$Br$_2$)) and imposed a Lower Boundary Condition (LBC) of 1.2 pptv each (adding up to 6 pptv of
total bromine at the model surface), which resulted in an additional bromine stratospheric injection of approximately 5 pptv
(Eyring et al., 2013; Hegglin et al., 2014). However, motivated by the rapid photochemical-degradation of VSLs in the
troposphere, spatially- and temporally-resolved emission inventories were implemented to account for oceanic halocarbon
emission sources of bromine and iodine (which include mostly natural sources), and was later extended to chlorocarbons, for
which the anthropogenic emissions dominate. These original VSL oceanic sources, which represent the initial step releasing
highly reactive Cl, Br and I atoms in the lower troposphere, were further extended by computing the on-line recycling of
inorganic halogen sources occurring at the ocean surface as well as over sea-salt aerosols. Sections 2.2.1 and 2.2.2 describe,
respectively, the natural and anthropogenic sources of VSLs halocarbons implemented in CESM2-SLH, while Section 2.2.3
describes the online recycling of halogen reservoirs that represent a net source of SLH to the atmosphere.

### 2.2.1 Natural Offline Halocarbon Emissions

The implementation of VSL halocarbons emitted from the ocean in CESM has been extensively described in the
original work of (Ordóñez et al., 2012). The methodology follows a top-down approach based on chlorophyll-a monthly
climatology from the SeaWIFS project for the 1998-2003 period (Melin, 2013), and the iterative adjustment of emission flux
strength by comparing modelled distributions and vertical profiles of VSL halocarbons with both available surface and aircraft



observations. Nine halogenated VSL species are included in the (Ordóñez et al., 2012) emission inventory, including individual emissions of two bromocarbons ($CHBr_3$ and $CH_2Br_2$), three bromo-chlorocarbons ($CH_2BrCl$, $CHBrCl_2$ and $CHBr_2Cl$), two iodocarbons ($CH_3I$ and $CH_2I_2$), one iodo-chloro ($CH_2ICl$) and another iodo-bromo ($CH_2IBr$) carbon, respectively. All of these species are well known to be emitted from biologically active oceans as side products of the metabolite of different micro- and macro-algae and phytoplankton colonies (Carpenter and Liss, 2000; Carpenter et al., 2003). In the (Ordóñez et al., 2012) inventory, the oceanic emission shows a seasonal and latitudinal distribution, with most of the emission (~70%) occurring over the tropics (20°N-20°S), approximately 25% arising from the north and south mid-latitudes (20°-50° on both hemispheres) and the remaining 5% released from ice-free oceanic regions from the high-latitudes and polar regions. The net oceanic emission flux of each halocarbon species ($E_{spec}$) is given by Eq. (1);

$$E_{spec} = 1.127 \times 10^5 \times f_{spec} \times r_{coast} \times chl\_a , \qquad\qquad (1)$$

where $f_{spec}$ is a species-dependent scaling factor iteratively adjusted to improve model-observation agreement, $r_{coast}$ is a constant enhancement factor applied only to coastal areas outside the tropics (20°N-20°S), and $chl\_a$ is the SeaWIFS monthly climatology chlorophyll-a distribution (Ordóñez et al., 2012). Note that, differing from previous VSL inventories that considered only latitudinal bands (Warwick et al., 2006b, 2006a; Butler et al., 2007; Jones et al., 2010), the (Ordóñez et al., 2012) inventory introduces a tropical geographically-heterogeneous and seasonally dependent variability, which allows for an improved spatio-temporal representation of VSL halocarbon distribution in the global troposphere. Despite the emission flux is read offline with a monthly mean resolution, routine *mo_srf_emissions.F90* was modified to apply an hourly dependent profile to the flux strength of all halocarbons due to their photosynthetic dependence on radiation intensity: i.e., emissions follow either a Gaussian or top-hat diurnal profile, with minimal (or zero) night-time emissions and peak emissions at local solar noon (Ordóñez et al., 2012). Given that no long-term trend has been established for natural oceanic organic halogen emissions, the emission strength of oceanic halocarbons is assumed to follow a constant climatology. To include the offline VSL halocarbon emissions from the (Ordóñez et al., 2012) inventory, users just need to include the name of the species within the *srf_emiss_specifier* namelist option, and point at the specific file for each individual species (see Section 2.5). All changes related to the implementation of offline oceanic VSLs sources have been performed in routines *mo_srf_emissions.F90* and *mo_usrrxt.F90* within the *chemistry/mozart* folder.

Finally, it is important to mention that the original (Ordóñez et al., 2012) inventory was developed almost 15 years ago and initially implemented in CAM4-Chem. Since then, several improvements in the representation of large-scale ascent and transport across the MBL have been included in CESM2 (Simpson et al., 2020). In addition, updated in the emission strength and distribution of air pollutants have modified the oxidative capacity of the troposphere (Emmons et al., 2020). Therefore, an equivalent top-down iterative process to reproduce VSL observations was performed in the current CESM2-SLH version, focused on replicating the CESM1 representation of the VSL SGs vertical profiles as well as the stratospheric bromine and iodine injection obtained with the different model configurations (i.e., free-running or nudged versions). As a result, the original bromine emission fluxes from (Ordóñez et al., 2012) were increased by a constant factor of 1.15 (i.e., 15%) in all locations, which leads to consistency with previous CESM1 results (Fernandez et al., 2014, 2021); see Section 3.2.2). The rationale for increasing the offline emission fluxes is that the reduction of transport across the MBL and the increase near-surface oxidative capacity (dominated by the larger surface OH abundance in CAM6-Chem relative to CAM4-Chem) results in more efficient conversion from SGs to PGs at lower model levels in the current CESM2- SLH version. Note that due to the much shorter lifetime, no scaling was applied to VSL iodocarbons.




**2.2.2 Anthropogenic Offline Halocarbon Emissions**

For the particular case of chlorine, anthropogenic SLH sources are included following the (Hossaini et al., 2019) and (Claxton et al., 2020) emission inventory, which considers time-dependent emissions of two of the dominant organic chlorine species observed in the atmosphere ($CH_2Cl_2$ and $C_2Cl_4$). Based on these works, an increasing trend from 2000 to 2020 is imposed to the emission files for the recent past. In addition, the contribution of $CHCl_3$ and $C_2H_4Cl_2$ to the total tropospheric VSL chlorine loading is considered by imposing surface LBCs for each species (i.e., following the initial approach applied to $CHBr_3$ and $CH_2Br_2$, see Section 2.2.1). Given that the dominant VSL chlorine sources are anthropogenic (between 90-100% for $CH_2Cl_2$, $C_2Cl_4$ and $C_2H_4Cl_2$, with only a significant natural fraction for $CHCl_3$) and presents a pronounced hemispheric asymmetry (WMO, 2022), hereafter we assume all of these offline emissions to have only an anthropogenic origin. Consistent with the scaling factor applied to the (Ordóñez et al., 2012) inventory, both VSLs chlorocarbon surface emissions and LBCs were globally scaled by a constant factor of 1.15 (15% enhancement) globally. The resulting global trend of anthropogenic chlorocarbon emissions implemented in CESM2-SLH is in agreement with current estimates and observations (Hossaini et al., 2019; Claxton et al., 2020; WMO, 2022) (see Section 3.2.2). It should be noted that some additional anthropogenic halocarbon sources, as those arising from the waste treatment of power-plants as well as oceanic aquaculture have been described in the literature (Carpenter et al., 2000; Leedham et al., 2013; Jia et al., 2023) but not yet implemented neither in CESM1 nor in CESM2-SLH. To include the offline anthropogenic VSL halocarbon emissions for chlorine, in addition to increasing the *srf_emiss_specifier* namelist option with $CH_2Cl_2$ and $C_2Cl_4$, users should expand the *flbc_list* to include $CHCl_3$ and $C_2H_4Cl_2$ (see Section 2.5). Differing from natural VSL sources, anthropogenic VSL emissions included in *mo_srf_emissions.F90* present a flat hourly profile (i.e., no diurnal variation is applied).

**2.2.3 Online Emissions of Inorganic Halogens**

In addition to the VSL halocarbon emissions, abiotic sources of inorganic halogens are implemented in the model. Here it is worth noting that these ocean-related sources have different routes for iodine compared to bromine and chlorine, and therefore are described separately. Inorganic iodine (HOI and $I_2$) emissions from the ocean surface are computed online following the ozone-driven oxidation of aqueous iodide occurring at the seawater surface (Carpenter et al., 2013). To include this emission in CESM2-SLH, variables HOI and $I_2$ must be included altogether with the standard *srf_emiss_specifier* used for VSL halocarbons, although the imposed offline file represents just a place-holder as the input values are forced to zero after being read within *mo_srf_emissions.F90* routine and replaced by the online computation. The implementation of this source, which in CESM2-SLH is now performed in a new routine called *iodine_emissions.F90*, has been described in detail in (Prados-Roman et al., 2015b), and is based on the parameterized expressions from (MacDonald et al., 2014). The online emission fluxes for HOI and $I_2$ are computed by Eq. (2a) and (2b), respectively,

$$F_{HOI} = [O_3] \times \left( 4.15 \times 10^5 \times \sqrt{\frac{[I_{aq}^-]}{w}} - \left(\frac{20.6}{w}\right) - 23600 \times \sqrt{[I_{aq}^-]} \right), \tag{2a}$$

$$F_{I2} = [O_3] \times [I_{aq}^-]^{1.3} \times \left( 1.74 \times 10^9 - (6.54 \times 10^8 \times ln(w)) \right), \tag{2b}$$

where $[O_3]$ and $w$ are the surface ozone abundance (ppbv) and wind speed, respectively, computed over the ocean at the lowest model level. $[I_{aq}^-]$ is the aqueous iodide seawater concentration computed by Eq. (2c), which in turn depends on the model SST,




$$\left[I^-_{aq}\right] = 1.46 \times 10^6 \times exp\left(\frac{-9134}{SST}\right)$$ (2c)

Note that the fitting approach applied to the laboratory observations resulted in an unrealistic iodine emission flux for low
wind conditions. Therefore, a wind speed mask imposing $w = 3$ m/s is applied to Eqs. 2a and 2b for all model gridpoints where
$w < 3$ m/s. Not doing so could result in an overestimation of iodine fluxes from the ocean during calm periods (Inamdar et al.,
2020). Finally, we highlight the first order dependence of $F_{HOI}$ and $F_{I2}$ with ozone abundance, which has important
consequences for the long-term trend of iodine emissions during the historical period (see Section 3).
For bromine and chlorine, additional abiotic inorganic source emissions arise from the so-called SSA-dehalogenation
reactions. In this case, the online computation of inorganic halogen emissions assumes an oxidized gas-phase species to be
deposited on the sea-salt aerosol surface, followed by an heterogeneous reaction that captures a reduced halide reservoir within
the substrate that is oxidized before being released to the gas-phase (Fernandez et al., 2014). This complex redox process is
parameterized to proceed in a single-step heterogeneous reaction dependent on the collisional frequency of the gas-phase
species and the substrate, as well as on the degassing efficiency of the halogenated product released to the atmosphere.
Differing from iodine, this recycling process is not included as a typical emission species within *srf_emiss_specifier*, but
instead has been implemented as non-stoichiometric halogen reactions within the *chem_mech.in* based on the Free-Regime
Approximation (FRA) approach (McFiggans et al., 2000) following Eq. (3),

$$F^{XY}_{SSA} = \frac{1}{4} \times \gamma^{ox} \times \left(100 \times \sqrt{\frac{8 \times R \times T}{\pi \times M^{ox}_w}}\right) \times SAD_{SSA} \times [Spec_{ox}] \times DF \times mask_{SSA} \,,$$ (3)

Here, $F^{XY}_{SSA}$ is the flux of inorganic halogens (in the form of $X_2$ or XY, with X,Y = Cl, Br or I) in molec. cm$^{-3}$ s$^{-1}$, $\gamma^{ox}$ represents
the accommodation coefficient of the gas-phase halogenated oxidized reservoir colliding with the SSA, $M_w^{ox}$ is the molecular
weight (g mol$^{-1}$) of the oxidized species used to compute the mean-root square molecular speed, which in turn depends on the
modelled temperature at each gridpoint ($R$ is the universal gas constant and $T$ units are in $K$). Furthermore, the surface area
density of SSA ($SAD_{SSA}$ in cm$^{-2}$ cm$^{-3}$) is computed considering the 3 (three) SSA bins represented in the model, while $[Spec_{ox}]$
is the atmospheric molecular concentration (molec. cm$^{-3}$) of the gas-phase oxidized halogen species (XONO$_2$, XNO$_2$ and/or
HOX, with X = Cl, Br and I) colliding with SSA. Finally, $DF$ is a seasonal-dependent normalized depletion factor that
represents the efficiency of the recycling reaction for any given atmospheric condition, while $mask_{SSA}$ is a logical mask that
limits the occurrence of the SSA-dehalogenation processes above a pressure threshold of 300 hPa. Here it should be noted that
the photochemical degradation of offline VSLs sources described in previous sections constitute the initial step releasing the
halogen atoms that partition between different gas-phase reservoirs that, on a subsequent step, can be deposited over the sea-
salt substrates and being recycled back to the gas phase. Further details on the SSA-dehalogenation implementation in CESM
can be found elsewhere (Ordóñez et al., 2012; Fernandez et al., 2014, 2021).
The FRA for several halogen and nitrogen reservoirs has been implemented in routine *mo_usrrxt.F90* and assumes
that the bromide and chloride content of SSA is sufficiently large to act as an effectively infinite halide reservoir capable of
sustaining the surface heterogeneous-redox reaction. This implies that the bromide and/or chloride content in the SSA bulk is
always in excess compared to the abundance of the gas-phase halogen reservoir that deposits over the aerosol surface, which
is valid for fresh SSA typically found close to the ocean surface. Table 1 summarizes all 9 independent reactions releasing
either gas phase $X_2$ or XY, where the non-stoichiometric coefficients between reactants and products for each gas-phase
halogen species determines the net inorganic halogen flux. For example, when brominated reservoirs are oxidized (*het_ss_0-*
*het_ss_2*), each cycle represents the additional release of 0.65 Br + 0.35 Cl to the gas-phase (in the form of Br$_2$ and BrCl),



while for the case of chlorinated reservoirs, only chloride is captured and released to the gas-phase as $Cl_2$. It should be noted
that for the particular case of iodine reservoirs (*het_ss_6-het_ss_8*), the SSA-dehalogenation process represents a net source
of chlorine and bromine (0.5 Br and 0.5 Cl in the form of IBr and ICl), but only a shift in partitioning among different iodine
species without constituting a net source of inorganic iodine to the atmosphere (i.e., the reaction is stoichiometric for iodine).
Given the large changes in SSA abundance and distribution between CESM1 and CESM2 (see Fig. S1), and with the objective
of reproducing previously estimated SSA-dehalogenation fluxes and burdens obtained in CESM1, the accommodation
coefficients ($\gamma^{ox}$) originally provided by (Ordóñez et al., 2012) have been readjusted within reasonable ranges in CESM2-SLH.
In doing so, we acknowledge that most of the reported $\gamma^{ox}$ values for halogenated reservoirs have very large uncertainties and
are strongly dependent on the composition of the aerosols (Sander, 2015) and therefore the final value considered is largely
based on adjusting the model output with atmospheric observations of individual chlorine, bromine and iodine species (see
Section 3.2.2).

**Table 1: Heterogeneous recycling processes representing a net source of inorganic halogens in CESM2-SLH.**

| Reaction tag | non-stoichiometric reaction | Molecular Weight [$] | Gamma ($\gamma^{ox}$) | net halogen release |
|---|---|---|---|---|
| **Halogenated-reservoirs** | | | | |
| het_ss_0 | $BrONO_2 \rightarrow 0.65*Br_2 + 0.35*BrCl$ | 1.22E+03 | 0.01 | 0.65 Br + 0.35 Cl |
| het_ss_1 | $BrNO_2 \rightarrow 0.65*Br_2 + 0.35*BrCl$ | 1.29E+03 | 0.005 | 0.65 Br + 0.35 Cl |
| het_ss_2 | $HOBr \rightarrow 0.65*Br_2 + 0.35*BrCl$ | 1.47E+03 | 0.0125 | 0.65 Br + 0.35 Cl |
| het_ss_3 | $ClONO_2 \rightarrow CL_2$ | 1.47E+03 | 0.006 | 1.00 Cl |
| het_ss_4 | $ClNO_2 \rightarrow CL_2$ | 1.61E+03 | 0.006 | 1.00 Cl |
| het_ss_5 | $HOCl \rightarrow CL_2$ | 2.01E+03 | 0.03 | 1.00 Cl |
| het_ss_6 | $IONO_2 \rightarrow 0.5*IBr + 0.5*ICl$ | 1.05E+03 | 0.003 | 0.50 Br + 0.50 Cl |
| het_ss_7 | $INO_2 \rightarrow 0.5*IBr + 0.5*ICl$ | 1.10E+03 | 0.006 | 0.50 Br + 0.50 Cl |
| het_ss_8 | $HOI \rightarrow 0.5*IBr + 0.5*ICl$ | 1.21E+03 | 0.0018 | 0.50 Br + 0.50 Cl |
| **Oxidized nitrogen-compounds** | | | | |
| het_ss_9 | $HNO_3 \rightarrow HCl$ | 1.83E+03 | 0.01 | 1.00 Cl |
| het_ss_11 | $N_2O_5 \rightarrow 2*HNO_3$ | 1.40E+03 | 0.0143 | $(1.00 - yield^{ClNO2})$ |
| het_ss_12 | $N_2O_5 \rightarrow ClNO_2 + HNO_3$ | 1.40E+03 | 0.0143 | $yield^{ClNO2}$ |

[$] Values for the molecular weight actually correspond to the following term from Eq. (3), $\left(\sqrt{\frac{8 \times R}{\pi \times M_w^{ox}}}\right)$, where temperature has
been excluded from the square-root and is later multiplied by an independent $\left(\sqrt{T}\right)$ term.

Careful consideration of SSA and other aerosols representation is key across CESM versions. In particular, any
change in the representation of both the abundance and distribution of SSA in the model (e.g. (Danabasoglu et al., 2020) for
CESM2) will have a direct impact on halogen production via SSA dehalogenation, consequently altering the SLH tropospheric
budget and burden. This was indeed the case when shifting from CAM4-Chem to CESM2-CAM6, where the SSA
representation changed from a continuous (bulk) representation considering 4 different bin sizes to a modal aerosol
representation considering only 3 (three) bins, and in addition, the vertical extent of sea-salt in the free-troposphere was largely
reduced to improve the model agreement with observations (Lamarque et al., 2012; Tilmes et al., 2023). Therefore, to easily
adjust the online efficiency of the SSA-dehalogenation flux and avoid changes in the atmospheric burden of inorganic halogens
in the troposphere, the released CESM2-SLH configuration now includes a group of SLH scaling factors (*&slh_nl,*) within the
user-defined namelist (*user_nl_cam*) to allow individual users to adjust the SSA-dehalogenation source in the different model





resolutions and configurations in order to assure a consistent halogen atmospheric loading that are consistent with those shown
in this work (see Section 2.5).
For chlorine, additional inorganic halogen sources arise from the heterogeneous recycling of nitrogen species ($N_2O_5$
and $HNO_3$) on halide-rich SSA (see Eq. 3 and Table 1). In the first case, the SSA uptake and recycling of $N_2O_5$ drives two key
processes: *i)* the well-documented hydrolysis to produce $HNO_3$ (*het_ss_11,* (Lamarque et al., 2012)); and *ii)* the production of
nitryl chloride ($ClNO_2$), which constitutes a net chlorine source (*het_ss_12*). The latter is of importance as $ClNO_2$ photolysis
in polluted environments lead to ozone production (Knipping and Dabdub, 2003; Thornton et al., 2010). Within CESM2-SLH,
the $ClNO_2$ yield is computed online based on a cubic expression dependent on the total sea-salt aerosol mass within each of
the sea-salt aerosol bins (Li et al., 2022). In the second case (*het_ss_9* in Table 1), the acid-displacement reaction $HNO_3 \rightarrow$
HCl represent the acid-uptake of chloride content from the sea-salt bulk, which in turn depends on a fixed $HNO_3$
accommodation coefficient (Sander, 2015), as implemented in other models including chlorine chemistry (Hossaini et al.,
2016). The CESM2-SLH implementation of both chlorine sources follows the FRA-approach described above for the oxidized
halogen reservoirs (Eq. 3), although in this case the gas-phase species that deposit over the sea-salt aerosol and drives the
heterogeneous recycling does not contain an oxidized halogen, but is the oxidized nitrogen species the one that captures the
sea-salt chloride from the aerosol bulk. To allow a direct adjustment of the source-strength for different representations of sea-
salt aerosol loadings, we also include individual *&slh_nl* scaling factors for $N_2O_5$ and $HNO_3$ recycling (see Section 2.5). We
note that stoichiometric $N_2O_5$ recycling over halide-poor tropospheric aerosols is also considered in the model, although for
these reactions the halogen uptake and recycling only constitute a partitioning shift between different halogen and nitrogen
species (see Section 2.3.3 for details).
Finally, we highlight the importance of adjusting the strong model dependence of online SLH sources and burdens
depending on the sea-salt abundances and distributions represented in different model configurations (see Fig. S1). Note that
the accommodation coefficients (γ) and depletion factors (DF) altering the reactive efficiency for the different species reported
in the literature possess a wide range of values and large uncertainties that depend on variables that are unconstrained in the
simplified parameterization implemented in CESM2-SLH (e.g., pH dependence, halide content, etc; (Sander, 2015; Burkholder
et al., 2019). Consequently, variations in SSA fields, pollutant emissions, and atmospheric composition (e.g., $HO_x/NO_x$
partitioning), combined with non-linear heterogeneous recycling (Table 1), may cause regional over-/under-estimations of
halogen content in localized regions in comparison with previous CESM1 studies (see Section 3.2.3).

## 2.3 Short-Lived Halogen Chemistry

In the following subsections, we provide an independent description of the main photochemical (Section 2.3.1), gas-
phase (Section 2.3.2) and heterogeneous-phase (Section 2.3.3) reactions implemented in the default CESM2-SLH chemical
scheme. This scheme builds upon the TS1.2 benchmark chemical mechanism used in CESM2 (Emmons et al., 2020), which
explicitly includes the volatility basis set (VBS) parameterization for secondary organic aerosol formation (Tilmes et al., 2019,
2023). The complete set of SLH reactions added to the TS1-simpleVBS branch within the precompiled *trop_strat_mam4_slh*
chemical pre-processor (chem_mech.in) is included in the Supplementary Information (Tables S1, S2 and S3). Overall, the
final SLH chemical mechanism implemented in CAM-Chem introduces 12 additional chlorine SLH species, resulting in a total
of 34 chlorinated compounds that participate in 67 new reactions. These new reactions are categorized as: photolysis (11),
odd-halogen reactions (6), organic-halogen reactions (20), sulfur-halogen reactions (2), heterogeneous recycling on
tropospheric aerosols (7), sea-salt recycling source (11), and stratospheric mapped reactions (10), where
11+6+20+2+7+11+10=67. Similarly, for bromine we incorporated 9 species on top of 19 brominated compounds to reach a
total of 53 new reactions (respectively distributed as 10+9+8+1+6+6+13=53 to the same categories), while for the case of
iodine, 19 new species and a total of 17+39+2+1+10+6+18=93 reactions were implemented. Note that some of the reactions



involve inter-halogen interactions (e.g., ClO + BrO), and therefore are double-counted within each family. Therefore,
considering all chlorine, bromine and iodine, a total of 31+46+24+4+17+16+25=163 new SLH reactions were implemented.
Finally, we highlight that the same SLH chemical scheme has also been merged into the TSMLT1 mechanism for WACCM
(tag MZ197, *waccm_tsmlt_mam4_slh*).

### 2.3.1 Photochemical Reactions

A comprehensive list of all photochemical halogen reactions included in the SLH scheme is shown in Supplementary
Table S1, based on the original publication compiled in (Ordóñez et al., 2012) and (Saiz-Lopez et al., 2014), with further
updates from (Badia et al., 2021; Saiz-Lopez et al., 2023). This includes 11, 10 and 17 new photolysis reactions for chlorine,
bromine and iodine, respectively, which adds up to previously defined photolysis reactions of long-lived halocarbons as well
as odd-chlorine and odd-bromine reactions that are required for properly representing stratospheric ozone depletion in CESM2
(Gettelman et al., 2019b). The absorption cross sections and quantum yields are taken from the latest JPL (Burkholder et al.,
2019) and IUPAC (Atkinson et al., 2007, 2008) handbooks, and are included within the *xs_long* and *xs_short* file typically
used in CAM-Chem and WACCM. Wavelength integrated *J-values* are computed as a function of temperature and height
considering the model actinic flux and a Look-Up-Table (LUT) approach (Kinnison et al., 2007; Lamarque et al., 2012;
Emmons et al., 2020).
Note that the photodegradation of anthropogenic VSLs ($CHCl_3$, $CH_2Cl_2$ and $C_2Cl_4$) results in the formation of
phosgene ($COCl_2$), which constitutes a halogen intermediate that accumulates in the upper troposphere until it photo-
decomposes in the lower stratosphere (see Fig. S3) (Hossaini et al., 2016). Therefore, phosgene does not participate in direct
ozone destruction in the lower-stratosphere until it releases the Cl atoms through photolysis. However, and given that the only
source of phosgene in the model is the degradation of VSL, we consider the contribution of $COCl_2$ within the PGs chlorine
fraction, in line with the latest (WMO, 2022) report. In addition, some laboratory measurements show that the absorption tail
for the $C_2Cl_4$ cross-section can extend beyond $\lambda > 270$ nm (Keller-Rudek et al., 2013). However, the JPL19-5 handbook
(Burkholder et al., 2019) recommends neglecting the photochemical breakdown above this threshold. Within CESM2-SLH,
forcing to zero the absorption above 270 nm was required to reproduce observations of this particular anthropogenic
halocarbon in the upper troposphere. Not doing so resulted in a complete decomposition of this compound driven by a too
efficient photodissociation all the way down to the surface (Roozitalab et al., 2024). Finally, note that Table S1 considers the
photodissociation of higher order iodine oxides ($I_xO_y$, with x = 2 and y = 2, 3, 4). The absorption cross sections of these species
were derived from solution spectrum measured at the University of Leeds (Lewis et al., 2020), which were subsequently scaled
to estimate absolute absorption cross-sections (Gómez Martín et al., 2005). Therefore, and supported by experimental
measurements (Saiz-Lopez et al., 2014) as well as on several follow-up papers demonstrating that considering $I_xO_y$ photolysis
was required to reproduce gas-phase and particulate iodine observations (Saiz-Lopez et al., 2015; Koenig et al., 2020), the
released CESM2-SLH chemical mechanism is based on the $J_{IxOy}$ scheme for iodine proposed in (Saiz-Lopez et al., 2014).

### 2.3.2 Gas-Phase Reactions

A complete description of all bimolecular Arrhenius type reactions of halogen species are shown in Supplementary
Table S2, where for the sake of simplicity we have ordered reactions in the following groups: Odd-Oxygen, Odd-Chlorine,
Odd-Bromine, Odd-Iodine, VSL halocarbon degradation and reactions with sulphur and carbon compounds. All these
developments have been initially described in (Ordóñez et al., 2012), with additional updates for iodine described in (Saiz-
Lopez et al., 2014). In addition, all termolecular reactions considered in the default SLH chemical mechanism are shown in
Table S3. Note that for the final implementation of SLH chemistry in CESM2, all expressions, reaction-rate coefficients and





temperature dependence factors have been updated to the reported values in the last JPL 19-5 handbook (Burkholder et al.,
2019).

Oxidation of DMS to produce sulphur dioxide ($SO_2$) by BrO, Cl and IO are also included in the updated SLH chemical
scheme. Here, we highlight that improvements in sulphur chemistry representation in CESM, including $SO_2$ and tropospheric
sulphate formation and washout, are a major source of uncertainty for radiative forcing estimations (Ge et al., 2022) and are
currently under development. Therefore we decided to maintain in the updated SLH mechanism the benchmark sulphur scheme
implemented in TS1.2 (Emmons et al., 2020). Similarly, CESM2-SLH uses the Modal Aerosol Model with 4 modes (MAM4)
scheme, which computes the formation and growth of sulphate, black carbon and organic matter, secondary organic aerosols,
sea salt, and dust  (Liu et al., 2016). MAM4 is the default aerosol scheme in CESM2 and has not been updated within the SLH
scheme, although the perturbations of reactive halogens on OH abundance can indirectly influence the aerosol burden (Saiz-
Lopez et al., 2023). In the following section, we describe the direct heterogeneous processes involving inorganic halogens.

**2.3.3 Heterogeneous Reactions**

Heterogeneous SLH reactions occurring over different types of atmospheric substrates have also been implemented
(see Table 2). It is worth noting that unlike the SSA-dehalogenation source described in Section 2.2.3, all of these reactions
are stoichiometric and therefore do not represent a net halogen source from the aerosol to the gas-phase, but instead, result in
a change in individual species partitioning between gaseous reactants and products. Indeed, in all cases, the substrate surface
(e.g., ice-crystals, cloud droplets and/or other anthropogenic aerosols typically considered in CAM6 like black-carbon, organic
carbon, sulphate, nitrate, etc.) acts like a catalyst and does not contribute with any halogen content. For heterogeneous reactions
occurring over ice-crystals, we compute the reaction rate constant ($rate_{ICE}$), using an equivalent FRA approach. This method
accounts for the total number of gas-phase species that are assumed to be uptake by the aerosol surface, resulting in the
following expressions for uni-molecular

$$rate_{ICE}^{uni} = \frac{1}{4} \times \gamma^X \times \left( 100 \times \sqrt{\frac{8 \times R \times T}{\pi \times M_w^X}} \right) \times SAD_{ICE} \times mask_{ICE-strat} \ , \qquad (4a)$$

and bimolecular

$$rate_{ICE}^{bi} = \frac{1}{4} \times \gamma^X \times \left( 100 \times \sqrt{\frac{8 \times R \times T}{\pi \times M_w^X}} \right) \times SAD_{ICE} \times mask_{ICE-strat} \times \frac{1}{[Y]} \ , \qquad (4b)$$

reactions shown in Table 2. Here, $\gamma^x$ and $M_w^X$ are, respectively, the accommodation coefficient and the molecular weight of
the gas-phase halogenated reservoir species (X) that is initially taken up by the ice-crystal, $SAD_{ICE}$ represents the surface area
density of ice crystals and is computed online based on the ice-water content in clouds (CLDICE) from the CAM6 model, and
$mask_{ICE-strat}$ is a logical mask imposed to limit the computation below the model tropopause and avoid double-counting
recycling reactions historically implemented to occur in the stratosphere (Kinnison et al., 2007; Fernandez et al., 2014). All
these developments are included in the original *mo_usrrxt.F90* routine. For bimolecular reactions, note that the reaction rate
is normalized by the atmospheric concentration of the most abundant halogenated species (either X or Y) involved in the
reaction. The original implementation of these reactions has been described in detail in (Fernandez et al., 2014) for bromine
and chlorine and in (Saiz-Lopez et al., 2015) for the case of iodine.





The heterogeneous recycling reaction of $N_2O_5$ shown in Table 2 takes place on tropospheric aerosols in continental
regions. In contrast to reaction *het_ss_12* in Table 1 that occurs only over sea-salt aerosols, the tagged reaction *usr_N2O5_aer2*
in Table 2 can also occur over tropospheric aerosols over continental domains, which produces $ClNO_2$ that can further
photolyze to release $NO_2$ and consequently result in additional ozone production within polluted environments (i.e., high-$NO_x$
regimes). However, and given that the model aerosol components within MAM4 (i.e., black-carbon, organic-carbon, sulphate
and/or nitrate) do not represent a halide reservoir, reaction *usr_N2O5_aer2* assumes that the reduced chlorine atom that is
oxidized over the substrate surface must be initially captured by the aerosol in the form of HCl, which further react at the
aerosol surface to produce $ClNO_2$ that is released back to the gas phase (reactive uptake). For this particular process, the
heterogeneous rate constant is computed following the standard *hetrxtrate* function in *mo_usrrxt.F90* (Lamarque et al., 2012),
which considers the accommodation coefficients and production yields from (McDuffie et al., 2018, 2019). The
implementation of the photochemical recycling of nitrogen oxides is based on (Li et al., 2022) following the $HNO_3$ acid-
displacement developments from (Hossaini et al., 2016).

**Table 2: Heterogeneous recycling processes representing a change in partitioning of inorganic halogens in CESM2-SLH.**

| tag | stoichiometric reaction | Molecular Weight $^\$$ | Gamma ($\gamma^{ox}$) |
|---|---|---|---|
| ice-crystals | | | |
| ice_trp_br_1 | $BrONO_2$ -> $HOBr$ + $HNO_3$ | 1.22E+03 | 0.3 |
| ice_trp_cl_1 | $ClONO_2$ -> $HOCl$ + $HNO_3$ | 1.47E+03 | 0.1 |
| ice_trp_hbr_5 | $HOCl$ + $HBr$ -> $BrCl$ + $H_2O$ | 2.01E+03 | 0.2 |
| ice_trp_hbr_6 | $HOBr$ + $HBr$ -> $Br_2$ + $H_2O$ | 1.47E+03 | 0.12 |
| ice_trp_hcl_5 | $HOCl$ + $HCl$ -> $Cl_2$ + $H_2O$ | 2.01E+03 | 0.2 |
| ice_trp_hcl_6 | $HOBr$ + $HCl$ -> $BrCl$ + $H_2O$ | 1.47E+03 | 0.3 |
| ice_trp_hi_5 | $HOCl$ + $HI$ -> $ICl$ + $H_2O$ | 2.01E+03 | 0.12 |
| ice_trp_hi_6 | $HOBr$ + $HI$ -> $IBr$ + $H_2O$ | 1.47E+03 | 0.12 |
| ice_trp_i_1 | $IONO_2$ -> $HOI$ + $HNO_3$ | 1.06E+03 | 0.1 |
| ice_trp_i_2 | $HOI$ + $HCl$ -> $ICl$ + $H_2O$ | 1.21E+03 | 0.12 |
| ice_trp_i_3 | $HOI$ + $HBr$ -> $IBr$ + $H_2O$ | 1.21E+03 | 0.12 |
| ice_trp_i_4 | $HOI$ + $HI$ -> $I_2$ + $H_2O$ | 1.21E+03 | 0.12 |
| tropospheric aerosols | | | |
| usr_N2O5_aer2$^\&$ | $N_2O_5$ + $HCl$ -> $ClNO_2$ + $HNO_3$ | 1.40E+03 | 0.02 |

$^\$$ Values for the molecular weight correspond to the following term from Eq. (3), $\left(\sqrt{\frac{8 \times R}{\pi \times M_w^{ox}}}\right)$, where
temperature has been excluded from the square-root and is later multiplied by an independent $\left(\sqrt{T}\right)$ term.
$^\&$ A reaction yield of 0.138 is considered for $ClNO_2$ production (Li et al., 2022).

Regarding stratospheric heterogeneous reactions involving halogenated reservoirs that are important for the ozone
layer, we complemented a full set of SLH reactions mapping the historical implementation performed in *mo_strato_rates.F90*
for WACCM (Marsh et al., 2013; Fernandez et al., 2017; Cuevas et al., 2022). This includes re-activation reactions occurring
over stratospheric sulphate ($SAD_{SULFC}$), ice-crystals ($SAD_{ICE}$) and nitric acid trihydrate ($SAD_{NAT}$) surfaces (Kinnison et al.,
2007), for which independent accommodation coefficients ($\gamma$) have been computed for the main reactions involving the major
chlorinated and brominated reservoirs, many of them including a logical condition of humidity and/or temperature-dependence
updates (Solomon et al., 2015). Based on (Cuevas et al., 2022), we extended the same approach to consider equivalent
heterogeneous reactions in the stratosphere for the minor halogenated reservoirs (mainly $HOCl$ and $HOBr$), as well as the



previously neglected iodine species (see Table 3). In doing so, and despite laboratory measurements suggesting that the reactive
efficiencies of iodine species are larger than those for bromine and chlorine (Solomon et al., 1994; Koenig et al., 2020), we
assumed identical accommodation coefficients and reaction yields for chlorine and bromine as those originally computed in
the default WACCM scheme. Based on the original implementation in CESM1 (Fernandez et al., 2017; Saiz-Lopez et al.,
2023), for FC compsets we imposed prescribed stratospheric sulphate aerosols above the tropopause (Mills et al., 2016), which
results in smaller $SAD_{SULFC}$ fields compared to those computed with MAM4 (see Fig. S2). Therefore, the modelled impact of
these reactions in the lower stratosphere should represent a lower limit compared to WACCM (Cuevas et al., 2022). Note that
equivalent logical conditions based on $mask_{ICE\text{-}strat}$ have also been applied to all equivalent tropospheric iodine reactions
occurring over tropospheric ice-crystals to avoid double counting.

**Table 3: Update of stratospheric heterogeneous reactions for minor bromine and iodine species in CESM2-SLH.**

| het-recycling reaction | Substrate/ tag | gamma | Substrate/ tag | gamma | Substrate/ tag | gamma |
|---|---|---|---|---|---|---|
| mapped to chlorine | sulphate | | NAT | | Water-ice | |
| $HOCl + HCl \rightarrow Cl_2 + H_2O$ | het5 | f(T,P,HCl,HOCl,H2O,r) | het10 | 0.1 | het16 | 0.2 |
| **$HOCl + HBr \rightarrow BrCl + H_2O$** | **het5_hbr** | **f(T,P,HCl,HOCl,H2O,r)** | **het10_hbr** | **0.1** | **het16_hbr** | **0.2** |
| **$HOCl + HI \rightarrow ICl + H_2O$** | **het5_hi** | **f(T,P,HCl,HOCl,H2O,r)** | **het10_hi** | **0.1** | **het16_hi** | **0.2** |
| mapped to bromine | | | | | | |
| $HOBr + HCl \rightarrow BrCl + H_2O^\$$ | het6 | f(T,P,HCl,HOBr,H2O,r) | **het10b** | **0.1** | het17 | 0.3 |
| **$HOBr + HBr \rightarrow Br_2 + H_2O$** | **het6_hbr** | **f(T,P,HCl,HOBr,H2O,r)** | **het10bhbr** | **0.1** | **het17_hbr** | **0.3** |
| **$HOBr + HI \rightarrow IBr + H_2O$** | **het6_hi** | **f(T,P,HCl,HOBr,H2O,r)** | **het10bhi** | **0.1** | **het17_hi** | **0.3** |
| new iodine (mapped to bromine) | | | | | | |
| $BrONO_2 \rightarrow HOBr + HNO_3$ | het3 | f(T,P,H2O,r) | het11 | 0.006 | het14 | 0.3 |
| $IONO_2 \rightarrow HOI + HNO_3$ | **slf_str_i_1** | **f(T,P,H2O,r)** | **nat_str_i_1** | **0.006** | **ice_str_i_1** | **0.3** |
| $HOI + HCl \rightarrow ICl + H_2O$ | **slf_str_i_2** | **f(T,P,HCl,HOBr,H2O,r)** | **nat_str_i_2** | **0.1** | **ice_str_i_2** | **0.3** |
| $HOI + HBr \rightarrow IBr + H_2O$ | **slf_str_i_3** | **f(T,P,HCl,HOBr,H2O,r)** | **nat_str_i_3** | **0.1** | **ice_str_i_3** | **0.3** |
| $HOI + HI \rightarrow I_2 + H_2O$ | **slf_str_i_4** | **f(T,P,HCl,HOBr,H2O,r)** | **nat_str_i_4** | **0.1** | **ice_str_i_4** | **0.3** |


Bold font style highlights the mapped reactions and gammas applied to the corresponding regular font expression originally implemented in
(Marsh et al., 2013). For chlorine and bromine reactions we extended the original reaction tag-name, while for iodine expressions we adopted
an independent tag-convention indicative of the substrate where each reaction occurs.
$\$$ For the case of $HOBr + HCl \rightarrow BrCl + H_2O$ reaction only tag het10b was updated on top of the original scheme.

**2.4 Species-specific dry and wet deposition**
The implementation of species-independent dry and wet deposition processes in CESM1 CAM4-Chem has been
previously described in the literature (Ordóñez et al., 2012; Fernandez et al., 2014; Saiz-Lopez et al., 2014). Here we briefly
summarize the main approaches used to compute the removal or sink of SLH from the atmosphere, focusing on the few cases
where a different approach was implemented in CESM2-SLH. Note that the model only considers the removal of inorganic
halogen species, and therefore the sinks of VSL halocarbons are only chemical while for inorganic PGs deposition represents
the major sink.





**Table 4: Independent halogen species considered for dry and wet deposition in CESM2-SLH.**

| species | Dry deposition velocity (m s$^{-1}$) | Wet deposition solubility ($K_H$, M atm$^{-1}$) | Ice_uptake[&] |
|---|---|---|---|
| **Chlorine** | | | |
| HCl | 2.00E-02 | 1.54E+00 | YES |
| HOCl | 1.00E-02 | 9.30E+02 | YES |
| $ClONO_2$ | 1.00E-02 | 1.00E+06 | YES |
| $ClNO_2$ | 5.00E-03 | 3.50E-02 | YES |
| $COCl_2$ | 1.00E-02 | 5.90E-04 | |
| $CHCl_2O_2$ | 1.00E-02 | 1.70E-03 | |
| **Bromine** | | | |
| HBr | 2.00E-02 | 7.20E-01 | |
| HOBr | 1.60E-02 | 1.90E+03 | |
| $BrONO_2$ | 5.00E-03 | 1.00E+06 | FRA |
| $BrNO_2$ | 5.00E-03 | 3.00E-01 | YES |
| $Br_2$ | 1.00E-02 | 7.60E-01 | |
| BrCl | | 9.40E-01 | |
| **Iodine** | | | |
| IBr | 1.00E-02 | 2.40E+01 | |
| ICl | 1.00E-02 | 1.10E+02 | |
| HI | 1.00E-02 | 7.80E-01 | FRA |
| HOI | 7.50E-03 | FRA | FRA |
| $IONO_2$ | 7.50E-03 | 1.00E+06 | FRA |
| $INO_2$ | 7.50E-03 | 3.00E-01 | |
| IO | | 4.50E+02 | |
| OIO | | 1.00E+04 | |
| $I_2O_2$ | 1.00E-02 | 1.00E+04 | YES |
| $I_2O_3$ | 1.00E-02 | 1.00E+04 | YES |
| $I_2O_4$ | 1.00E-02 | 1.00E+04 | YES |

[&] FRA indicate that for that particular case, the Free Regime Approximation was
applied, see Table 5.

The depositional flux of gas-phase halogenated inorganic species due to dry deposition is calculated as the product

of the deposition velocity of each individual species times its concentration at the lowest model surface (Ordóñez et al., 2012),
following the original implementation within *mo_drydep.F90* in CAM-Chem (Lamarque et al., 2012). Table 4 summarises the
complete list of gas-phase species that are considered in the *drydep_list*, and shows the individual deposition velocity for each
species. In addition, inorganic halogens are also removed by wet-deposition, following the default NEU scheme (Neu and
Prather, 2012). Both nucleation scavenging (rainout) and impaction scavenging (below-cloud washout) are implemented in
the wet-removal schemes based on (Lamarque et al., 2012), although it is worth noting that we updated the
*mo_neu_wetdep.F90* routine to avoid mapping the ice-uptake of halogen species to that of $HNO_3$, as this resulted in too
efficient washout of bromine and iodine in the upper troposphere (Fernandez et al., 2014). This was achieved by including the
additional variable *gas_wetdep_ice_uptake_list* within the *&wetdep_inparm* namelist group (see Section 2.5). The individual



Henry law coefficients ($k_H$) for all chlorine, bromine and iodine species shown in Table 4 are mostly based on the compilation
of Henry Laws constants from (Sander, 2015) and IUPAC (Atkinson et al., 2007, 2008).

**Table 5: Free Regime Approximation reactions representing a net sink of inorganic halogens in CESM2-SLH.**

| tag | het-recycling washout | Molecular Weight [$] | Gamma ($\gamma^{ox}$) | type of surface |
|-----|----------------------|----------------------|----------------------|-----------------|
| | Iodine | | | |
| ice_fr_hi | HI -> | 1.29E+03 | 0.02 | $SAD_{ICE}$ |
| ice_fr_hoi | HOI -> | 1.21E+03 | 3.00E-04 | $SAD_{ICE}$ |
| ice_fr_iono2 | IONO$_2$ -> | 1.06E+03 | 0.02 | $SAD_{ICE}$ |
| liq_fr_hoi | HOI -> | 1.21E+03 | 3.00E-04 | $SAD_{LIQ}$ |
| | | | | |
| ss_ixoy_2 | I$_2$O$_2$ -> | 8.61E+02 | 0.0025 | $SAD_{SSA}$ |
| ss_ixoy_3 | I$_2$O$_3$ -> | 8.38E+02 | 0.0025 | $SAD_{SSA}$ |
| ss_ixoy_4 | I$_2$O$_4$ -> | 8.16E+02 | 0.0025 | $SAD_{SSA}$ |
| | | | | |
| | Bromine | | | |
| ice_fr_brono2 | BrONO$_2$ -> | 1.22E+03 | 1.00E-02 | $SAD_{ICE}$ |

[$] Values for the molecular weight actually correspond to the following term from Eq. (3), $\left(\sqrt{\frac{8 \times R}{\pi \times M_w^{ox}}}\right)$, where temperature has
been excluded from the square-root and is later multiplied by an independent $\left(\sqrt{T}\right)$ term.

Halogenated reservoir species are assumed to be adsorbed by liquid droplets and ice-crystals, where they can undergo
either: *i)* reactive uptake (chemical recycling and re-emission back to the atmosphere) or ii) permanent removal from the gas
phase through washout (substrate capture/adsorption). Consequently, the modelled total inorganic halogen loading (Cl$_y$, Br$_y$
and I$_y$), particularly in the Tropical Tropopause Layer (TTL), is determined by the competition between the wet deposition
efficiency relative to the heterogeneous recycling (Aschmann et al., 2011; Aschmann and Sinnhuber, 2013). For iodine, (Saiz-
Lopez et al., 2014, 2015) determined that it was not possible to reproduce IO observations in the free troposphere because the
NEU scheme washout of major iodine reservoirs was too efficient and iodine was completely removed within the lower
troposphere. Therefore, for the particular case of HOI, HI and IONO$_2$, we calculate wet deposition with the FRA unimolecular
approach for the dominant I$_y$ species, considering Eq. (4a) to determine the collisional frequency. These ice-uptake processes
are assumed to lead to deposition of iodine from the atmosphere (see non-stoichiometric removal reactions in Table 5). For
HOI, which is the dominant iodine species in the lower troposphere, the FRA was implemented for both liquid-droplets and
ice-crystals, where the former considers the surface area density of liquid clouds ($SAD_{LIQ}$) and the latter considers $SAD_{ICE}$
(Saiz-Lopez et al., 2014). Note that the non-reactive uptake or substrate capture of higher-order iodine oxides is also assumed
to proceed efficiently on sea-salt aerosols following the FRA approach (Saiz-Lopez et al., 2015), although due to the efficient
I$_x$O$_y$ photolysis this additional sink of atmospheric iodine is a minor contributor compared to scavenging of major I$_y$ species.
Finally, we note that the ice-uptake for BrONO$_2$, one of the most abundant brominated reservoirs in the TTL and lower
stratosphere, followed the NEU scheme and was assumed to be infinitely efficient in CESM1 (Fernandez et al., 2014).
Following the FRA approach implemented for IONO$_2$, we have now implemented the FRA for BrONO$_2$ over ice-crystals in
CESM2-SLH, which led to better agreement of the contribution of inorganic bromine to the total stratospheric bromine budget
within tropical regions (see Section 3.2.2).



**2.5 CESM2-SLH release**

The implementation of short-lived halogen sources and chemistry was performed on top of version 2.2.0 of CESM2 (Danabasoglu et al., 2020), particularly over branch *cesm2.2-asdbranch*, which includes CAM6 tagged version *cam_cesm2_2_rel_09*. Based on this, we created additional forks, respectively called *cesm2.2-asdbranch_slh* and *cam_cesm2_2_rel_09_slh*, respectively, which incorporate SLH updates within the main FORTRAN routines of each CESM2 component, as well as necessary modifications to configuration files (e.g., *cime* and *cime_config*) and default namelist variables (see Chart S1 in the Supplementary Material). These changes enable the direct building and/or cloning of different SLH compsets. In particular, the updates in the CESM2-SLH release include: *i)* increasing the total number of halogenated species considered in the chemical mechanism, for some of which various species independent deposition velocity and Henry coefficients were required; *ii)* extending the number of species for which off-line and on-line emissions are considered, as well as those that are included as LBCs; *iii)* replacing the absorption cross-section files to include new SLH species that are not available in the default CESM2 files; *iv)* incorporating a new namelist group section for SLH (*&slh_nl*) along with additional namelist variables to consider independent ice-uptake efficiencies for some halogen species; *v)* expanding the total number of default namelist files that provide individual namelist values for the different SLH configurations and user's cases; *vi)* including pre-compiled chemical mechanism with all SLH updates for CAM-Chem and WACCM; and *vii)* mapping and updating the original compsets available in the benchmark CESM2.2.0 version to directly create and compile the different CESM2-SLH model cases. Section 2.5.1 describes the main changes implemented on each of the SLH compsets as well as the most important namelist updates and setup options that are required to properly configure a CESM2-SLH run. Finally, Section 2.5.2 summarizes the few SLH developments that have not been included in the present release, and highlights the associated limitations.

**2.5.1 Available Compsets and Resolutions**

Two main CESM2-SLH compsets have been developed based on the original CESM2 configurations: the *FCHIST_slh* and *FCnudged_slh* compsets. The default and suggested configurations that have been scientifically validated considers the coarse *f19_f19_mg17* (1.9º latitude × 2.5º longitude, hereafter 2°×2°) resolution and 32 vertical levels from the surface to approximately 40 km (4 hPa) during the 1980-2020 period. Unless stated otherwise, all external forcings and namelist options, including prescribed ocean SST and ice-coverage fields as well as solar radiation, cloud microphysics, gravity-wave dragging and dust emission factors are maintained identical to the default *FCHIST* and *FCnudged* parent's configuration (Danabasoglu et al., 2020; Emmons et al., 2020). To allow performing model simulations beyond the historical period, prescribed emission and LBC files based on the Climate Model Intercomparison Project – Phase 6 (CMIP6) data until year 2015, followed by the CMIP6 Shared Socio-economic Pathway 3.70 (SSP-370) scenario during the period 2016-2100 (Meinshausen et al., 2017; IPCC, 2022) were concatenated. Despite none of them have been used in this work, we note that additional emission and LBC files for other configurations using different SSP scenarios are available and can be easily changed using user-defined namelist options (see below). In addition, equivalent model configurations using a finer resolution *f09_f09_mg17* grid (0.9º latitude × 1.25º longitude, hereafter 1°×1°) have also been tested and adjusted to reproduce results in comparison with the (2°×2°) grids, although we recommend new users to select the coarse resolution setup as many of the online photochemical sources and recycling reactions described in Sections 2.2 and 2.3 depends on highly variable and resolution-dependent atmospheric fields, such as $SAD_{SSA}$ and $SAD_{ICE}$. Similarly, we highlight that the existent resolution-related variability in lightning-NOx production within CESM2 (Emmons et al., 2020; Wild et al., 2020) significantly impact on the chemical partitioning and washout efficiency of SLH. Therefore, caution should be taken when shifting any SLH compset from a coarse to a fine resolution, as the resulting halogen abundances can significantly vary between them. Finally,



a fully-coupled *f19_g16 BWHIST_slh* as well as a whole atmosphere *f19_f19_mg17 FWnudged_slh* setups, both of them at (2º×2º) resolution and with 70 vertical levels, were also mapped to their corresponding WACCM compsets. Table 6 summarizes the names and configuration of the main CESM2-SLH compsets described in this work, as well as the time periods and conditions in which they have been tested and validated.

It is worth highlighting that most of the previous SLH studies based on CESM1 focused on a direct quantification of the changes in atmospheric composition by comparing a sensitivity including (*compset_slh*) and neglecting (*compset_noh*) short-lived halogen chemistry. However, the current CESM2-SLH release includes only the complete SLH configuration (e.g., *compset_slh*), and therefore, any study willing to perform an SLH vs. noSLH inter-comparison must configure the corresponding *compset_noh* setup. For example, note that the default CESM2 compset *FCHIST* considers LBCs for the two main bromocarbons (CHBr$_3$ and CH$_2$Br$_2$) in order to achieve a consistent stratospheric bromine loading, but the corresponding CESM2-SLH *FCHIST_slh* compset has replaced those LBCs by offline emissions files (see Section 2.2.1). Consequently, comparing *FCHIST_slh* vs. its parent *FCHIST* compset will not allow to address the impact of including (or not) CHBr$_3$ and CH$_2$Br$_2$, as both species are included in both configurations although with a different approach. Thus, new CESM2-SLH users willing to run a *FCHIST_noh* sensitivity, must start from the *FCHIST_slh* configuration, and then disable SLH sources, sinks, and /or chemical reactions involved according to their needs and the particular case and/or halogen family they are studying.

**Table 6: Summary of the different model compsets and configurations implemented in CESM2-SLH.**

| Simulation name | SLH compset | parent's compset | resolution (lat × lon) | Vertical levels | chem_mech.in (chem_proc) | Period of time evaluated |
|---|---|---|---|---|---|---|
| **suggested** | | | | | | |
| FCnudged_slh | Fcnudged_slh | FCnudged | (1.9º × 2.5º) | 32L | trop_strat_mam4_slh | 1980 - 2020 |
| FCHIST_slh | FCHIST_slh | FCHIST | (1.9º × 2.5º) | 32L | trop_strat_mam4_slh | 1950 - 2020 |
| | | | | | | |
| **available** | | | | | | |
| FCnudged_slh | Fcnudged_slh | FCnudged | (0.9º × 1.25º) | 32L | trop_strat_mam4_slh | 2000-2005 |
| FCHIST_slh | FCHIST_slh | FCHIST | (0.9º × 1.25º) | 32L | trop_strat_mam4_slh | 2000-2005 |
| FWnudged_slh | FWnudged_slh | FWSD | (1.9º × 2.5º) | 70L | waccm_tsmlt_mam4_slh | 2000-2005 |
| BWHIST_slh | BWHIST_slh | BWHIST_BCG | (1.9º × 2.5º) | 70L | waccm_tsmlt_mam4_slh | 2000-2005 |

The main changes that must be applied to the default namelist variables are listed below (a default SLH *user_nl_cam* highlighting the required modifications is provided in Chart S2 in the Supplementary Material):

- **Initialization file:** The initial condition (*ncdata*) data has been replaced to ensure the stabilization of all SLH species and atmospheric compounds. Note that while 3 years of spin-up are sufficient to achieve tropospheric stabilization of SLH SGs and PGs, we recommend performing at least 7-10 years of spin-up for complete stratospheric stabilization when starting from an initial condition that neglects SLH.
- **Offline emissions:** The *srf_emis_specifier* within the *&chem_inparm* group has been expanded to include the offline emissions of CHBr$_3$, CH$_2$Br$_2$, CH$_2$BrCl, CHBr$_2$Cl, CHBrCl$_2$, CH$_3$I, CH$_2$I$_2$, CH$_2$IBr, CH$_2$ICl, CH$_2$Cl$_2$, C$_2$Cl$_4$, I$_2$, and HOI. Note that although HOI and I2 are included in this list, this serves just as a placeholder as their emissions are forced to zero after being read and are subsequently computed online within the *iodine_emissions.F90* routine.
- **Boundary conditions:** CH$_3$Cl and C$_2$H$_4$Cl$_2$ have been included in the *flbc_list* within the *&chem_surfvals_nl* group, as these species contribute to the atmospheric SLH chlorine loading. An updated *flbc_file* including the projected trend of these anthropogenic VSLs for the different SSPs has also been developed.





- **Absorption cross-sections:** To compute the photolysis of gas-phase organic and inorganic halogen species, wavelength-dependent absorption data of several SLH have been added to the default short-wave (*xs_short_file*) and temperature-dependent long-wave (*xs_long_file*) cross-section files within *&chem_inparm*.

- **Dry deposition:** The following species have been included in the *drydep_list* within *&drydep_inparm*: $\underline{ClONO_2}$, $\underline{HCl}$, $\underline{HOCl}$, $ClNO_2$, $\underline{BrONO_2}$, $\underline{HBr}$, $\underline{HOBr}$, $BrNO_2$, $Br_2$, $IONO_2$, HI, HOI, $INO_2$, $I_2O_2$, $I_2O_3$, $I_2O_4$, $CHCl_2O_2$, and $COCl_2$. The underlined species were already included in the default CESM2 setup.

- **Wet deposition:** The *gas_wetdep_list* within *&wetdep_inparm* has also been extended with the following species: $\underline{ClONO_2}$, $\underline{HCl}$, $\underline{HOCl}$, $\underline{COFCl}$, $\underline{BrONO_2}$, $\underline{HBr}$, $\underline{HOBr}$, $ClNO_2$, $BrNO_2$, $Br_2$, BrCl, $IONO_2$, $INO_2$, HI, IO, OIO, ICl, IBr, $I_2O_2$, $I_2O_3$, $I_2O_4$, $CHCl_2O_2$, and $COCl_2$. Additionally, the new namelist variable *gas_wetdep_ice_uptake_list* was included with the following halogen variables: $\underline{HNO_3}$, $ClONO_2$, HCl, HOCl, $BrNO_2$, $ClNO_2$, $I_2O_2$, $I_2O_3$ and $I_2O_4$,. Note that *ice_uptake = .true.* is always imposed for $HNO_3$ within *mo_neu_wetdep.F90* by default.

- **SLH scaling factors:** The new *&slh_nl* namelist section was incorporated, providing a set of scaling factors intended to adjust the recycling efficiency of the main processes affecting SLH sources and sinks when shifting between configurations.

**Table 7: Adjusted values of *&slh_nl* scaling factors for each of the main CESM2-SLH compsets and resolutions.**

| compset | FCnudged_slh | FCHIST_slh | FCnudged_slh | FCHIST_slh | Fwnudged_slh | BWHIST_slh |
|---|---|---|---|---|---|---|
| resolution | $(1.9° \times 2.5°)$ | $(1.9° \times 2.5°)$ | $(0.9° \times 1.25°)$ | $(0.9° \times 1.25°)$ | $(1.9° \times 2.5°)$ | $(1.9° \times 2.5°)$ |
| **SSA-Sources** | | | | | | |
| SSAdehal_ScalingFactor | 1.4 | 1.7 | 1.5 | 1.5 | 1.7 | 1.7 |
| SSAhno3_ScalingFactor | 1 | 1 | 1 | 1 | 1 | 1 |
| SSAn2o5_ScalingFactor | 1 | 1 | 1 | 1 | 1 | 1 |
| **FRA-Sinks** | | | | | | |
| LIQfraprx_ScalingFactor_I | 1.5 | 1 | 1.5 | 1 | 1 | 1 |
| ICEfraprx_ScalingFactor_I | 1.3 | 1.7 | 0.9 | 0.9 | 1.9 | 2.8 |
| ICEfraprx_ScalingFactor_Br | 1 | 2 | 1 | 1 | 3 | 3 |

The calibrated *&slh_nl* scaling factors for the main CESM2-SLH compsets and resolutions are provided in Table 7. In case other model compsets and/or resolutions are used, it is the user´s responsibility to evaluate the global halogen loading in comparison with the validated *FCHIST_slh* and *FCnudged_slh* (2°×2°) results. A comparison of the SLH changes within the main configurations and resolutions considered in this work are shown in Section 3.4 below.

## 2.5.2 SLH Developments not Implemented in CESM2

As described in Section 2.1, the initial implementation and the subsequent updates of SLH sources and chemistry in CESM1 were performed in consecutive studies focused on different regions of the atmosphere and/or different processes affecting each of the individual halogen families within the complete SLH scheme. Simultaneously, non-SLH related model developments in different versions of CESM were also implemented, which have introduced variations in the distribution and impacts of SLH compared to the initial studies. Therefore, with the intention of: *i)* including the most scientifically-validated version of the SLH influence on atmospheric composition within the latest released version of CESM; and *ii)* keeping the closest model setup to the current CESM2 configurations typically used for climate and air quality studies on the global scale; the following SLH developments from CESM1 have not been included in the released CESM2-SLH version. The rationale



behind these exclusions and their potential implications for understanding SLH impacts on atmospheric composition are detailed below:

- **Emission-driven Methane simulations:** Despite (Li et al., 2022) highlighted the importance of performing emission-driven CH$_4$ simulations to properly evaluate the SLH influence on methane burden and radiative forcing, all default CESM2 configurations consider CH$_4$ LBCs altogether with other long-lived halogenated ODS and greenhouse gases (N$_2$O and CO$_2$). Consequently, current CESM2-SLH configuration does not allow to evaluate the direct (Cl-driven) and indirect (OH-driven) impacts of SLH chemistry on methane lifetime and radiative balance as those described in (Li et al., 2022) and (Saiz-Lopez et al., 2023).

- **Continental Inorganic Halogen emissions:** In addition to the anthropogenic halocarbon emissions of VSL chlorine, an anthropogenic emission inventory of continental inorganic halogens arising mostly from coal-burning and other sectors was developed in (Saiz-Lopez et al., 2023). Despite this inventory may be of significant relevance for regional or global air quality studies (Chang et al., 2024; Fu et al., 2024), they do not contribute to the stratospheric halogen loading while in addition it has been found that the SLH influence on global atmospheric composition is dominated by natural SLH (the so called AANE sensitivity in (Saiz-Lopez et al., 2023).

- **Polar sea-ice halogen sources and chemistry:** Given the significant architectural code changes in many model components between CESM1 and CESM2 (Danabasoglu et al., 2020) the inclusion of sea-ice halogen emissions and chemistry from the polar regions (Fernandez et al., 2019, 2024) have not been completely tested and evaluated, particularly for the southern hemisphere, and therefore are not considered in current CESM2-SLH. This omission includes other sea-ice related halocarbon emissions as those described in (Abrahamsson et al., 2018).

- **Halogen sources from dust:** The photocatalytic chlorine production (van Herpen et al., 2023) and iodine release (Koenig et al., 2021) from dust have also not been implemented in the released CESM2-SLH version. We believe that further research is required before a complete representation of these complex heterogeneous redox processes is parameterized and included alongside the rest of the widely validated SLH developments.

- **Expanded sulphur chemistry scheme:** This include MeSH emissions and chemical processing reported in (Wohl et al., 2024), as well as the hydroperoxymethyl thioformate (HPMTF) updated chemical scheme of DMS oxidation reported in (Veres et al., 2020). Despite the important role played by SLH in the atmospheric sulfur cycle, these CESM1 additions are not directly related to SLH chemistry. Therefore, we have decided not to include them in the final CESM2-SLH release, maintaining the default CESM2 sulphur chemistry.

## 3 Results

This section presents a comprehensive description of the main SLH effects on atmospheric composition resulting from the new CESM2-SLH model setup, and is oriented to provide a general view for new users not familiarized with SLH chemistry. We focus on mean tropical, global, zonal and surface quantities, as well as their integrated values within the troposphere and stratosphere. Results presented here show the net changes due to the SLH implementation, and, unless stated otherwise, are based on the *FCnudged_slh* and *FCHIST_slh* comparison with their corresponding *FCnudged_noh* and *FCHIST_noh* sensitivities for the coarse (2°×2°) horizontal resolution.

Section 3 is organized as follows: first, the individual emissions of organic and inorganic halogens are quantified in Section 3.1, altogether with the mean SLH surface and tropospheric burden changes across the historical (1980-2020) period. Section 3.2 evaluates the modelled spatio-temporal distribution of organic VSLs and inorganic halogens (reactive and reservoir species) during present-day conditions, defined as the mean 2015-2020 period. Section 3.3 assesses the impacts of SLH chemistry on the main atmospheric components during present-day, including O$_3$ and OH destruction, as well as the main changes in the chemical loss and production channels induced by SLH. For the particular case of ozone, the distinctive




influence over the troposphere and stratosphere is evaluated separately. Finally, Section 3.4 presents a model inter-comparison
between the different CESM2-SLH resolutions and configurations. We highlight that in this work we do not attempt to provide
a complete/exhaustive comparison with all previously published results as it would imply to replicate a large number of
simulations with different configurations and periods of time (see Section 2.1). Instead, we provide a general description of
the global levels and distributions of chlorine, bromine and iodine in CESM2-SLH as well as their global implications on
atmospheric composition, and demonstrate that they are in line with previous CESM1 results. Nevertheless, we note that some
discrepancies remain due to CESM2 updates in some features and parameters that directly or indirectly affect SLH abundance
and chemistry. Therefore, we highlight that the current CESM2-SLH release has been optimized to obtain a good agreement
for global emissions, budgets and distribution of SLH in comparison with previous studies, which in turn result in consistent
halogen impacts on atmospheric composition (e.g. $O_3$, OH and $NO_2$).

### 3.1 Evolution of SLH Emissions and Burdens during the Historical Period

Figure 1 shows the global mean evolution of SLH emissions from 1980 to 2020, as well as the hemispheric
distribution of the total halogen sources discriminated for each individual halogen family. Note that the emission flux of natural
oceanic VSL halocarbons is read offline and remains constant throughout the entire period. This constitutes the only source of
natural bromo- and iodo-carbons (Ordóñez et al., 2012). However, additional $CH_2Cl_2$ and $C_2Cl_4$ emissions from anthropogenic
sources dominate the VSL chloro-carbon source (Hossaini et al., 2019; Claxton et al., 2020). In all cases, we highlight that all
emitted VSLs halocarbons remain unreactive until they photolyze and/or are photochemically degraded by OH, where they
release Cl, Br and I atoms that, due to their very high reactivity, react with other atmospheric components and partition among
the different species within the $Cl_y$, $Br_y$ and $I_y$ halogen families. Although VSL are not the primary source of halogens in the
MBL, their photodecomposition constitutes the critical first step in releasing inorganic chlorine and bromine to the atmosphere.
These initially released halogens can subsequently be amplified through non-stoichiometric heterogeneous recycling
processes, particularly via sea-salt aerosol dehalogenation (SSA-dehalogenation).

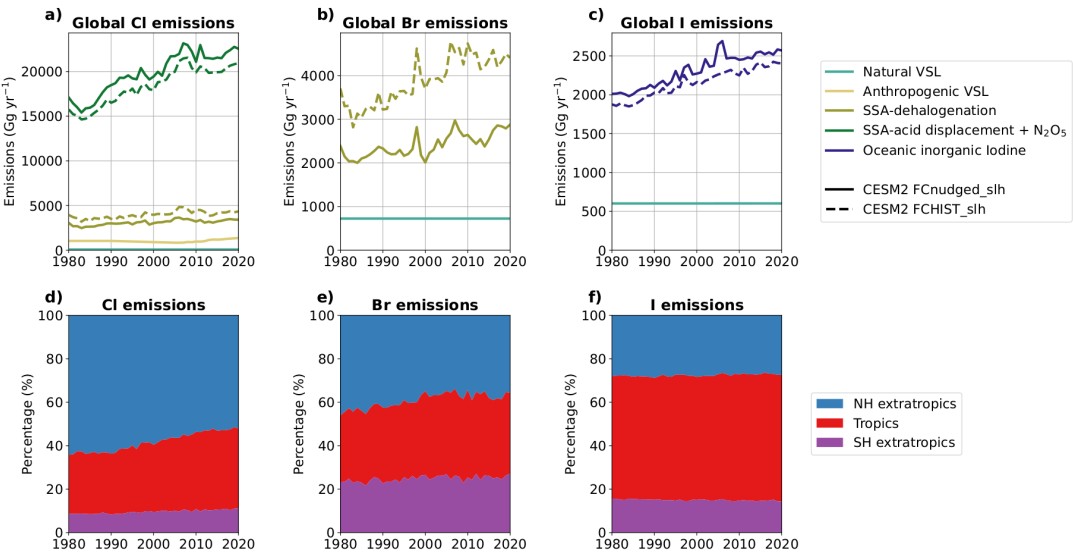

**Figure 1: Historical evolution of global annual mean SLH emissions during 1980-2020. Top-row shows the individual contribution from offline and online sources for a) chlorine, b) bromine and c) iodine, while bottom-row shows the percentage distribution within the Tropical (20°N – 20°S) as well as Northern Hemisphere (NH = 20°N - 90°N) and Southern Hemisphere (SH = 20°S - 90°S) extra-tropical regions. Online emissions for the *FCnudged_slh* (solid) and *FCHIST_slh* (dashed) configurations are distinguished, while the contribution from offline sources are equivalent for both compsets.**





In addition to offline VSL halocarbon emissions, inorganic halogen sources for chlorine, bromine and iodine are also

computed online (see Section 2.2.3), which in all cases represent the dominant source of halogens to the atmosphere (Figure

1 and Table 8). Here we highlight that despite the main drivers of these parameterized expressions are significantly different

for chlorine and bromine with respect to iodine, all of them show an increasing emission trend between 1980 and 2020 (Fig.

1). Indeed, (Prados-Roman et al., 2015b) suggested that this on-line ocean–atmosphere coupling represents a negative

geochemical feedback loop by which current ocean emissions of iodine ($HOI + I_2$) would be 2-3 times higher than in pre-

industrial times and act as a natural buffer for ozone pollution in the global marine environment, which was later demonstrated

by ice-core iodine observations (Cuevas et al., 2018). Results for both *FCnudged_slh* and *FCHIST_slh* compsets show a global

oceanic inorganic iodine flux increasing from 1.9-2.0 Tg I yr$^{-1}$ in 1980 to 2.4-2.5 Tg I yr$^{-1}$ in 2020 (Fig. 1c), with a dominant

contribution from the tropical regions (Fig. 1f). In absolute terms, this values are larger than previously reported results

obtained in CESM1, which ranged from 1.7-2.1 Tg I yr$^{-1}$ depending on the period of time and model configuration (Prados-

Roman et al., 2015b; Iglesias-Suarez et al., 2020; Barrera et al., 2023; Saiz-Lopez et al., 2023). The larger iodine flux in

CESM2-SLH is coherent with the higher CESM2 surface ozone abundances compared to CESM1 (Fig. 2d) (Emmons et al.,

2020), which is the main driver of the oceanic iodine flux enhancement (see Section 2.2.3). However, we note that current

CESM2-SLH oceanic iodine flux remain below other models that implemented iodine emissions based on CESM1 (~2.7 Tg

793        yr$^{-1}$ in GEOS-Chem and ~2.9 Tg yr$^{-1}$ in SOCOL; (Sherwen et al., 2016b; Karagodin-Doyennel et al., 2021)).

Regarding chlorine and bromine, the online recycling emissions are driven by, respectively, the acid-displacement

reaction occurring over sea-salt aerosols and the SSA-dehalogenation source (Fig. 1a,b and Table 8). Similar to the oceanic

inorganic iodine source, the modelled chlorine and bromine sources also increase from 1980 to 2020, although in this case the

main driver is the positive trend in anthropogenic $NO_x$ surface abundances (Fig. 2f). The net chlorine and bromine emissions

respectively increase from approximately 15-16 Tg Cl yr$^{-1}$ and 2.0-3.0 Tg Br yr$^{-1}$ in 1980-1985 to 21-23 Tg Cl yr$^{-1}$ and 2.8-

4.4 Tg Br yr$^{-1}$ during 2015-2020. Here, we note that chlorine sources present a larger inter-hemispheric difference due to the

dominant contribution of $NO_x$ emissions in the NH extratropical bands, which in turn increase the $N_2O_5$ and $HNO_3$ recycling

(Fig. 1d). Following the larger SSA abundance within the *FCHIST_slh* setup (Fig. S1), the SAA-debromination source for this

compset is larger than that for *FCnudged_slh*, although the corresponding bromine sinks for the former is also larger than for

the latter, resulting in equivalent bromine tropospheric burdens (see Table 8). Further details on the contributions of individual

emissions processes during present-day are provided in Section 3.2.

Figure 2 shows the inorganic halogen surface abundance during the 1980-2020 period for the *FCnudged_slh* and

*FCHIST_slh* configurations. In order to compare the current CESM2-SLH inorganic halogen abundances with previous

studies, Fig. 2 also shows the corresponding CESM1 *Serial_slh* and *Cyclical_slh* sensitivities corresponding to (Li et al., 2022)

and (Saiz-Lopez et al., 2023), respectively. Despite not strictly identical, these CESM1 configurations were selected as they

allow: *i)* to provide a quantitative visualization of the absolute abundances of each halogen family and their reactive to

inorganic halogen ratios ($XO_x/X_y$, with $XO_x = X + XO$ and $X = Cl, Br$ and $I$); and *ii)* to show a qualitative comparison of the

main atmospheric components driving the trends in the inorganic halogen budget within the different CESM versions. For

example, given that the dominant halogen sources in the troposphere are computed online depending on the background

concentration of $O_3$ and $NO_2$ (Figure 2d,f), the surface abundance of $Cl_y$, $Br_y$ and $I_y$ (Figure 2a-c) follows the temporal evolution

of the former species within each model version. Here it is worth noting that driven by the different trend in surface ozone

between CESM1 (cyan lines for *Serial_slh*) and CESM2 (blue for *FCHIST_slh* and pink for *FCnudged_slh*), surface $I_y$

abundance for CESM2-SLH during the 1980-2020 period presents a larger increment than in CESM1, particularly after year

2000 (Fig. 2c), while $I_y$ abundances in CESM1 remained approximately constant after year 2000 following the stable $O_3$ trend.

A similar behaviour is also observed for $Cl_y$ and $Br_y$, as their main sources depends mostly on the $HNO_3$ and $N_2O_5$ abundance

as well as on the extent of partitioning of halogen reservoirs to $ClONO_2$ and $BrONO_2$, all of which increase for larger $NO_x$



background conditions (Barrera et al., 2023). The total surface $Br_y$ abundance for *FCHIST_slh* is larger than for *FCnudged_slh*
but lie between previous CESM1 studies, while for the case of chlorine, both *FCnudged_slh* and *FCHIST_slh* configurations
are similar and show surface $Cl_y$ abundances that are lower than in (Li et al., 2022) and similar to (Saiz-Lopez et al., 2023).

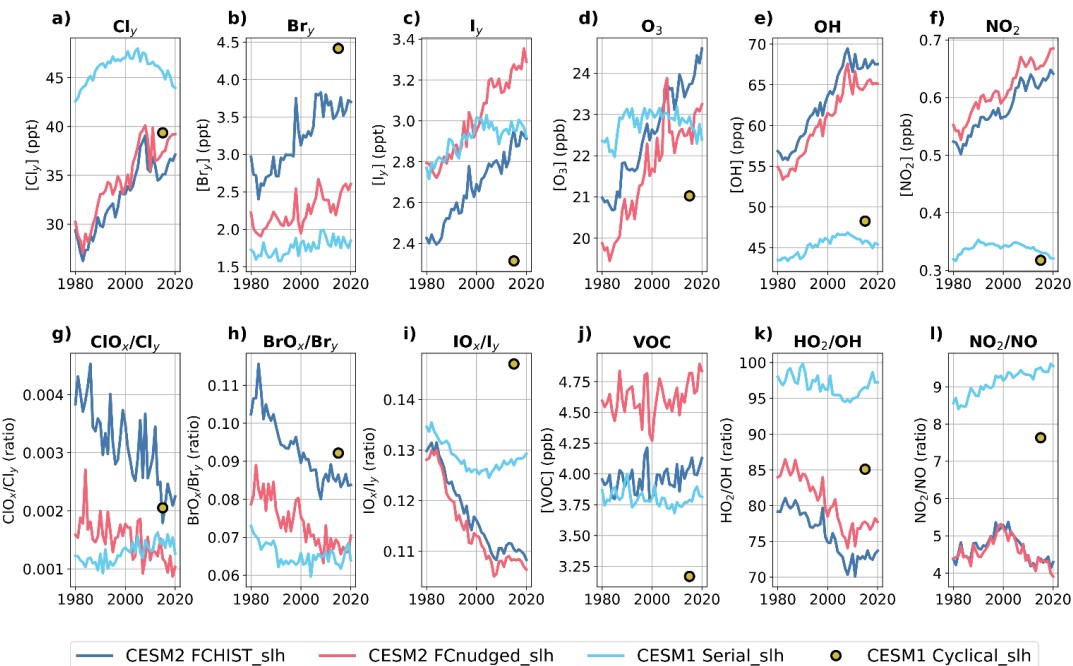


**Figure 2: Historical evolution of global annual mean surface abundance and partitioning of main atmospheric components during
1980-2020. Top-row shows the total inorganic halogen surface abundance for a) $Cl_y$, b) $Br_y$ and c) $I_y$, as well as the surface mixing
ratio for d) $O_3$, e) OH and f) $NO_2$. Panel j) shows the corresponding surface mixing ratio of major VOCs (sum of species: ISOP,
MTERP, $CH_3OH$, $C_2H_5OH$, $CH_2O$, $CH_3CHO$, $CH_3COOH$, $CH_3COCH_3$, HCOOH, $C_2H_2$, $C_2H_4$, $C_2H_6$, $C_3H_8$, $C_3H_6$, BIGALK,
BIGENE, MEK, TOLUENE, BENZENE, XYLENES). Bottom row shows the reactive (XOx) to reservoir ($X_y$) surface ratio (i.e.,
$XO_x / X_y$) for g) chlorine, h) bromine and i) iodine, as well as the k) $HO_2$ / OH and l) $NO_2$ / NO mean ratio at the model surface.
Results for the main *FCnudged_slh* (pink) and *FCHIST_slh* (blue) compsets obtained with the CESM2-SLH release are compared
with those obtained in previous CESM1 studies: *Serial_slh* (cyan line) setup from (Li et al., 2022) as well as *Cyclical_slh* (yellow
circle) configuration from (Saiz-Lopez et al., 2023).**


Figure 3 shows the tropospheric, stratospheric and total halogen burden across the 1980-2020 period resulting from

the CESM2-SLH *FCnudged_slh* and *FCHIST_slh* (solid lines) configurations along with the corresponding *FCnudged_noh*
and *FCHIST_noh* (dashed lines) sensitivities. Equivalent comparisons between the CESM1 *Serial_slh* and *Serial_noh*
configurations are also shown. Several distinctive features are of major relevance to support the final CESM2-SLH
configuration, as described below:

First, the temporal evolution of the stratospheric burden follows the expected trend with a maximum peak just before

and after year 2000 for chlorine and bromine, respectively, which is in line with the temporal variation of the dominant long-
lived ODSs that dominate the inorganic halogen burden in the stratosphere (WMO, 2018, 2022). In both cases, the increase in
stratospheric $Cl_y$ and $Br_y$ burden due to SLH are equivalent between CESM1 and CESM2 configurations. For stratospheric
iodine, all configurations show an equivalent stratospheric burden that remains constant with time (see Section 3.2.2). Reaching
an equivalent stratospheric halogen enhancement for all configurations provides confidence that the modelled influence of
SLH in the lower stratosphere within CESM2-SLH are in line with previous studies (see Section 3.3.4).



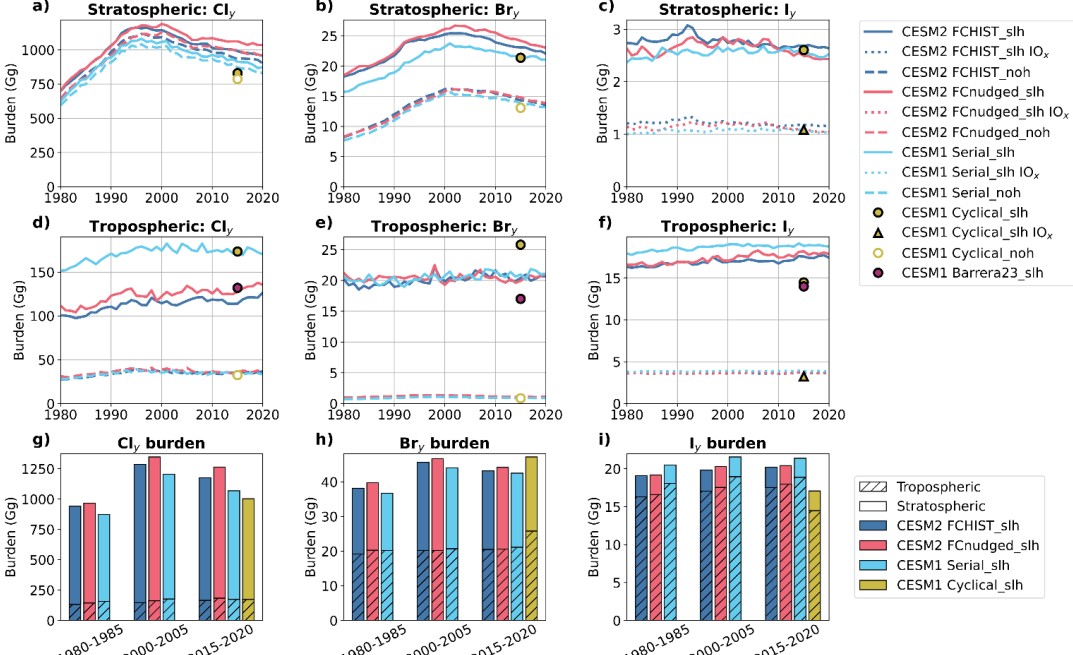


**Figure 3: Historical evolution of global mean inorganic halogen burden during 1980-2020.** Top-row shows the stratospheric burden for a) $Cl_y$, b) $Br_y$ and c) $I_y$, while middle-row shows the corresponding tropospheric burden for d) $Cl_y$, e) $Br_y$ and f) $I_y$ for different model configurations including (*compset_slh*, solid lines) and neglecting (*compset_noh*, dashed lines) the contribution of SLH sources and chemistry. Note that as *compset_non* configurations do not consider iodine chemistry, panels c) and f) shows the reactive iodine fraction ($IO_x = I + IO$) for the *compset_slh* using dotted lines to avoid confusion with $I_y$. Results for the main *FCnudged_slh* (pink) and *FCHIST_slh* (blue) compsets obtained with the CESM2-SLH release are compared with those obtained in previous CESM1 studies: *Serial_slh* (cyan line) setup from (Li et al., 2022) as well as *Cyclical_slh* configurations from (Saiz-Lopez et al., 2023) (yellow circle) and (Barrera et al., 2023) (red circle). $IO_x$ results are shown with yellow triangles. Bottom-row shows the mean total inorganic halogen burden for the 1980-1985, 2000-2005 and 2015-2020 periods using stacked columns distinguishing the tropospheric (striped) and stratospheric (empty) contributions for g) chlorine, h) bromine and i) iodine.


Second, although the global mean tropospheric halogen burden for *FCnudged_slh* and *FCHIST_slh* are not identical,
the spread between the different CESM2-SLH compsets is smaller than that obtained with the different CESM1 configurations
used in previous studies. Indeed, for the case of bromine, the tropospheric $Br_y$ burden for both *FCnudged_slh* and *FCHIST_slh*
compsets are equivalent to those obtained for the CESM1 *Serial_slh* setup, although they lie in between the CESM1
*Cyclical_slh* simulations from (Saiz-Lopez et al., 2023) and (Barrera et al., 2023). Here we note that the tropospheric $Br_y$
burden of the former is larger as that setup considered the original sea-salt aerosol logical mask typically applied to the SSA-
dehalogenation (Fernandez et al., 2014, 2021), while the later (*Barrera23_slh* in Fig. 3) imposed a latitudinal dependent sea-
salt mask that resulted in significantly lower bromine recycling (Barrera et al., 2023), highlighting the large sensitivity of the
SSA-dehalogenation source to the modelled changes in SSA abundance and distribution (see Fig. S1). Most notably, both
CESM2-SLH *FCnudged_slh* and *FCHIST_slh* configurations now result in a global tropospheric $Br_y$ burden that lies in
between those obtained in previous CESM1 studies (Fig. 3e). In addition, the tropospheric $Cl_y$ burden for any CESM2-SLH
(2°×2°) setup is lower than in any of the CESM1 configurations, which ensures that in any case the impact of chlorine chemistry
on tropospheric composition (e.g., the degradation of VOCs and $CH_4$ by Cl atoms) is, in any case, in the lower edge of
previously published studies (Li et al., 2022; Bossolasco et al., 2025).
Third, it is worth highlighting that despite the tropospheric $I_y$ trend for all timeline simulations follows the oceanic
iodine flux increase due to the enhanced ozone levels between 1980 and 2020 (Fig. 2f), the overall stratospheric burden remains
constant with time (Fig. 3c). This implies that most of the inorganic iodine enhancement occurring in the MBL and close to



the surface is washed out in the free troposphere before reaching the tropopause. The larger tropospheric $I_y$ burden between
current *FCHIST_slh* and *FCnudged_slh* compsets (17.2-17.7 Tg I) and the recent estimations from (Saiz-Lopez et al., 2023)
and (Barrera et al., 2023) using CESM1 (14-14.5 Tg I) for present-day conditions are explained by the significant changes in
reactive ($IO_x$) vs. reservoir iodine partitioning ($IO_x / I_y$) between CESM1 and CESM2. Despite the differences in $I_y$ burdens,
the tropospheric $IO_x$ burden for all CESM1 and CESM2 configurations are equivalent (see dotted lines in Fig. 2f), and therefore
the influence of iodine on atmospheric composition shown in Section 3.3.3 remains consistent with previous CESM1 studies.

**3.2 SLH Abundance During Present-Day**
Table 8 summarizes the global halogen sources, LBCs, tropospheric abundances and sinks during present-day for the
*FCnudged_slh* and *FCHIST_slh* (2º×2º) compsets. These global-mean reference values should be taken as guideline for any
other CESM2-SLH configuration and/or resolution considered. Equivalent results highlighting the main similarities and
differences with respect to the remaining CESM2-SLH compsets are shown in Section 3.4.
The global annual emission flux for VSL halogens in both CESM2-SLH compsets during present-day is 65.3 Gg Cl
$yr^{-1}$, 721.3 Gg Br $yr^{-1}$, and 599.7 Gg I $yr^{-1}$. Here, we recall that due to changes in transport and OH abundance within the
MBL between CESM1 and CESM2 (see Section 2.2.1), we increased the flux strength of the original VSL inventory (Ordóñez
et al., 2012) by 15% for all VSL bromo- and chloro-carbon but retained the magnitudes of iodo-carbon species, including
mixed species ($CH_2ICl$, $CH_2IBr$). Thus, VSL iodine mean global emissions are the same between CESM1 and CESM2, while
the Cl and Br emissions are ~14% and 6% larger in CESM2 compared to CESM1. The higher (lower) percentage change for
bromine (chlorine) arises due to the minor $CH_2IBr$ (predominant $CH_2ICl$) relative contribution of iodo-carbons to the total
VSL bromocarbon (chlorocarbon) flux (Ordóñez et al., 2012). We highlight that the contribution of anthropogenic VSLs
(1265.5 Gg Cl $yr^{-1}$) has also been increased by 15% and results in surface fluxes up to 20 times larger than the natural oceanic
source from VSL chlorocarbons (65.3 Gg Cl $yr^{-1}$), in agreement with previous estimates (Villamayor et al., 2023). Here, it
should be noted that the additional contribution from other anthropogenic VSL chlorocarbons ($CHCl_3$ and $C_2H_4Cl_2$) are also
included at the model surface as LBCs instead of considering offline emissions, reaching a total of ~48 pptv at the model
surface.
The largest variability among all online sources arises from the uptake and release of bromine and chlorine from sea-
salt aerosol. In absolute terms, the global annual mean SSA-dehalogenation source for the *FCHIST_slh* compset is 877.3 Gg
Cl $yr^{-1}$ (26%) and 1616.7 Gg Br $yr^{-1}$ (58%) larger than for the *FCnudged_slh* setup (see Table 8). This is explained by the
large non-linear response of the heterogeneous recycling process implemented in Eq. (3) to the large variability in the
parameterized fields of $SAD_{SSA}$ (Fig. S1), as well as to the change in partitioning between inorganic halogen reservoirs (Fig.
7). Similarly, the chlorine source from the acid-displacement for the *FCnudged_slh* compsets is larger by ~1700 Gg Cl $yr^{-1}$,
which in relative terms represents only ~8% variability but in absolute terms surpasses the total contribution from VSL chlorine
sources. Despite the variability in modelled online sources, the estimated global fluxes are in line with previous estimations
obtained with different models. For example, our present-day global chlorine source is equivalent to CESM1 (21.5 Gg Cl $yr^{-1}$)
and around 4 times smaller than an equivalent implementation of acid displacement dehalogenation in the TOMCAT model
(~90 Tg Cl $yr^{-1}$; (Hossaini et al., 2016) and between 1.5 and 3 times lower than the range of values estimated by (Graedel and
Keene, 1995) (~37−73 Tg Cl $yr^{-1}$). For the case of bromine, which presents the largest variability between CESM2-SLH
compsets, our global annual flux remains lower than the recent estimates obtained with GEOS-Chem when the reactivity of
bromine with tropospheric aerosols is considered (3.5−6.4 Tg $yr^{-1}$) (Chen et al., 2018; Zhu et al., 2018), highlighting that in
any case the influence of SLH chlorine and bromine in the troposphere represents a lower limit compared to other models.
Regardless of the large variability on the online computation of inorganic halogen sources, we highlight that CESM2-
SLH considers a species-independent removal of inorganic halogens from the atmosphere, and consequently, the larger the



emission flux for any specific configuration, the larger the corresponding global sink. Indeed, inspection of Table 8 shows that absolute and percentage changes in both dry and wet deposition between *FCnudged_slh* and *FCHIST_slh* are of equivalent magnitude to their corresponding sources. Here, it should be noted that the surface dry deposition occurring only at the model surface accounts for almost half of the total halogen sink, representing ~45%, ~40% and ~50% for chlorine, bromine and iodine, respectively. For the case of wet deposition tendencies, the NEU scheme represents the largest individual sink for bromine and chlorine, while for iodine, the NEU scheme accounts only for approximately half of the total wet deposition, with a substantial contribution from the FRA removal of HOI in liquid clouds. In contrast, the FRA only represents a small fraction (< 5-7%) of the total bromine wet deposition compared to the NEU scheme, as only ice_uptake of $BrONO_2$ is considered to occur through FRA (see Section 2.4). Finally, it is worth noting that despite dry and wet deposition are computed for all individual species compiled in Table 4, the net sink of each halogen family is typically dominated by a single or a couple of species that dominates the halogen partitioning in different layers of the troposphere (e.g., HCl for chlorine, HOBr and HBr for bromine and HOI for iodine, see (Fernandez et al., 2014; Saiz-Lopez et al., 2014)).

Table 8 also quantifies the mean surface abundance and global tropospheric burdens for both organic VSL and inorganic SLH during present-day. Despite minor differences between the two compsets that in general remain below ~10%, the surface abundance and tropospheric burden for all halogen species are slightly larger for the *FCnudged_slh* configuration. The exception is surface $Br_y$, which is ~44% larger in *FCHIST_slh* due to its larger SSA-dehalogenation source. However, note that the tropospheric $Br_y$ burden between both CESM2 configurations is almost identical (20.1 Gg Br for *FCnudged_slh* vs. 20.0 Gg Br for *FCHIST_slh*). Surface chlorine abundance is dominated by anthropogenic VSL reaching 135-139 pptv that largely surpass the contribution from oceanic VSL (less than 1 pptv), while surface inorganic $Cl_y$ reaches approximately 40 pptv. The corresponding values for brominated compounds reach 7.8-8.4 pptv for VSL bromine and 2.5-3.6 pptv $Br_y$, while for the case of iodine global mean values are 1.1-1.2 pptv and 2.9-3.3 pptv, respectively. Here we highlight that for the particular case of inorganic chlorine and bromine, current results for *FCHIST_slh* and *FCnudged_slh* configurations in CESM2-SLH have been adjusted to achieve global annual surface values and tropospheric burdens that lie in between previous CESM1 studies (Li et al., 2022; Barrera et al., 2023; Saiz-Lopez et al., 2023). Regarding the partitioning between reactive and reservoir halogen species for the different compsets, our model results show that surface and tropospheric $XO_x$ / $X_y$ ratio for chlorine, bromine and iodine ranges between 0.2-1.5%, 6-12% and 10-20%, respectively, with general larger values for *FCHIST_slh* compared to *FCnudged_slh*.

In contrast to bromine and chlorine, surface $I_y$ abundance in CESM2-SLH is ~25-40% higher compared to CESM1 (2.3 pptv $I_y$), which is mainly due to the different $IO_x/I_y$ and $OH/HO_2$ ratios between CESM versions (see Fig. 2i,k): the larger OH abundance and lower $HO_2/OH$ ratio in CESM2-SLH compared to CESM1 increases the contribution of HOI to the total $I_y$ loading, and therefore, there is a major shift on the partitioning from reactive to reservoir species for iodine (see Table 8). Therefore, the reactive $IO_x$ abundance between the different compsets remains similar compared to (Saiz-Lopez et al., 2023) and (Li et al., 2022) both at the surface as well as integrated in the whole troposphere. Indeed, Table 8 and Fig. 3 show that despite the large surface $I_y$ differences, the $IO_x$ burden change between CESM2 (3.5-3.6 Gg I) and CESM1 (3.2 Gg I) remains close to ~10%.

Finally, we highlight the different contributions of inorganic chlorine, bromine and iodine to the total (tropospheric + stratospheric) burden when SLH are considered (Fig. 3 bottom-row). We found that for present-day conditions, the contribution of tropospheric $Cl_y$, $Br_y$ and $I_y$ to the total halogen loading represents ~14%, ~47% and ~87%, respectively. This increasing contribution of tropospheric $X_y$ content going from chlorine to bromine to iodine is in line with the enhanced efficiency/reactivity of each halogen species in reacting with tropospheric ozone ($X + O_3 \rightarrow XO + O_2$, see R2), and result in tropospheric iodine to be the dominant species affecting the tropospheric oxidative capacity and ozone abundance (see Section 3.3.2). Given the turnover in stratospheric halogen loading from anthropogenic ODSs around year 2000, we note that the percentage contributions change for the different time periods.




**Table 8: Quantitative values for global annual SLH burdens during present-day conditions**

| Species | Simulation | Chlorine | | Bromine | | Iodine | |
|---|---|---|---|---|---|---|---|
| | | Fcnudged_slh | FCHIST_slh | Fcnudged_slh | FCHIST_slh | Fcnudged_slh | FCHIST_slh |
| **Offline Sources** | | | | | | | |
| Natural VSL | Surface (Gg yr⁻¹) | 65.3 | 65.3 | 721.3 | 721.3 | 599.7 | 599.7 |
| Anthro VSL | Surface (Gg yr⁻¹) | 1265.5 | 1265.5 | - | - | - | - |
| Anthro VSL (LBCs) | Surface (pptv) | 48.1 | 48.1 | - | - | - | - |
| **Online Sources** | | | | | | | |
| SSA-dehalogenation | Tropo (Gg yr⁻¹) | 3365.3 | 4242.6 | 2764.3 | 4381.1 | - | - |
| HNO₃ | Tropo (Gg yr⁻¹) | 20844.4 | 19257.8 | - | - | - | - |
| N₂O₅ | Tropo (Gg yr⁻¹) | 1235.0 | 1123.9 | - | - | - | - |
| Oceanic Iodine | Surface (Gg yr⁻¹) | - | - | - | - | 2540.3 | 2387.6 |
| **Abundance** | | | | | | | |
| Long-Lived Halogens | Surface | 3139.9 | 3139.9 | 14.2 | 14.2 | - | - |
| | Tropo | 15970.3 | 15987.3 | 161.1 | 161.5 | - | - |
| **Short-Lived Halogens** | | | | | | | |
| Natural VSL | Surface (pptv) | 0.8 | 0.7 | 8.4 | 7.8 | 1.2 | 1.1 |
| | Tropo (Gg) | 2.5 | 2.4 | 55.1 | 53.8 | 4.8 | 4.7 |
| Anthro VSL | Surface (pptv) | 138.7 | 135.1 | - | - | - | - |
| | Tropo (Gg) | 574.2 | 571.8 | - | - | - | - |
| Inorganic Halogens (Xᵧ) | Surface (pptv) | 41.0 | 38.7 | 2.5 | 3.6 | 3.3 | 2.9 |
| | Tropo (Gg) | 177.4 | 159.4 | 20.1 | 20.0 | 17.7 | 17.2 |
| Reactive Halogens (XOₓ) | Surface (pptv) | 0.1 | 0.1 | 0.2 | 0.3 | 0.4 | 0.3 |
| | Tropo (Gg) | 2.1 | 2.9 | 2.1 | 2.4 | 3.5 | 3.6 |
| **Sinks** | | | | | | | |
| Dry deposition | Surface (Gg yr⁻¹) | 14102.1 | 13066.9 | 1484.76 | 2168.33 | 1680.93 | 1483.13 |
| Wet deposition (NEU) | Tropo (Gg yr⁻¹) | 16057.9 | 16481.4 | 1952.3 | 2808.8 | 836.711 | 941.331 |
| Wet deposition (FRA) | Surface (Gg yr⁻¹) | - | - | 106.148 | 188.316 | 634.221 | 574.448 |




**3.2.1 Latitudinal and Vertical Distribution of SLH**

Figure 4 shows the zonal average latitudinal-height distribution of total organic long-lived (top row) and short-lived

(bottom row) halogens for the *FCnudged_slh* configuration during present-day. Here it is evident that while long-lived
halogenated ODSs remain unreactive throughout the troposphere and are converted to inorganic $Cl_y$ and $Br_y$ only after crossing
the tropopause, VSL halocarbons photolyze at much lower heights in the atmosphere. Consequently, iodine VSL is nearly
completely decomposed in the lower troposphere (Fig. 4f), whereas only a small fraction of the emitted VSL bromine and
chlorine SGs is transported unaltered to the stratosphere (Fig. 4d,e). In contrast to the naturally emitted bromine and iodine
VSLs, the modelled VSL chlorine abundances show a remarkable inter-hemispheric difference with larger values over the NH,
due to its dominant anthropogenic sources (Fig. 4d).

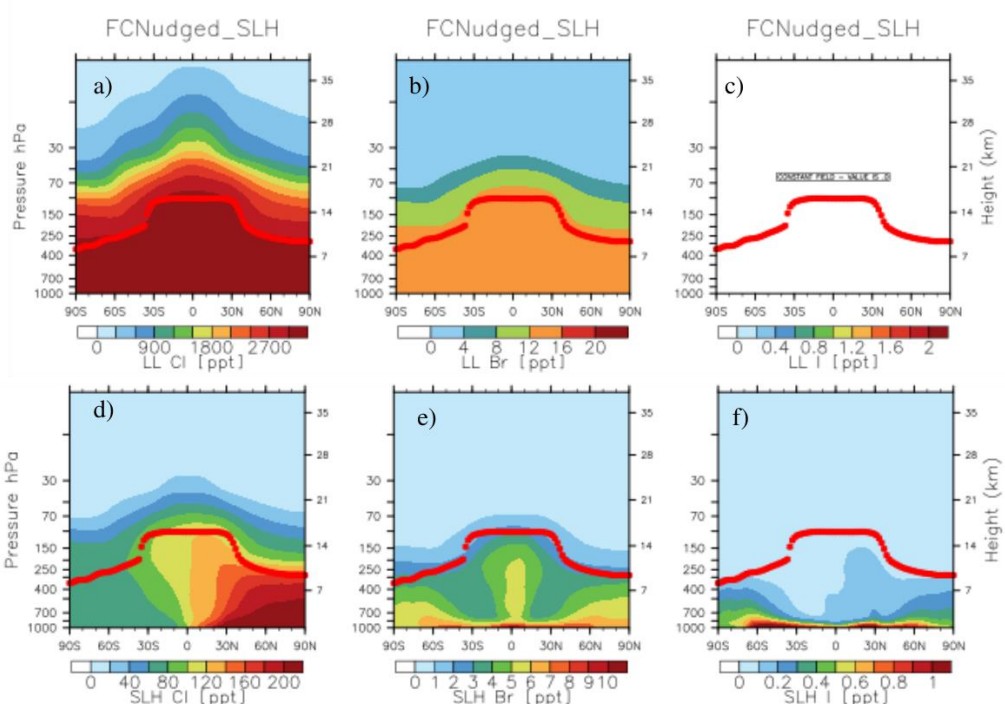


**Figure 4: Zonal Average distribution of the organic (carbon bonded) halogen fraction during present-day (2015-2020) for the**
***FCnudged_slh* compset. Top row shows the long-lived (LL) distribution for a) chlorine ($LL_{Cl}$) and b) bromine ($LL_{Br}$). Note that**
**CESM2-SLH does not include long-lived iodine species (empty panel c). Bottom row shows the very short-lived (VSL) distribution**
**for d) chlorine ($VSL_{Cl}$), e) bromine ($VSL_{Br}$) and f) iodine ($VSL_I$). The red-dotted line shows the mean model tropopause.**

In addition, Figure 5 shows the corresponding zonal average distribution of inorganic halogens ($Cl_y$, $Br_y$ and $I_y$) from

the model surface up to the middle stratosphere for the *FCnudged_slh* (top row) and *FCnudged_noh* (middle-row), as well as
the difference between both configurations (bottom row). Note that the *FCnudged_slh* configuration does not consider any
long-lived iodine source (Fig. 4c) while the *FCnudged_noh* configuration does not include iodine chemistry at all (Fig. 5f).
Notably, not only the total $X_y$ abundance at the model top, but also the conversion from organic SGs to inorganic halogen
(PGs) shown in Figure 5 occurs at lower heights for the *FCnudged_slh* (top panels) than for the *FCnudged_noh* (bottom
panels), following the different ODS and VSL photodecomposition shown in Fig. 4. Given the typically longer photochemical
lifetimes of both long-lived chlorinated compounds compared to brominated halons (see for example Table A-1 in the Annex
of (WMO, 2022) and elsewhere), the conversion from unreactive organic to reactive inorganic species occurs faster for bromine



than for chlorine, which results in a steeper gradient for the former in the lower stratosphere, and an almost complete conversion
to inorganic bromine in the upper stratosphere (see also Fig. 6).

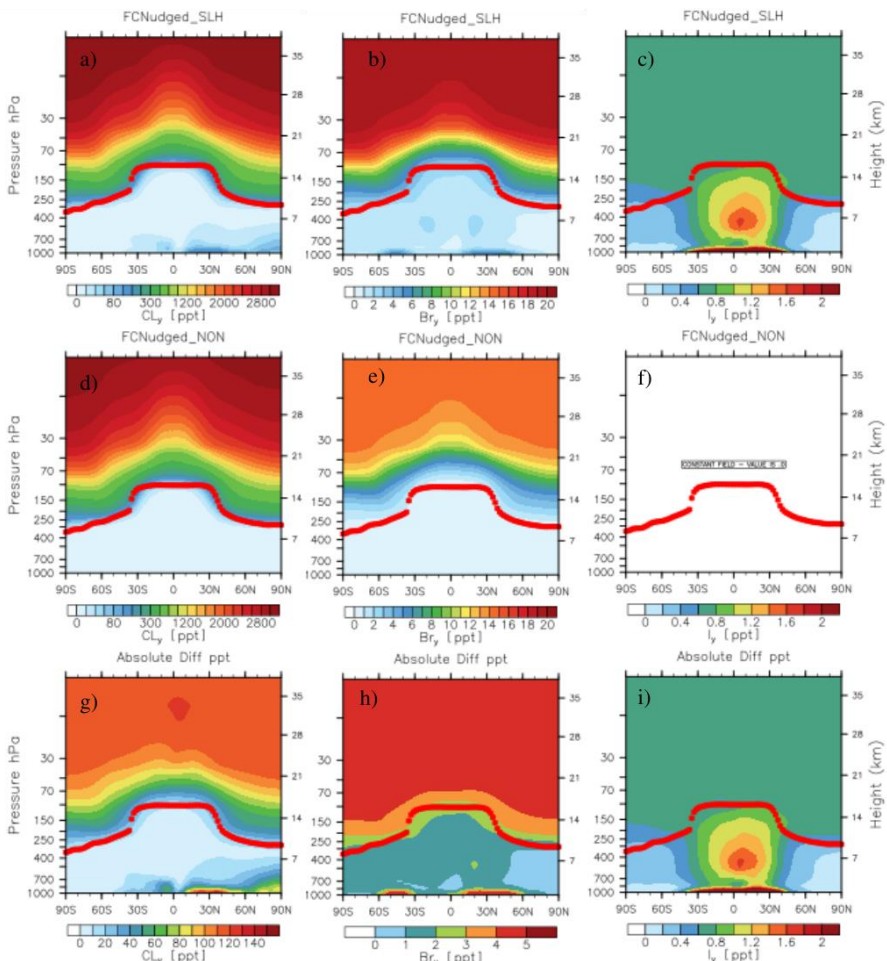


**Figure 5: Zonal Average distribution of total inorganic halogen ($X_y$) during present-day (2015-2020) for different model**
**configurations including (top-row; *FCnudged_slh* ) and neglecting (middle-row; *FCnudged_noh*) the contribution of SLH sources**
**and chemistry. The bottom-row show the absolute difference between both compsets (*FCnudged_slh − FCnudged_noh*). Left column**
**(a, d, g) show the distribution for chlorine ($Cl_y$), while the middle (b, e, h) and right (c, f, i) columns show the distribution of bromine**
**($Br_y$) and iodine ($I_y$), respectively. The red-dotted line shows the mean model tropopause.**

The change in inorganic halogen abundance $X_y$ due to SLH (Figure 5, bottom panels) highlights the different
behaviour for each individual halogen family in different regions of the atmosphere. For example, within the troposphere, both
$Cl_y$ and $Br_y$ show a pronounced enhancement in the MBL and close to the surface mostly in the NH mid-latitude regions that
is rapidly reduced in the free-troposphere, which is driven by the efficient SSA-dehalogenation source and acid displacement
that presents a sharp vertical profile (see Fig. 7). For the case of iodine, the largest abundances at the surface are driven by the
strong $I_2$/HOI emission occurring in the tropical regions (Fig. 1f). In this region, up to 1 pptv of $I_y$ is transported to the tropical
free troposphere where, due the local changes in ozone abundance and temperature, a pronounced partitioning shift lead to the
formation of the tropical rings of atomic halogens (Saiz-Lopez and Fernandez, 2016). A fraction of this tropospheric inorganic
iodine is washed out below the tropopause, and the remaining fraction is injected to the stratosphere as inorganic PGs





throughout tropical injection with an almost negligible contribution from VSL SGs (see Section 3.2.2). In contrast, Fig. 5g shows a gradual enhancement in $Cl_y$ occurs in the middle stratosphere, which arises due to the longer photochemical lifetime of chlorinated VSLs which are mostly transported unaltered to the lower stratosphere (Fig. 4a). For bromine, Fig. 4e and Fig. 5h highlights that both organic VSL SGs and inorganic $Br_y$ PGs contribute to the total stratospheric bromine loading due to SLH.

**Figure 6: Global mean vertical profile of organic and inorganic halogens during present-day (2015-2020) for a) chlorine, b) bromine and c) iodine. Each coloured line shows individual contributions from organic long-lived (LLH$_X$, yellow) and very short-lived (VSL$_X$, blue) halogens, as well as for the total inorganic halogen fraction (X$_y$, red) for the *FCnudged_slh* (empty symbols) and *FCHIST_slh* (filled symbols) compsets. The total sum of all three contributions for each individual halogen species (X = Cl, Br, I) is shown in black. The solid and dashed horizontal line shows the mean height ± standard deviation of the global tropopause.**

Figure 6 summarizes the global mean (90ºN – 90ºS) vertical profiles of the sum of all organic and inorganic halogen species from the surface to the middle stratosphere for the *FCnudged_slh* and *FCHIST_slh* configurations. The figure distinguishes the contributions from long-lived chlorine and bromine species, as well as the total halogen fraction transported as VSL SGs and inorganic PGs, consistent with the separate depictions presented in Figures 4 and 5. The conversion of unreactive organic SGs to reactive PGs is found to be very similar between the two model compsets. However, minor differences, primarily within the MBL and the Upper Troposphere/Lower Stratosphere (UTLS), appear due to the differing meteorological representations (free-running vs. nudged dynamics) between the *FCHIST_slh* and *FCnudged_slh*. Three main differences characterize the vertical distribution between unreactive organic and reactive inorganic species for each halogen family. First, the tropospheric chlorine abundance is dominated by long-lived species, with minor contributions from VSL chlorine and $Cl_y$. Due to the long lifetimes of CFCs and $CH_3Cl$, the crossing point between $LLH_{Cl}$ and $Cl_y$ representing the photochemical conversion from the organic to the inorganic fractions occurs in the middle stratosphere (~40 hPa). Second, the VSL bromine contribution is comparable (although smaller) to the contribution from long-lived halons and $CH_3Br$, and therefore the initial enhancement in $Br_y$ abundance, as well as the crossing point with $LLH_{Br}$, occurs at lower heights in the stratosphere (~75 hPa). Third, given that there are no long-lived iodine species and the short photochemical lifetimes of $VSL_I$ (particularly $CH_3I$), both the tropospheric and stratospheric abundance is dominated by inorganic $I_y$. This is in line with the tropospheric and stratospheric mean burdens shown in the stacked bars of Fig. 3g-i.





**Figure 7: Global mean vertical profile of inorganic halogen release from different types of sources (left column) and the corresponding abundance for the different species conforming each halogen family (right column) during present-day (2015-2020). Individual panel show results for (a, b) chlorine, (c, d) bromine and (e, f) iodine for the *FCnudged_slh* configuration. The halogen atom release arising from the photochemical (Oh + hv) degradation of organic very short-lived (VSL, blue) and long-lived (LLH, yellow) halogens, including the independent contribution from CH₃Cl and CH₃Br (orange) are distinguished on the left panels. The vertically-resolved inorganic halogen source arising from SSA-dehalogenation and acid-displacement processes is shown in green, while the surface oceanic iodine emission is indicated by a pink triangle. Right panels show the contribution of each individual halogen species to the total inorganic halogen loading ($X_y$, black), where red and blue lines highlight the abundance of the dominant reactive halogen species (X and XO) for each family (X = Cl, Br, I). The dashed lines shows the global mean height of the global boundary layer and the tropopause.**

Given that the atmospheric impact of SLH over different atmospheric components depend on the overall abundance of reactive halogens, whose initial step is the release of a halogen atom Cl, Br and I through reactions R1a,b, left panels in Figure 7 shows the halogen atom release from the different organic and inorganic sources as a function of height. Here note





that only emissions or processes that lead to net inorganic halogen production are considered. In addition, right panels in Figure 7 show the chemical partitioning between all halogen species constituting each inorganic halogen family ($X_y$) which interconvert between one and the other following gas-phase and heterogeneous phase reactions involving other atmospheric components such as $O_3$, OH and $NO_2$ (see summarized R1-R13 scheme in Section 1 and Tables S1-S3 in the Supplementary Material).

As summarized in Table 8, SSA-dehalogenation is the dominant source of bromine and the second largest source for chlorine. However, Figure 7c highlights that these sources are primarily confined to the MBL. For chlorine, the contribution of the acid-displacement HCl release dominates, particularly in the NH mid-latitudes and coastal locations. Above the boundary layer, the photochemical degradation of both $VSL_{Cl}$ and $VSL_{Br}$ dominates the release of Br and Cl atoms throughout the free troposphere, although a particular difference becomes evident: the biogenic ocean flux is the sole source of bromine, while for the case of chlorine the contribution from anthropogenic $VSL_{Cl}$ sources are at least two orders of magnitude larger than the natural oceanic source. In addition, the release of Cl atoms from long-lived $CH_3Cl$ accounts for an equivalent source as that arising from anthropogenic $VSL_{Cl}$ even below the tropopause (Fig. 7a). In contrast, bromine release from long-lived $CH_3Br$ only surpass that arising from $VSL_{Br}$ in the lower stratosphere (Fig. 7c). For iodine, ocean $HOI/I_2$ release dominates the surface emissions, while the photochemical degradation of $CH_3I$ alone controls the organic to inorganic iodine conversion throughout the free troposphere.

Figure 7b shows that the dominant inorganic chlorine species throughout the troposphere is HCl, which represents between 63% and 85% of the total $Cl_y$ partitioning. The Cl atom and ClO partitioning are generally 4 and 1-2 orders of magnitude smaller than the global mean $Cl_y$, respectively, although due to rapid heterogeneous recycling their contributions on the regional scale can increase, particularly in coastal locations (see Fig. 10). For bromine (Fig. 7d), the dominant species in the lower troposphere is HBr both during day and night (Fernandez et al., 2014). However, as it is transported to the upper troposphere and reacts heterogeneously on tropospheric ice-crystals (see Table 2), HBr is converted at first to HOBr and finally to the dominant $BrONO_2$ fraction in the lower stratosphere. The vertical profile of iodine partitioning in Fig. 7f shows that HOI is the dominant iodine species throughout the troposphere and therefore its washout controls the transport and abundance of inorganic iodine from their dominant surface sources to the stratosphere. It should be noted that the chemical partitioning from reservoir to reactive halogen species is the largest for iodine, followed by bromine and last by chlorine (i.e., $IO_x/I_y >$ $BrO_x/Br_y > ClO_x/Cl_y$). This is associated with the much faster photochemistry of iodine compared to bromine and, to a larger extent, chlorine, which in turn controls the efficiency of each halogen family in altering the composition and oxidative capacity of the troposphere (see Section 3.3.2).

### 3.2.2 Validation of SLH Abundance with Observations

This section presents a general validation of the model performance by comparing CESM2-SLH results with observations. In doing so, we highlight that all SLH developments were implemented over the well validated CESM2 chemistry-climate model that has been widely used for climate projections and air-quality studies (Danabasoglu et al., 2020; Emmons et al., 2020; Jo et al., 2023; Tilmes et al., 2023), and consequently we do not evaluate all aspects of the model's chemistry and dynamics. For this exercise, we perform standardized evaluations against mean SLH observations and distributions compiled in international assessments.

Several studies highlight the importance of properly distinguishing between the contribution of Source Gas Injection (SGI) and Product Gas Injection (PGI) to the stratosphere when representing the contribution of SLH to stratospheric ozone depletion (Fernandez et al., 2014, 2021; WMO, 2018, 2022). This is mainly because the ozone destruction efficiency of SLH in the lowermost stratosphere depends on the net fraction of the emitted VSLs that has already been converted to the reactive inorganic form (Fernandez et al., 2021). In order to quantify the total contribution of SLH to the net SGI + PGI of each halogen



species during stratospheric injection, we computed the tropical mean (20ºN – 20ºS) vertical profiles of all VSL source gases
(VSL$_X$) as well as the change in inorganic halogen product gases ($\Delta X_y$) from the Earth's surface to the stratosphere (Fig. 8).
Here, the $\Delta Cl_y$, $\Delta Br_y$ and $\Delta I_y$ profiles are computed as the difference between the *FCnudged_slh* − *FCnudged_noh* as well as
for the *FCHIST_slh* − *FCHIST_noh* configurations. Therefore, they only account for the additional contribution of SLH to
the total inorganic halogen loading at any given height, without considering the small (but not negligible) contribution from
long-lived chlorine and bromine photodecomposition close to the tropopause (see Figs. 4d,e and 7a,c). In addition, Fig. 8
shows the sum of SG + PG for each halogen family (i.e., VSL$_{Cl}$+$\Delta Cl_y$, VSL$_{Br}$+$\Delta Br_y$ and VSL$_I$+$\Delta I_y$) accounting for the total
additional chlorine, bromine and iodine due to SLH transported from the model surface to the stratosphere. Given that for
bromine the contribution of SGs and PGs are of similar magnitude, we first provide a brief description of the individual VSL$_{Br}$
and $\Delta Br_y$ vertical profiles, and then describe the more extreme contributions for chlorine and iodine below.



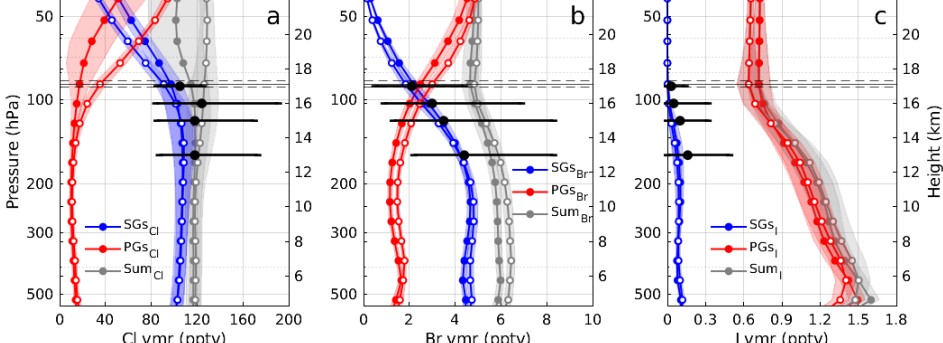


**Figure 8: Tropical vertical profile of organic very short-lived (VSL$_X$) source gases (SGs) and change in inorganic halogens ($\Delta X_y$)**
**product gases (PGs) during present-day (2015-2020) for a) chlorine, b) bromine and c) iodine. The VSL$_{Cl}$, VSL$_{Br}$ and VSL$_I$ vertical**
**profile (blue lines) correspond to the *FCnudged_slh* (empty symbols) and *FCHIST_slh* (filled symbols) compsets, while $\Delta X_{Cl}$, $\Delta X_{Br}$,**
**and $\Delta X_I$ (red lines) show the corresponding difference with respect to the configuration neglecting SLH sources and chemistry (i.e.,**
***compset_slh − compset_noh*). The total additional halogen abundance due to SLH (SGs + PGs)$_X$ is shown in grey (Sum$_x$). The tropical**
**(20°N – 20°S) mean ± range (computed as the standard deviation for the 2015-2020 period) is shown by the solid-symbol lines and**
**coloured shading, respectively. The observed mean ± range mole fractions within the tropical tropopause layer compiled in Table 1-**
**5 of (WMO, 2022) are shown by black symbols and solid-thick horizontal lines at 13, 15, 16 and 17 km, respectively. The solid and**
**dashed horizontal line shows the mean height ± standard deviation of the tropical tropopause.**

Based on a mean modelled tropical tropopause at approximately 85 hPa (17 km), we found that the modelled CESM2-
SLH contribution of SGI and PGI to the total bromine injection reaches 1.8-2.2 pptv + 2.5-3.0 = 4.3-5.2 pptv during present-
day, where the modelled range indicate the variability between compsets. Notably, both *FCHIST_slh* and *FCnudged_slh*
configurations simulate SGs values that are within the accepted range reported in the last Ozone Assessment Report (WMO,
2022) throughout the tropical troposphere. Indeed, our model results show a continuous conversion from SGs into PGs just
below the tropopause that results in a larger fraction of inorganic bromine (PGI) compared to the organic halocarbon fraction
(SGI). We highlight that once all organic VSL$_{Br}$ has been converted to reactive Br$_y$, the total contribution of SLH to the
stratospheric bromine loading remains at ~5 pptv roughly from the tropopause to the model top (grey line in Fig. 8b). Finally,
note that the dominant contribution of VSLs to the total SGI arises from the major CH$_2$Br$_2$ and CHBr$_3$ emissions, with minor
contributions from the remaining VSL bromocarbons (Fernandez et al., 2014). In contrast, CH$_2$IBr emissions, which constitute
the largest source of bromine mass to the atmosphere (Ordóñez et al., 2012), barely contributes with a negligible fraction to
the stratospheric SGI due to its particularly short lifetime.
Differing to bromine and due to the larger lifetime of chlorinated VSLs, the contribution to the total chlorine injection
(114-126 pptv) arising from SGI (91-97 pptv) is notably larger than that from PGI (17-35 pptv), in agreement with reported



ranges (WMO, 2022). Here note that due to the large anthropogenic contribution from developed regions over US, Europe and

Asia (Hossaini et al., 2019; Claxton et al., 2020), the $VSL_{Cl}$ distribution for the Northern and Southern mid-latitudes shown in

Fig. 4 shows a clear hemispheric asymmetry just below the tropopause. This asymmetry on VSL distributions have already

been described in the literature for chlorine (Roozitalab et al., 2024) as well as bromine (Keber et al., 2020; Jesswein et al.,

2022), although note that anthropogenic $VSL_{Br}$ sources are not considered in this work (see Section 2.2.2). For the case of

iodine, we highlight that due to the faster photochemistry compared to chlorine and bromine, almost the complete injection

occurs through $PGI = I_y$ (Fig. 8c), with a total injection of approximately 0.65-0.72 pptv, in agreement with observations and

previous estimations using CESM1 (Saiz-Lopez et al., 2015; Koenig et al., 2020).

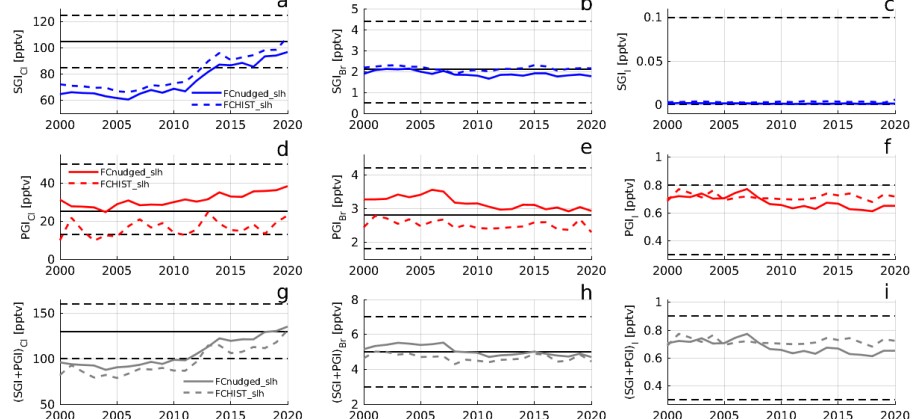

**Figure 9: Temporal evolution of source gas injection (SGI$_X$, top-row), product gas injection (PGI$_X$, middle-row) and total SLH injection (SGI$_X$ + PGI$_X$. bottom-row) between year 2000 and 2020 for the *FCnudged_slh* (solid lines) and *FCHIST_slh* (dashed lines) compsets. Individual results for (a, d, g) chlorine, (b, e, h) bromine and (c, f, i) iodine are shown on the left, center and right columns, respectively. Stratospheric injection values have been computed as the sum of the halogen mixing ratio of all species at the model tropical tropopause, considering the *compset_slh* output for organic VSL SGI$_X$ and the difference between *compset_slh* – *compset_noh* for PGI$_X$ (see text for details). The best estimation of the SLH contribution to the total halogen stratospheric injection for each individual family compiled in Table 1-6 of** (WMO, 2022) **is shown by the horizontal solid and dashed lines on each panel. Note the intermediate species COCl$_2$, CHCl$_2$O$_2$ and COFCl (see Fig. S3) has been considered to contribute to PGI$_{Cl}$ following** (WMO, 2022)**.**

Figure 9 shows the temporal evolution of SGI = VSL$_X$ and PGI = ΔX$_y$ occurring at the model tropical tropopause for

chlorine, bromine and iodine for the *FCnudged_slh* and *FCHIST_slh* compsets, where the modelled values are compared with

the latest assessed values of SLH injection to the stratosphere (i.e., see Table 1-6 in (WMO, 2022)), which are assumed constant

along the recent past. Most notably, the contribution of SLH to PGI for all three halogens does not show a significant trend

and lies within the range of WMO-estimated values throughout the entire modelling period. For the particular case of bromine

and chlorine, note that the *FCnudged_slh* results show slightly larger values than the *FCHIST_slh* compset, which are coherent

with the changes in the representation of convective transport within the tropics and the nudging to observed trends in the

Brewer-Dobson circulation. Regarding SGI$_{Cl}$, note that model simulations before 2010 show a negative bias in comparison

with the observational mean within the tropics, presenting an increasing trend after 2010 which is in agreement with previous

reports (Hossaini et al., 2019; WMO, 2022). Here we recall that in CESM2-SLH we applied a 15% enhancement to the natural

oceanic bromine source as well as to the anthropogenic chlorine emissions in comparison to CESM1 (see Section 2.2.2). Based

on Fig. 9a, we recognize that current CESM2-SLH configuration lies at the low edge of the estimated $VSL_{Cl}$ contribution, and

therefore the original emissions and LBCs for anthropogenic chlorinated halocarbons from (Hossaini et al., 2019; Claxton et

al., 2020) could be increased by a factor larger than 15% to reproduce previous stratospheric chlorine SGI results with CESM1

and other models (Li et al., 2022; Saiz-Lopez et al., 2023; Hossaini et al., 2024). Having said this, we acknowledge that the



model underestimation of $SGI_{Cl}$ to the stratosphere should not represent a significant bias in our model implications on other
atmospheric components. This is because chlorine influence on stratospheric ozone is dominated by long-lived CFCs and
HCFCs, while in the troposphere $VSL_{Cl}$ photodegradation represent only a minor contribution to atomic Cl release, which is
largely dominated by the acid-displacement online sources (see Fig. 7 and Section 3.1). In quantitative terms, we highlight that
for bromine, SLH contributes to a significant enhancement on present-day stratospheric halogen loading which represents
approximately 25-26% in all the different configurations, while for chlorine, given the larger contributions of CFCs and HCFCs
to the total stratospheric chlorine loading, the contribution of SLH represents only an increase 3.7-3.8%, in agreement with the
latest estimation from (WMO, 2022). Finally, the iodine SGI contribution is almost zero and all the iodine injection to the
stratosphere occurs as PGI (Fig. 9c).

**3.2.3 Geographical Heterogeneity and Seasonality of SLH**
To highlight the geographical heterogeneity of SLH abundances, Figure 10 shows the spatial distribution of carbon-
bonded VSLs (top row), total inorganic halogens (middle-row) and the most relevant reactive fraction (bottom-row) of each
halogen family over the boundary layer (i.e., mean modelled value from the surface up to ~850 hPa). Note that depending on
the halogen family considered, the bottom row presents the most relevant reactive species for each family; namely Cl atom for
chlorine, BrO for bromine and the sum of $I + IO = IO_x$ for iodine. Several distinctive features are observed for each individual
family as detailed below:
Chlorine: Figure 10a shows a pronounced North-South latitudinal gradient for $VSL_{Cl}$ between 250 and 200 pptv,
driven by the major anthropogenic emissions arising from the NH, particularly over China where maximum values exceed 500
pptv. The inorganic $Cl_y$ abundance also maximizes over the NH, with hot-spots over coastal oceanic and continental locations
where the mixing of SSA-halogen rich and enriched-nitrogen plumes (with high levels of $HNO_3$ and $N_2O_5$) results in an
enhanced acid-displacement source (see section 2.2.3). Maximum Cl atom values are found over the coastal regions of Europe,
US and East Asia, with maximum values smaller than $0.4 \times 10^{-3}$ pptv and a global mean average of $4 \times 10^{-5}$ pptv, which are
consistent with maximum and mean $Cl_y$ levels of ~640 pptv and ~47 pptv, respectively.
Bromine: This species presents a more homogeneous $VSL_{Br}$ distribution with the largest abundances found over the
equatorial regions due to the dominant oceanic sources arising from the tropical oceans (Ordóñez et al., 2012). Minimum,
maximum and mean global $VSL_{Br}$ mixing ratios reach 2.9 pptv, 12.9 pptv and 6.8 pptv, respectively. In line with chlorine,
modelled $Br_y$ also maximizes over coastal regions co-located with high SSA abundance (Fig. S1). However, the inter-
hemispheric bromine enhancement is less pronounced than for chlorine because SSA-debromination does not directly depend
on $HNO_3$ and $N_2O_5$ abundance, but rather on the partitioning shift of $Br_y$ to $BrONO_2$, $BrNO_2$ and HOBr. Indeed, despite
maximum $Br_y$ peaks in the NH are up to 2 times larger than in the SH, the corresponding maximum BrO abundances are of
equivalent magnitude (between 1.1 and 1.2 pptv, respectively) compared to the NH, highlighting the strong dependence of
halogen partitioning on the background atmospheric composition of $NO_x$ and $HO_x$.
Iodine: Despite high local emissions of $CH_3I$, primarily from rice paddies over east Asia (Fig. 10c; see (Ordóñez et
al., 2012)), the main near-surface $VSL_I$ and $I_y$ distributions depends on $HOI/I_2$ oceanic emissions that peak within the tropics.
The mean modelled $I_y$ and I+IO values reach ~1.25 pptv and ~0.15 pptv, respectively, with corresponding (minimum –
maximum) mixing ratios ranging between (0.05 - 21) pptv and (0.008 - 0.90) pptv over the boundary layer. Notably, the mean
$IO_x / I_y$ partitioning for the global mean average reaches ~0.12, while at locations with the highest values of $I_y$, the $IO_x / I_y$ ratio
remains below ~0.04.

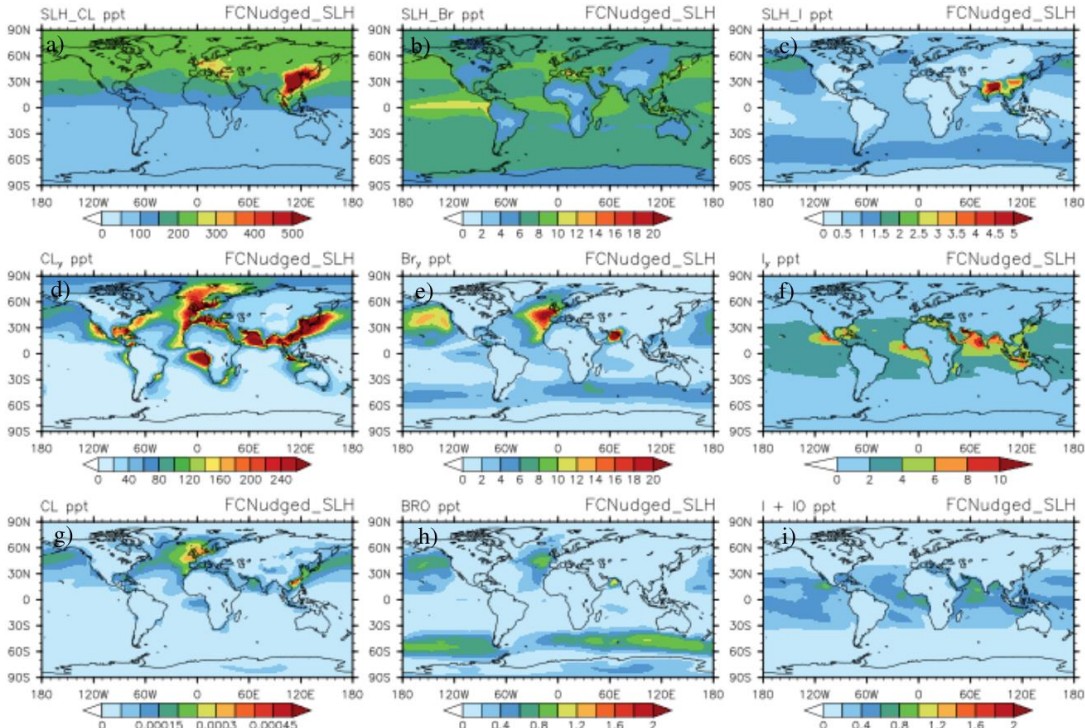

**Figure 10: Annual mean geographical distribution of organic and inorganic halogens during present-day (2015-2020) for the _FCnudged_slh_ compset. Top row shows the very short-lived (VSL) distribution for a) chlorine (VSL$_{Cl}$), b) bromine (VSL$_{Br}$) and c) iodine (VSL$_I$), while middle-row show the distribution of the total inorganic halogen content for d) chlorine (Cl$_y$), e) bromine (Br$_y$) and f) iodine (I$_y$). The bottom-row show the corresponding distribution for the most important reactive halogen species for each family: g) Cl atom, h) bromine monoxide (BrO) and i) reactive iodine (IOx = I + IO). All magnitudes have been averaged within the boundary layer (from the model surface up to ~850 hPa).**

Figure 11 presents the seasonal variation of surface organic and inorganic halogen species averaged within different latitudinal bands. To highlight the distinctive behaviour between the northern (NH) and southern (SH) hemispheres, we distinguish between the tropics (20ºN-20ºS), and the extratropical domains in the NH (20ºN-90ºN) and the SH (20ºS-90ºS). The stacked bars shown in Fig. 11 represent the total halogen abundance decomposed into the reactive (XO$_x$), reservoir (X$_y$) and organic (VSL$_X$) species for each of the the _FCnudged_slh_ and _FCHIST_slh_ compsets. The corresponding zonal average seasonality of VSL$_X$ halocarbons, as well as the geographical X$_y$ seasonal cycle over the boundary layer are shown in Figures S4 and S5, respectively. We first describe the seasonality in the NH extratropics and then highlight the main similarities and differences with the SH extratropics and the tropical mean.

The NH extratropical Cl$_y$ presents a pronounced seasonal cycle with approximately doubled values during the boreal winter (~90-100 pptv) compared to the summertime (~45-50 pptv). This is coherent with the seasonal variation of the heterogeneous recycling efficiency that reduces the conversion of both chlorinated (e.g., ClONO$_2$) and nitrogen (e.g., N$_2$O$_5$ and HNO$_3$) reservoirs to more reactive species. For the same reason, the organic VSL$_{Cl}$ and VSL$_{Br}$ abundances also peak during the winter when the photochemical decomposition is smallest (see Fig. S4). In contrast, the iodine seasonal cycle shows maximum abundances during the summertime, both for I$_y$ and VSL$_I$. This is explained by the dominant contribution of oceanic sources for iodine, which increase associated with the largest fraction of ice-free ocean in winter and the enhanced tropospheric ozone levels in winter promoting deposition over the ocean. The inorganic Br$_y$ abundance is lower and maximizes during the spring for _FCnudged_slh_ in comparison with _FCHIST_slh_, which shows larger values and a maximum peak during the



summer, due to the larger efficiency of SSA-dehalogenation recycling in *FCHIST_slh*. Here we recall that sea-ice polar halogen

emissions are not yet considered in CESM2-SLH, and therefore, the modelled seasonal cycle observed for the different halogen

species differs from those described in the NH high- and mid-latitudes as well as near Antarctica in previous work (Fernandez

et al., 2019, 2024).

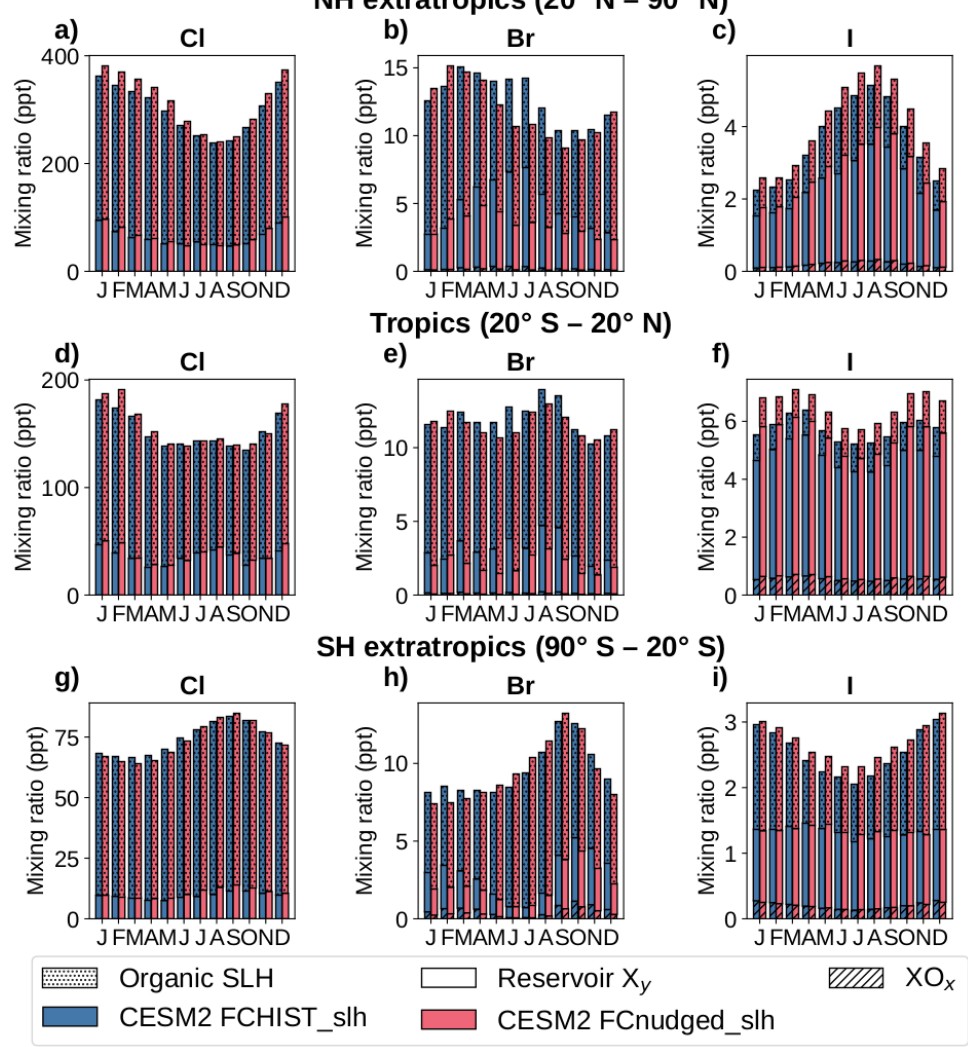

**Figure 11: Seasonal distribution of total SLH abundance during present-day (2015-2020) for (left column) chlorine, (center column) bromine, and (right column) iodine. Model results for the Northern Hemisphere (NH = 20°N - 90°N) and Southern Hemisphere (SH = 20°S - 90°S) extra-tropical regions are shown in the top and bottom rows, respectively, while Tropical (20°N – 20°S) mean values are shown in the middle-row. Each panel distinguish the contribution from reactive (XO$_x$, striped bars), reservoir (X$_y$, empty bars) and organic VSL$_X$ (stippled bars) halogens using coloured stacked bars for the *FCnudged_slh* (pink) and *FCHIST_slh* (blue) compsets.**

The predicted seasonal cycle of iodine over the SH extratropical regions presents a similar pattern to that in the NH,

although with a 6-months shift, mostly for VSL$_I$. However, and due to the much lower influence of anthropogenic ozone

pollution over the SH, the I$_y$ abundance remains less variable throughout the entire year. In contrast, VSL$_{Br}$ show larger

abundances during the austral winter due to the reduced photodecomposition. The same seasonal pattern is also observed for



VSL$_{Cl}$, although we note that total mixing ratio are remarkably lower in the SH compared to the NH extratropics. Larger Br$_y$ abundance is predicted during austral spring and summer following the seasonal changes in sea-salt distributions particularly over the Southern Ocean, resulting in higher Br$_y$ enhancement for *FCHIST_slh* than for *FCnudged_slh* (see Fig. 11h). Is worth noting that over the SH extratropical latitudes, the Cl$_y$ seasonality follows that of Br$_y$, which is explained by the fact that within this region, the SSA-dehalogenation flux for chlorine (which is linked to that for bromine, see Table 1) dominates over the acid-displacement reaction controlling the total chlorine flux over the NH.

Finally, the tropical VSL$_I$ abundance is the lowest compared to the NH and SH extratropics due to the higher photochemical decomposition close to the Equator. In contrast, I$_y$ abundance is the largest due to the dominant I$_2$/HOI source (Fig. 1f). On the other side, and driven by the much larger lifetimes of VSL$_{Br}$ and VSL$_{Cl}$ compared to iodine, the tropical abundances of VSL$_{Br}$ are larger than over the extratropical regions (Fig. 11e), while VSL$_{Cl}$ show intermediate abundances between the NH and SH extratropics. In both cases, surface VSL$_{Cl}$ and VSL$_{Br}$ tropical surface abundances lack of a pronounced seasonal cycle, although a regional seasonality is predicted in their contribution to the SGI due to the changes in their convective transport to the tropical tropopause layer (Fernandez et al., 2014, 2021). The double peak annual cycle in reservoir X$_y$ observed within the tropics is attributed the seasonal changes in atmospheric rainout during the wet and dry seasons that affect the washout of inorganic halogens.

### 3.3 Influence of SLH on Atmospheric Composition

This section describes how the changes in SLH influence the baseline abundance and distribution of other atmospheric components (O$_3$, OH and NO$_2$), which in turn determine the oxidative capacity of the atmosphere. We begin quantifying the global mean surface and tropospheric responses to SLH, then address the zonal, latitudinal and vertical distribution of the halogen-driven chemical changes, and finalize with a comprehensive description of the SLH-induced perturbations on the modelled ozone and OH budgets and burdens for the different *FCnudged_slh* an *FCHIST_slh* compsets in 2°×2° resolution. A comparison with previous work using CESM1 is also included to facilitate the interpretation and contextualization of CESM2-SLH results, pointing at specific references where additional details have been provided for individual processes and/or locations.

Table 9 presents the global mean O$_3$, OH and NO$_2$ abundance represented by the main CESM2-SLH compsets during the 2015-2020 period, distinguishing the absolute and percentage differences between equivalent simulations including (*compset_slh*) and neglecting (*compset_noh*) SLH within the surface and for the entire troposphere. Including SLH chemistry in the model results in a significant (16.9-17.3%) reduction in the tropospheric ozone burden, decreasing from 336-345 Tg O$_3$ for the *FCnudged_noh* and *FCHIST_noh* configurations to 278-287 Tg O$_3$ for the *FCnudged_slh* and *FCHIST_slh*, respectively. The negative O$_3$ response is more remarkable at the surface with a reduction of 21.4-22.5%. Given the general higher ozone abundance in CESM2-SLH compared to CESM1 (Emmons et al., 2020), the absolute SLH-driven tropospheric and surface ozone reductions are larger for *FCnudged_slh* and *FCHIST_slh* compared with *Cyclical_slh* and *Serial_slh*, although we highlight that the corresponding percentage decreases of global mean ozone at the surface (~24.4%) and the troposphere (~16.9%) are in very good agreement with previous CESM1 studies, ranging from −13% to −24% at the surface and from −16% to −20% in the troposphere (Iglesias-Suarez et al., 2020; Badia et al., 2021; Barrera et al., 2023; Saiz-Lopez et al., 2023). In addition, previous CESM1 estimations of stratospheric ozone depletion due to SLH were −5.2 DU (−2.0%) (Saiz-Lopez et al., 2023), which falls within our CESM2-SLH simulated values of ozone depletion between −6.4 DU and −4.0 DU (−2.5% and −1.6%), respectively obtained with the *FCnudged_slh* and *FCHIST_slh* configurations (see Section 3.3.4).



**Table 9: Quantitative changes in atmospheric composition between SLH and NoSLH sensitivities.**

| Species Abunance | | $O_3$ | | | OH | | $NO_2$ | |
|---|---|---|---|---|---|---|---|---|
| Compset | Sensitivity | Surface | Tropo& | Strat$ | Surface | Tropo& | Surface | Tropo& |
| | | ppbv | Tg | DU | ppqv | Mg | pptv | Gg |
| FCnudged_slh | NOH | 29.2 | 336.0 | 254.6 | 67.1 | 236.2 | 673.2 | 383.0 |
| | SLH | 22.9 | 277.9 | 248.2 | 65.2 | 222.8 | 669.2 | 355.4 |
| | Absolute Change | -6.2 | -58.1 | -6.4 | -1.8 | -13.4 | -4.0 | -27.6 |
| | Percentage Change (%) | -21.4 | -17.3 | -2.5 | -2.7 | -5.7 | -0.6 | -7.2 |
| FCHIST_slh | NOH | 31.1 | 345.6 | 251.8 | 70.4 | 240.2 | 639.7 | 381.8 |
| | SLH | 24.1 | 287.1 | 247.8 | 67.6 | 226.2 | 635.3 | 349.1 |
| | Absolute Change | -7.0 | -58.5 | -4.0 | -2.8 | -14.0 | -4.4 | -32.7 |
| | Percentage Change (%) | -22.5 | -16.9 | -1.6 | -3.9 | -5.8 | -0.7 | -8.6 |
| CESM1 (Cyclical_slh) | NOH | 27.8 | 317.1 | 264.4 | 53.0 | 209.2 | 313.3 | 266.0 |
| | SLH | 21.0 | 263.5 | 259.2 | 48.3 | 188.6 | 317.1 | 251.8 |
| | Absolute Change | -6.8 | -53.7 | -5.2 | -4.7 | -20.6 | 3.8 | -14.2 |
| | Percentage Change (%) | -24.4 | -16.9 | -2.0 | -8.8 | -9.8 | 1.2 | -5.4 |
| CESM1 (Serial_slh) | NOH | 28.3 | 326.0 | 259.8 | 48.4 | 212.7 | 320.5 | 263.3 |
| | SLH | 22.6 | 277.5 | 255.2 | 45.5 | 198.6 | 326.6 | 256.6 |
| | Absolute Change | -5.7 | -48.5 | -4.6 | -2.8 | -14.0 | 6.0 | -6.8 |
| | Percentage Change (%) | -20.1 | -14.9 | -1.8 | -5.9 | -6.6 | 1.9 | -2.6 |

&All tropospheric magnitudes have been computed considering the chemical tropopause ($O_3$ < 150 ppbv).

Similar to ozone, SLH sources and chemistry globally reduce OH and $NO_2$. Regardless of the compset, this reduction
ranges between −2.7% and −3.9% at the surface and approximately −6% integrated in the troposphere for OH, while for $NO_2$
differences remain below −1% at the surface and ranging between −7.2% and −8.6% in the troposphere. In comparison with
CESM1, the percentage change induced by SLH in CESM2-SLH is smaller for OH and larger for $NO_2$ (see Table 9). Note that
not only the absolute abundance but also the $HO_2$/OH and $NO_2$/NO ratio are significantly different between CESM1 and
CESM2 (see Fig. 2k,l), which also impact on the estimated global mean values. Given the large number of factors influencing
the rapid interconversion between $HO_x$ and $NO_x$ species, such as the strength and spatial distribution of sources and the changes
in chemical schemes between CESM versions, a deeper analysis of the dominant processes controlling these changes is
encouraged for future studies.
Figures 12 and 13 show the absolute and percentage change, respectively, induced by SLH over the atmospheric
abundance of $O_3$, OH and $NO_2$ between the *FCnudged_slh* and *FCnudged_noh* compsets. The changes are shown for the zonal
average (top), within the boundary layer (middle) and integrated in the troposphere (bottom row). Despite distinctive features
for each individual compound, SLH globally reduce the abundance of all species, particularly over the ocean and at high
latitudes, with exception of some continental regions where the model shows an enhancement in OH and $NO_2$ abundance. The
largest reductions in ozone mixing ratios occur in the lower stratosphere, although due to the high $O_3$ levels within the ozone
layer, the tropospheric percentage changes are larger than in the stratosphere. Similarly, the larger background ozone
abundance over the NH results in larger absolute changes north of the Equator, although in percentage terms, the largest
changes are observed in the SH, with minimum ozone changes occurring over the tropical regions.





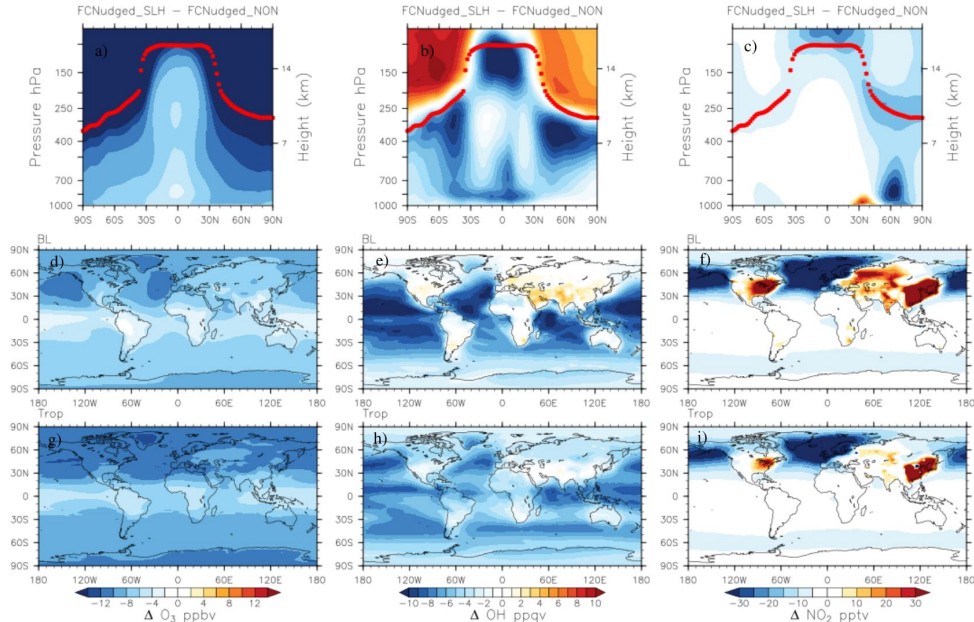

**Figure 12: Geographical distribution of the SLH effect on the O₃, OH and NO₂ abundances during present-day (2015-2020). First row: zonal average vertical distribution of a) O₃, b) OH and c) NO₂. Second row: boundary layer mean (below 850 hPa) for d) O₃, e) OH and f) NO₂. Third row: tropospheric mean (surface up to the tropopause) for g) O₃, h) OH and i) NO₂. The absolute difference computed as (*FCnudged_slh − FCnudged_noh*) is shown in all panels. Equivalent results with the percentage change are shown in Fig. 13.**

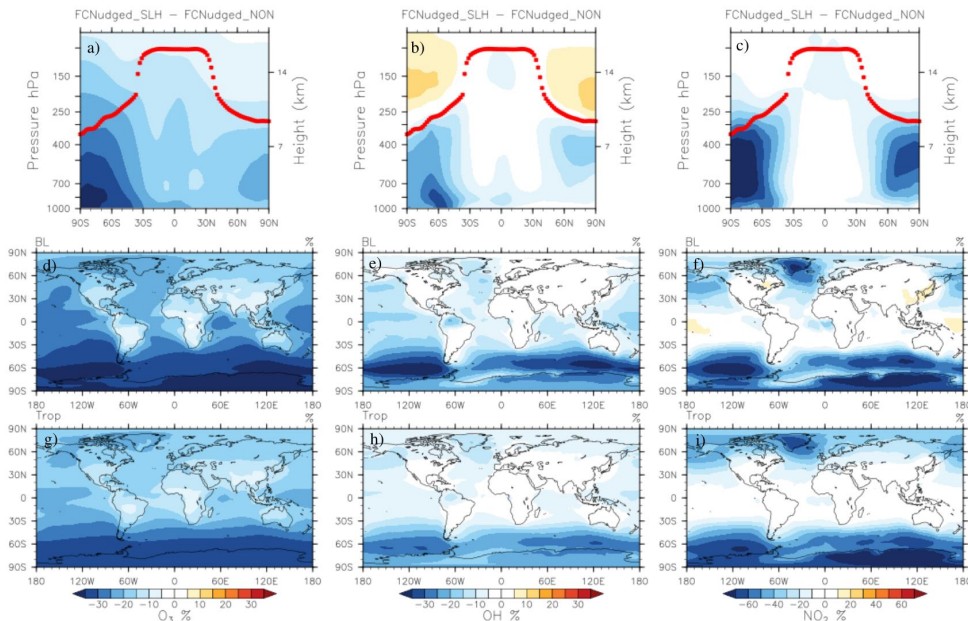

**Figure 13: Geographical distribution of the SLH effect on the O₃, OH and NO₂ abundances during present-day (2015-2020). First row: zonal average vertical distribution of a) O₃, b) OH and c) NO₂. Second row: boundary layer mean (below 850 hPa) for d) O₃, e) OH and f) NO₂. Third row: tropospheric mean (surface up to the tropopause) for g) O₃, h) OH and i) NO₂. The percentage difference computed as (*FCnudged_slh − FCnudged_noh*) / *FCnudged_noh* × 100% is shown in all panels. Equivalent results with the absolute difference are shown in Fig. 12.**





Given the dominant role of $O_3$ as the primary source of OH in the troposphere, the spatial distribution of OH changes is similar to the analogous pattern of $O_3$. However, and driven by its higher reactivity, OH changes are more pronounced than for $O_3$, with the largest reductions over the ocean as well as some continental enhancements (Bossolasco et al., 2025). In percentage terms, OH reduction due to SLH is lower over tropical regions, where most of the global tropospheric oxidation takes place. Maximum percentage differences are found over the pristine southern ocean due to the smaller contribution of the dominant secondary OH source driven by $NO_x$ (see Section 3.3.2) compared with the north Atlantic and Pacific oceans, which presents the larger absolute differences. Similarly, the large $NO_2$ changes exceeding 50 pptv over continental regions barely represent a small percentage. Indeed, and despite the large heterogeneity, the global mean change of surface $NO_2$ remains below ~1% (see Table 9).

### 3.3.1 Evaluation of Model Performance with Air-Pollutants

To evaluate the performance of CESM2-SLH, surface ozone abundance from the compsets *FCnudged_noh* and *FCnudged_slh* was compared with observations reported in the first phase of Tropospheric Ozone Assessment Report (TOAR-I) (https://igacproject.org/activities/TOAR/TOAR-I) (Schultz et al., 2017). The central panel in Figure 14 shows the geographical distribution of annual mean surface ozone for the *FCnudged_slh* compset during year 2015, along with 12 surrounding panels showing the seasonal comparison with observational sites at coastal or close-to-coastal locations (red points). The central panel shows that the model reproduces the hemispheric asymmetry in surface ozone, with background levels below 40 ppbv in the pristine SH, contrasting with peak concentrations (60–75 ppbv) over continental and coastal Asia. Moreover, the ozone peak on the African continent is mainly linked to biomass burning. These maxima values are consistent with those reported by previous studies using CESM2 (Emmons et al., 2020). Moreover, Fig. 14 also shows a visual comparison of the monthly (lines) and annual mean (markers) surface ozone concentrations between the *FCnudged_slh* (red) and *FCnudged_noh* (blue) simulations and TOAR-I observations (black) across multiple stations located between latitudes 60N-60S. Overall, the model captures the observed seasonal ozone cycles across most stations. This agreement is particularly strong at northern mid-latitude stations (>30°N), where the model reproduces both the magnitude and seasonality of ozone, with peaks in spring-summer driven by enhanced photochemical production. Similarly, good representation of ozone seasonality is observed at station 7, located in central Chile as well as in Station 8 east of Madagascar. The inclusion of SLH sources and chemistry in the *FCnudged_slh* simulation improves the agreement with observations in terms of both monthly variability and annual mean, particularly at coastal-stations. However, discrepancies emerge at some stations like 11 and 12, where the model successfully reproduces the observed ozone magnitude but fails to capture their seasonal cycle. This divergence likely stems from incomplete representation of ozone sources and sinks in this pristine region. Finally, note that the coarse resolution of the model is a factor that certainly impact in the comparison with in-situ observations, as the local ozone values in some observational sites do not necessarily represent the mean background levels predicted by a 1.9° × 2.5° pixel size in the model. Nevertheless, Fig. 14 clearly shows that the model values are in the same range as observations, reproduce the monthly variability, and demonstrate that the inclusion of SLH chemistry in the model helps to close the gap with observations.

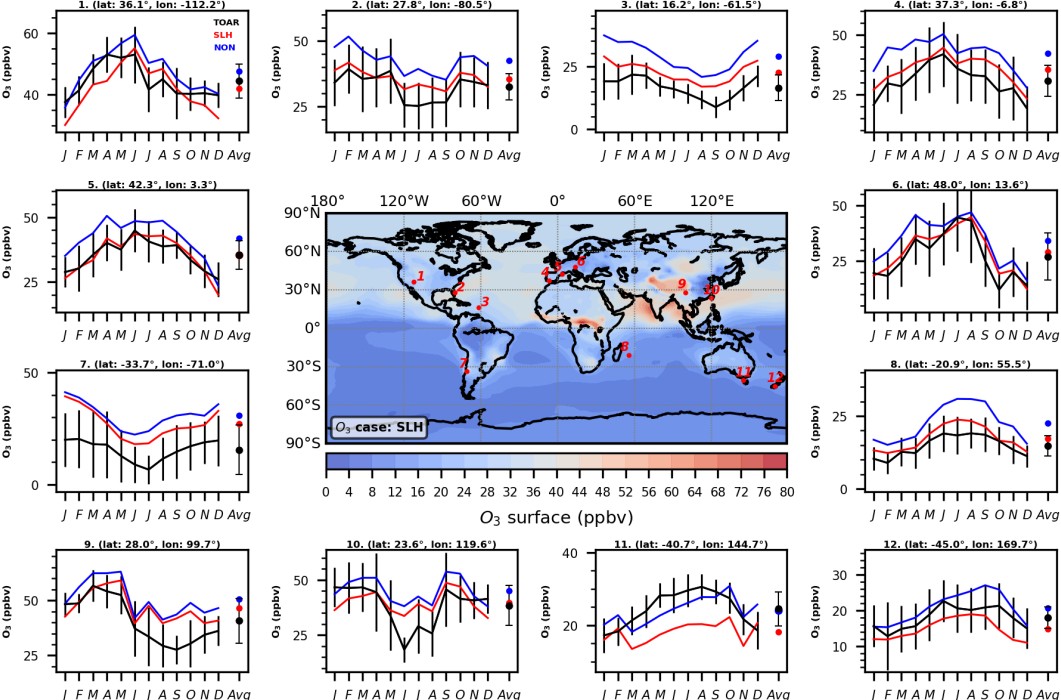

**Figure 14: Comparison of CESM2-SLH *FCnudged_slh* and *FCnudged_noh* compset with TOAR-I observations. The center panel shows the geographical distribution of surface ozone on *FCnudged_slh* compset and the observational sites location corresponding to the side panels (red points). The side panels show the comparison of the seasonal distribution and the annual average (marker) of surface ozone between *FCnudged_slh* (red) and *FCnudged_noh* (blue) simulations/compsets with respect to surface observations (black) reported in the first phase of Tropospheric Ozone Assessment Report (TOAR-I)** (Schultz et al., 2017) **(https://igacproject.org/activities/TOAR/TOAR-I). Both simulations and observations correspond to the year 2015.**

### 3.3.2 SLH Enhanced Ozone Chemical Loss

This section highlights the effect of SLH on ozone destruction by means of computing the odd-oxygen chemical loss (*OddOx_Loss*) as performed in previous studies, following the definitions presented in Table 5 of (Saiz-Lopez et al., 2014) (see Chart S2 in the Supplementary Material). Figure 15 shows the annual mean *OddOx_Loss* as a function of latitude and height for the *FCnudged_slh* compset, where the individual contribution of the different families contributing to ozone depletion are distinguished. Going from top to bottom, in the upper stratosphere, *OddOx_Loss* is dominated by $NO_x$ cycles (Fig. 15c), while from the middle and lower stratosphere down to the free-troposphere, the dominant *OddOx_Loss* family is $HO_x$ (Fig. 15b) until $O_x$ becomes the dominant ozone loss channel over the boundary layer (Fig. 15a). Note the halogen-driven *OddOx_Loss* shown in Fig. 15d-f maximize in the tropical upper troposphere (dominated by $IO_x$) and global lower stratosphere (both $BrO_x$ and $IO_x$ increase, with a minor role of $ClO_x$), where they significantly contribute to additional ozone destruction. Most of this halogen-driven ozone loss results from: *i)* faster recycling and gas-phase iodine chemistry compared to bromine and chlorine; and *ii)* the larger conversion of iodine to reactive product gases occurring in the lower troposphere. In addition, note the important contribution of $BrO_x$ and $ClO_x$ cycles on ozone depletion over the Antarctic lower stratosphere of the southern hemisphere high latitudes (Fig. 15e,f), which maximizes during spring and is responsible for the formation of the Antarctic ozone hole (see Section 3.3.4).





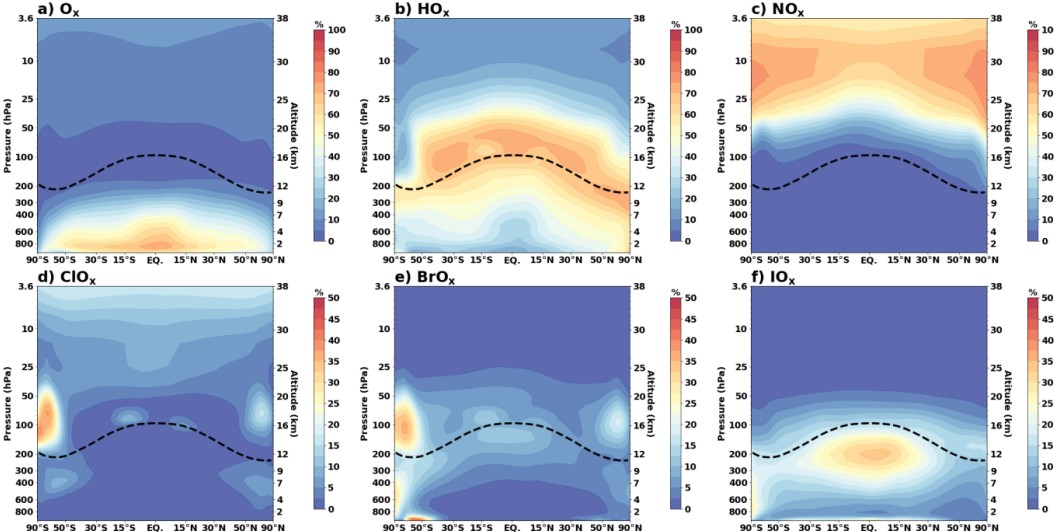

**Figure 15: Zonal average distribution of Odd-oxygen loss rates (*OddOx_Loss*) for different ozone depleting families during present-day (2015-2020). Each panel presents the percentage contribution of each *OddOx_Loss* family to the total ozone loss for the *FCnudged_slh* compset. The top-row show results for the main *OddOx_Loss* channels when SLH are not considered: a) O$_x$, b) HO$_x$ and c) NO$_x$ loss cycles, while the bottom-row shows the percentage contribution from the individual halogen-driven *OddOx_Loss* cycles: d) ClO$_x$, e) BrO$_x$ and f) IO$_x$. The black-dashed line indicates the model mean tropopause.**

Figure 16 shows the mean vertical distribution of the dominant *OddOx_Loss* channels averaged globally (top row) and within the tropical regions (bottom row). To highlight the influence of SLH on tropospheric chemistry, we quantify in absolute terms the contribution from each family for the *FCnudged_noh* and *FCnudged_slh* simulations, as well as the differences between them in a separate panel. The dominant ozone losses in the troposphere for the *FCnudged_noh* compset (Fig. 16a) are due to direct O$_x$ photolysis, followed by HO$_x$ cycles, both of which are significantly reduced when SLH are included, particularly in the lower and free troposphere (note NO$_x$ only contribute significantly to *OddOx_Loss* in the middle and upper stratosphere). Indeed, for the *FCnudged_slh* setup (Fig. 16b), the total contribution of halogens increases significantly and represents between 10% and 30% globally (Fig. 16c), with a variable vertical profile that is dominated by IO$_x$ cycles in the troposphere, and an increasing contribution mostly from BrO$_x$ close to the tropopause. To compensate the increase in the halogen contribution to total *OddOx_Loss*, Fig. 16c,f shows a proportional decrease in the *OddOx_Loss* destruction by HO$_x$ and O$_x$ in comparison with the *FCnudged_noh* sensitivity (negative change). Most notably, the largest halogen-driven contribution to total ozone loss is due to iodine chemistry, which represents up to 30% within the tropical free troposphere in agreement with previous estimates (Saiz-Lopez et al., 2012, 2014), followed by bromine and with a minor contribution by chlorine. Given that only the reactive fraction of inorganic Br$_y$ is capable of participate in ozone destruction, and most of its contribution occurs in the lower stratosphere (see Fig. 15e), an adequate representation of the SGI and PGI from bromine occurring over the tropical regions (Figs. 8 and 9) is of major importance to properly quantify the impact of SLH bromine and iodine on lowermost stratospheric ozone (see Section 3.3.4).





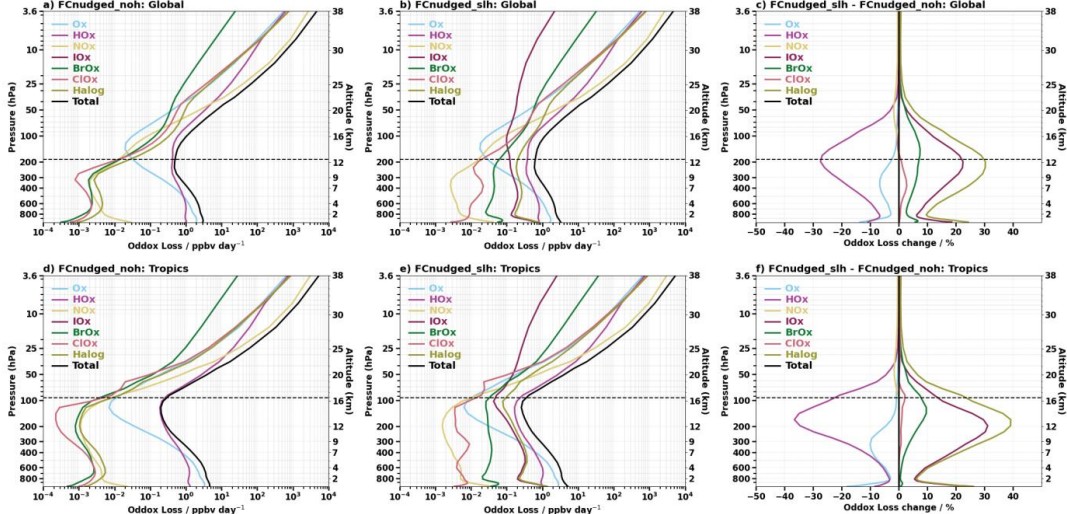

**Figure 16: Vertical profile distribution of Odd-oxygen loss rates (*OddOx_Loss*) cycles during present-day (2015-2020) for different model configurations neglecting (left-column; *FCnudged_noh*) and including (center column; *FCnudged_slh*) the contribution of SLH sources and chemistry. The right-column show the percentage difference between both compsets computed as (*FCnudged_slh − FCnudged_noh*) / *FCnudged_noh* × 100%. Results for the global mean (90°N – 90°S) are shown in the top row while tropical mean (20°N – 20°S) vertical profiles are shown in the bottom-row. The horizontal dashed line on each panel shows the mean model tropopause.**

### 3.3.3 Impact of SLH in the Troposphere

Table 10 summarized the annual mean odd-oxygen loss channels, as well as the net chemical ozone production and net chemical change for the different simulations considered. Globally, of the total halogen driven tropospheric ozone destruction (505 Tg yr$^{-1}$ for the *FCHIST_slh* and 522 Tg yr$^{-1}$ for the *FCnudged_slh*), IO$_x$-induced *OddOx_Loss* represents the dominant family, accounting for 76-79%, with minor contributions for bromine and chlorine that represents, respectively, 16-18% and 5%. These values show excellent agreement with previously reported halogen influences obtained with CESM1 (Badia et al., 2021; Barrera et al., 2023). Table 10 also shows the ozone photochemical production, which remains almost unaltered (with minor changes below 2%) regardless of the consideration or not of SLH sources and chemistry, consistent with previous studies. In contrast, and driven mainly by the larger ozone destruction when SLH are considered, the estimated O$_3$ dry deposition, typically considered in tropospheric budget analysis, is reduced by approximately 16% for *FCnudged_slh* and *FCHIST_slh* compared to their corresponding *FCnudged_noh* and *FCHIST_noh* configurations, which is in line with the 11% reduction previously estimated in CESM1. Finally, the net stratospheric to tropospheric exchange (STE) for *FCnudged_slh* and *FCHIST_slh* reaches 164 and 261 Tg yr$^{-1}$, respectively. Even though in absolute terms the contribution of STE to the tropospheric ozone budget in CESM2-SLH is smaller than for CESM1 (e.g., STE accounts for 341 Tg yr$^{-1}$ for the *Cyclical_slh* (Saiz-Lopez et al., 2023)), the inclusion of SLH compared to the no-halogen setup results in an equivalent halogen-driven reduction in the contribution of STE of approximately 20%, particularly for the nudging setup (Barrera et al., 2023). This highlights that, despite the dynamical changes in the representation of air subsidence and downward ozone transport across the tropopause between the old nudging over model layers or interfaces used in CESM1 (SD, specified dynamics) and the most recent nudging approach on model levels implemented in CESM2 (Davis et al., 2022), the chemical influence of SLH on tropospheric and stratospheric transport is of equivalent magnitude.

Table 11 shows the most important chemical pathways contributing to the formation of OH, where the individual contributions have been separated between the dominant primary production (*P*), which is the production of O$^1$D by ozone





photolysis at hv < 330 nm followed by reaction with tropospheric water vapour; and secondary production ($S$), which accounts
for all the remaining processes involving VOCs, $NO_x$ and $O_x$ (Lelieveld et al., 2016), as well as the reactions of hydrogen
peroxide and halogenated hypohalous acids (see (Bossolasco et al., 2025)). Due to the larger $O_3$ burden in CESM2-SLH
compared with CESM1, we note that the changes in OH production are also expected to differ between the different model
versions.

**Table 10: Tropospheric ozone budget with and without SLH**

| Simulation | FCHIST_slh | FCHIST_noh | FCnudged_slh | FCnudged_noh |
|---|---|---|---|---|
| $O_3$ photochemical production ($P_{O3}$; Tg yr$^{-1}$) | 5317.0 | 5490.0 | 5409.0 | 5580.0 |
| $O_3$ chemical Sinks (Tg yr$^{-1}$) | | | | |
| $O_x$-Loss | 2379.0 | 2713.0 | 2387.0 | 2734.0 |
| $NO_x$-Loss | 16.0 | 13.6 | 16.3 | 13.6 |
| $HO_x$-Loss | 1670.0 | 2061.0 | 1677.0 | 2097.0 |
| $ClO_x$-Loss | 34.0 | 4.0 | 30.0 | 3.9 |
| $BrO_x$-Los | 123.0 | 4.0 | 107.4 | 4.7 |
| $IO_x$-Loss | 505.0 | 0.0 | 522.0 | 0.0 |
| $Halog_x$-Loss | 662.0 | 8.0 | 659.4 | 8.6 |
| *Other $O_x$-Loss* | 73.8 | 78.4 | 101.8 | 104.1 |
| $O_3$ gross chemical loss ($L_{O3}$; Tg yr$^{-1}$) | 4800.8 | 4874.0 | 4841.5 | 4957.3 |
| $P_{O3}$ - $L_{O3}$ | 516.2 | 616.0 | 567.5 | 622.7 |
| $O_3$ dry depostion (*DryDep*) | 777.0 | 928.0 | 731.0 | 872.0 |
| $O_3$ STE ($P_{O3}$ - $L_{O3}$ - *DryDep*) | -261.0 | -314.0 | -164.0 | -251.0 |
| $O_3$ Burden (Tg) | 311.1 | 370.0 | 297.5 | 360.0 |
| $O_3$ Lifetime (days) | 23.6 | 27.7 | 22.4 | 26.5 |

All tropospheric magnitudes have been computed considering the internal model tropopause (TROP_P, hPa), defined
as the level where the minimum lapse rate pressure is derived.

The tropospheric ozone depletion induced by SLH result in a reduction of $P$ from approximately 116 Tg yr$^{-1}$ for the
*FCHIST_noh* and *FCnudged_noh* sensitivities (which represent approximately 47-48% of the total OH source) to
approximately 101 Tg yr$^{-1}$ (44-45%) for the *FCHIST_slh* and *FCnudged_slh* compsets. Similarly, the secondary ozone-driven
pathway ($O_3 + HO_2$) reduces from approximately 12% to 10% due to the inclusion of SLH, inducing minor alterations in the
other secondary channels that remain below a few percent (see Table 11). These reductions in $P$ and $S$ are compensated by
~5% increase in the contribution from the photolysis of HOCl, HOBr and HOI, which accounts for more than 11 Tg yr$^{-1}$ of
the global annual production. We note that the largest percentage contribution is observed within the free troposphere.
Consequently, there is a continuous shift in the chemical OH production from primary to secondary sources, which implies
that once formed, OH has an increased capability of being chemically re-generated in the atmosphere when SLH are
considered. Therefore, and despite SLH result in a net reduction of OH abundance, the tropospheric oxidation capacity of OH
on a per atom basis increases. This is typically measured by computing the OH recycling probability, $r = 1 - P / (P + S)$
(Lelieveld et al., 2016), which represents the capability of OH to be regenerated from secondary sources. Most notably,
considering SLH in CESM2 increases $r$ from approximately 52-53% to 55-56% (see Table 11). Despite percentage values for
$r$ being lower in CESM2-SLH than in CESM1, the overall global $P$ and $S$ terms are in line with previous studies using other





models (Lelieveld et al., 2002), while we also acknowledge that several anthropogenic halogen sources considered in the work
of (Bossolasco et al., 2025) have not been implemented in CESM2 (see Section 2.5.2).

**Table 11: Global OH production with and without SLH**

| Sources (Tmol yr⁻¹) Primary (*P*) | FCHIST_slh | FCHIST_noh | *FCnudged_slh* | *FCnudged_noh* |
|---|---|---|---|---|
| $O^1D + H_2O$ | 100.7 (44.5%) | 115 (48%) | 101.2 (44%) | 115.7 (47%) |
| Secondary | | | | |
| $NO_x + HO_2$ | 71.7 (31.7%) | 75.1 (31.4%) | 72.9 (31.5%) | 75.8 (31%) |
| $O_3 + HO_2$ | 23.1 (10.2%) | 28.7 (12%) | 24 (10.4%) | 30 (12%) |
| $H_2O_2 + h\nu$ | 13 (5.7%) | 14 (5.8%) | 15 (6.5%) | 16 (6.5%) |
| VOCs, ROOH +h$\nu$ | 5.8 (2.6%) | 6 (2.5%) | 6.6 (2.8%) | 6.8 (2.8%) |
| HOX + h$\nu$ | 11.1(4.9%) | - | 11.6 (5%) | - |
| Others | 0.26 (0.11%) | 0.28 (0.12%) | 0.26 (0.11%) | 0.28 (0.11%) |
| Total Secondary (*S*) | 125 (55.6%) | 124 (52%) | 130 (56%) | 129 (53%) |
| Net | | | | |
| Gross (*P* + *S*) | 226 | 239 | 231.6 | 245 |
| Recycling probability, r (%) | 55.6 | 52 | 56 | 53 |

The recycling probability (*r*) in percentage (%) is calculated as $1 - (P/G) \times 100\%$, where *P* is the primary
OH production channel, *S* is the net sum of secondary production sources and $G = P + S$ is the gross OH
production.

**3.3.4 Impact of SLH in the Stratosphere**
Finally, in this section we evaluate the combined role of SLH chlorine, bromine and iodine over stratospheric ozone
using CESM2-SLH, distinguishing the impact over the lowermost global stratosphere as well as its influence over the Antarctic
ozone hole as performed in previous studies (Saiz-Lopez et al., 2015; Fernandez et al., 2017; Cuevas et al., 2022; Barrera et
al., 2023; Villamayor et al., 2023). Figure 17a shows the change in $O_3$ partial column densities (DU) as a function of latitude
and height for *FCnudged_slh* − *FCnudged_noh*, where it is evident that SLH influence on stratospheric ozone maximizes in
the lowermost stratosphere just above the tropopause, which is of major importance as the radiative perturbations in this region
of the atmosphere are most sensitive to small changes in ozone abundance (Riese et al., 2012; Saiz-Lopez et al., 2012, 2023).
Most notably, the SLH influence on lowermost stratospheric ozone presents a significant latitudinal variation, with increasing
reductions moving from the Equator and tropical regions to the mid-to-high latitudes (Fig. 17b,c). This is due to the larger
conversion from SGI to PGI as air is transported from the tropical regions towards the poles through Brewer-Dobson
circulation, as well as due to the lower temperatures prevailing at high latitudes that accelerate the halogen chemical cycles.
Here is worth noting that, despite the SLH influence in lower stratospheric ozone is dominated by bromine and iodine chemistry
(see Figs. 15-16), the net effect of the additional halogen injection due to SLH is particularly sensitive to the background
inorganic chlorine ($Cl_y$) levels in the stratosphere (Barrera et al., 2020; Villamayor et al., 2023), and therefore the total SLH
impact will change for different periods of time following the overall stratospheric chlorine loading before and after the
turnover in the concentration peaks of ODS in the stratosphere following the Montreal protocol.



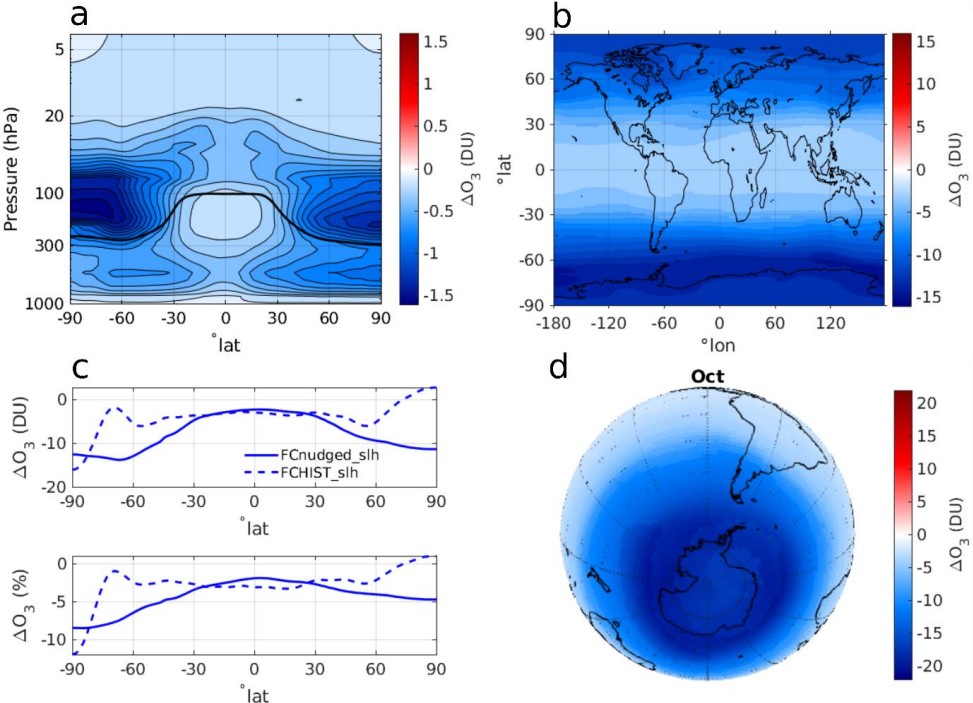

**Figure 17: Column integrated ozone change ($\Delta O_3$) in Dobson units (DU) due to SLH during present-day (2015-2020) for the (*FCnudged_slh* − *FCnudged_noh*) compset: a) Annual mean zonal average of partial columns. The solid black-line shows the tropopause height (TROP_P). b) Annual mean geographical distribution of $\Delta O_3$ in the lower stratosphere integrated between the tropopause and the 20 hPa model level. c) Absolute (top) and percentage (bottom) latitudinal distribution of $\Delta O_3$ for the *FCnudged_slh* (solid) and *FCHIST_slh* (dashed) compsets with respect to their corresponding *compset_noh*. d) South-polar view of stratospheric $\Delta O_3$ for the October mean integrated between the tropopause and ~20 hPa.**

Figures 17b and 17c shows the net $O_3$ change in the lowermost stratosphere integrated from the model tropopause up to ~20 hPa, where most of the SLH influence occurs. The $\Delta O_3$ ranges from −2.2-3.3 DU within the tropics (20ºN – 20ºS) up to −9.5-13.8 DU over the high-latitudes (poleward of 60º). These values are larger than those described in previous CESM1 works (Barrera et al., 2020; Fernandez et al., 2021) because those studies only considered SLH bromine, while the CESM2-SLH configuration used here includes SLH iodine as well as the anthropogenic contribution of anthropogenic VSL chlorine. These ozone changes represent, respectively, −1.8-2.7% and −4.0-8.5% of the lower stratospheric ozone for the annual mean, depending on latitude. Due to the efficient chemical coupling of SLH with the large levels of inorganic bromine and chlorine from long-lived CFCs and halons, the net influence of SLH on stratospheric ozone maximizes during austral spring over the South Polar Vortex, where SLH bromine and iodine increased the ozone hole destruction by more than −20 DU and −14%, in agreement with previous reports performed with CESM1 (Fernandez et al., 2017; Cuevas et al., 2022).

### 3.4 Comparison Between Different Model Resolutions and Configurations

All results using CESM2-SLH presented in this works have been performed using the *FCnudged_slh* and *FCHIST_slh* compsets with (2º×2º) resolution. We recommend selecting this coarse resolution configuration because, as highlighted above, several on-line halogen sources, heterogeneous recycling reactions and washout rates depend on the abundance, extent and distribution of CAM6-Chem atmospheric fields such as sea-salt aerosols, liquid-clouds and ice-crystals, which are well known to differ between CESM2 configurations. Specifically, for studies comparing SLH vs. noSLH, we recommend starting with





the *FCnudged_slh* compset. This ensures that dynamical transport is consistently represented across simulations, which is
crucial for isolating and quantifying chemical changes between sensitivities. Depending on the specific applications, many
users may need to develop other CESM2-SLH configurations differing from the ones considered in this work, and therefore
here we provide a basic guideline of model resolution and configuration inter-comparison that, at minimum, should be
performed to validate the SLH model performance. We emphasize that it is the final user's responsibility to properly evaluate
that the halogen abundance and distribution in any new user-defined configuration remain consistent with the results provided
in this work for the standard *FCnudged_slh* and *FCHIST_slh* (2°×2°) compsets. Below we provide a first-order general inter-
comparison of organic VSL and inorganic halogen distributions that is recommended to execute to validate model
performance. Given the large spatio-temporal heterogeneity of short-lived species, particularly $Cl_y$, $Br_y$ and $I_y$ (as well as their
reactive fractions $ClO_x$, $BrO_x$ and $IO_x$), additional validation for region- and period- specific output should be performed
against local observations to avoid obtaining unrealistic or biased results of the influence of SLH on atmospheric composition.

Figure 18 shows a comparison of the VSL zonal average for the 6 (six) different SLH compsets and resolutions

defined in Table 6 (see Section 2.5.1), where all distributions have been averaged for the 2000-2005 period. Most notably, all
configurations show an equivalent distribution for all halocarbons and families, with slightly lower values occurring at the
surface for the finer resolution (1°×1°) grids (*f09_f09_mg17*), particularly for $VSL_{Cl}$ (see Table 12). Figure 19 shows a
comparison of the tropical (20°N – 20°S) and global (90°N – 90°S) mean vertical profiles of $Cl_y$, $Br_y$ and $I_y$ for the same set
of simulations. In particular, for the *FWnudged_slh* and *BWHIST_slh* compset using WACCM 70L, a bi-linear interpolation
of model output to the default CAM6-Chem 32L vertical grid has been performed using the NCAR-provided *interpic* routine.
Despite minor discrepancies in the vertical distribution for the different halogen species, we highlight that all model
configurations result in a very similar profile all the way from the surface to the upper stratosphere. Indeed, the corresponding
tropospheric percentage variability ranges from −0.4% to +20 % for bromine and −24% to −0.2% for iodine (see Fig. S6),
where as expected, the largest percentage differences appear at those heights where the $X_y$ abundance is small. Note that for
the particular case of $Br_y$ within the boundary layer, the largest differences appear for the *FCHIST_slh* and *FCnudged_slh*
(2×2) configurations, due to the very different representation of sea-salt abundance and distribution, particularly over the
Southern Ocean (see Fig. 20). However, when integrated over the troposphere, note that the *FCnudged_slh* (1×1) and
*BWHIST_slh* configurations present a larger $Br_y$ burden (see Table 12). For the particular case of chlorine, which of all halogen
species show the largest range in $Cl_y$ abundance from the surface to the top of the model, tropospheric changes between the
different compsets remain below ~7.5% and maximize close to the tropopause. Here, we highlight that most of the modelled
$Cl_y$ change is not driven by the inclusion of SLH sources, but mostly due to the contribution of long-lived CFCs and HCFCs
that dominates the transport and distribution of inorganic chlorine in the stratosphere species within each independent case.
Indeed, equivalent $Cl_y$ changes for the different *compset_noh* are also predicted. With exception of the *BWHIST_slh*
configuration which results in an approximately 1 pptv $Br_y$ larger abundance at the model top, all sensitivities show an
equivalent stratospheric bromine and iodine loading regardless of the compset considered. Most notably, within the model top
level where most of the long-lived and short-lived halogens have been converted to inorganic chlorine, bromine and iodine,
the $\Delta Cl_y$, $\Delta Br_y$ and $\Delta I_y$, between the different compsets remain below ~5%.

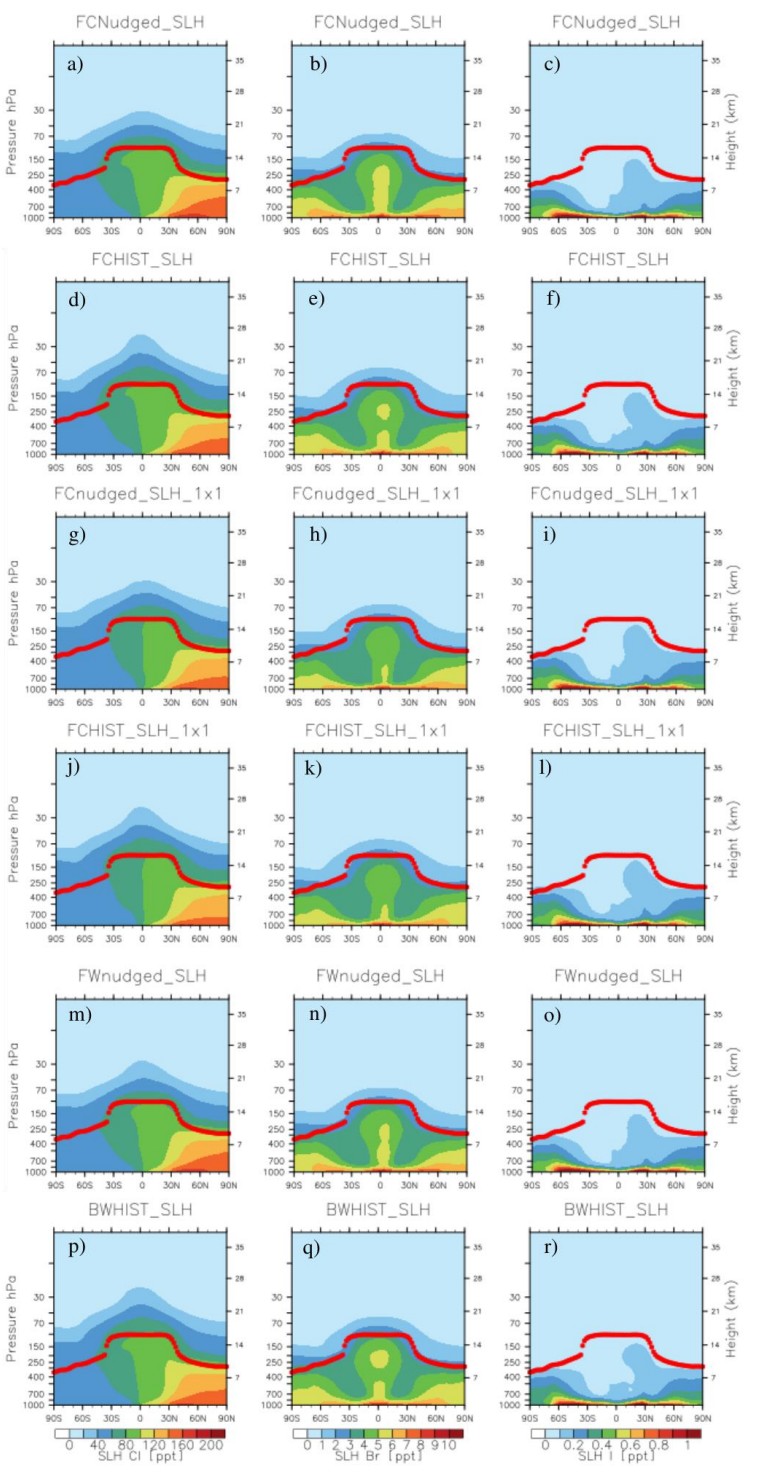

**Figure 18: Comparison of VSL abundance between different CESM2-SLH compsets and resolutions. Left, middle and right columns show the zonal average distribution for chlorine (VSL$_{Cl}$), bromine (VSL$_{Br}$) and iodine (VSL$_I$), respectively. Each row presents results for an individual CESM2-SLH configuration: FCnudged_slh 2×2 (1st row), FCHIST_slh 2×2 (2nd row), FCnudged_slh 1×1 (3rd row), FCHIST_slh 1×1 (4th row), FWnudged_slh 2×2 (5th row) and BWHIST_slh 2×2 (6th row). All output correspond to the annual mean for the 2000-2005 period. The red-dotted line shows the mean model tropopause.**



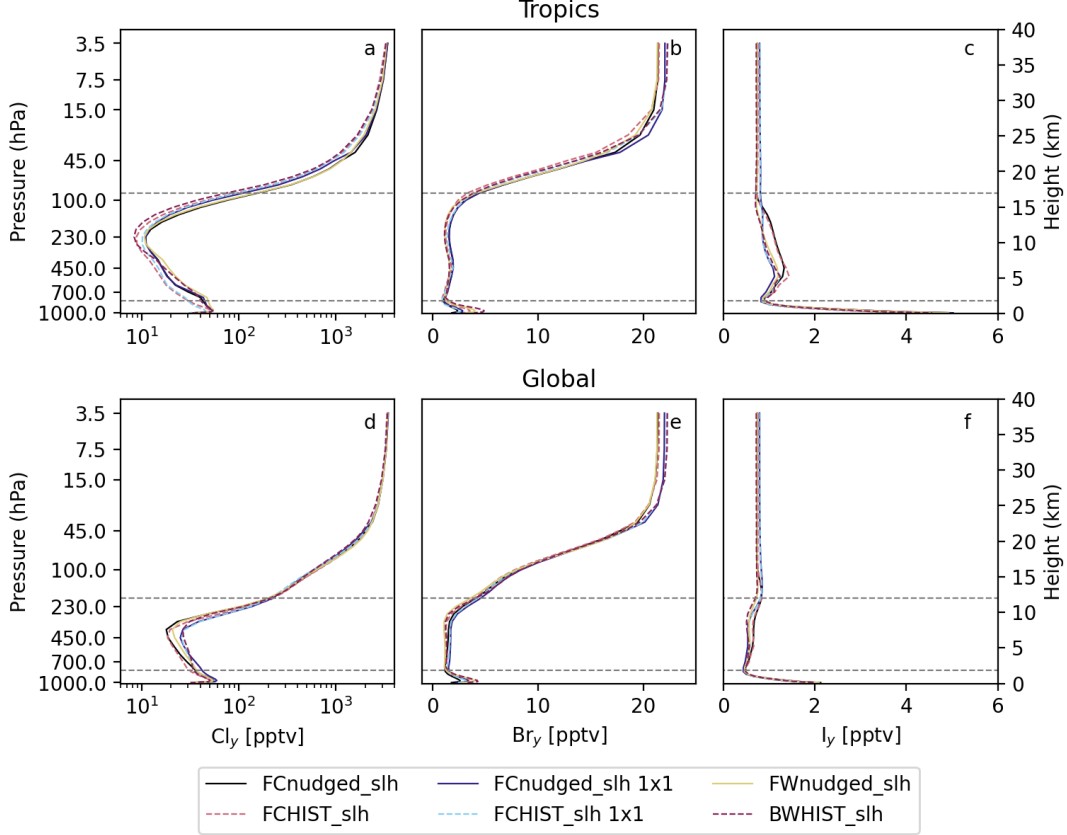

**Figure 19: Comparison of the Tropical (top-row) and global (bottom-row) vertical profile distribution of total inorganic halogen abundance ($X_y$) between the 5 different CESM2-SLH compsets and resolutions. Left, middle and right columns show results for (a, d) chlorine, (b, e) bromine and (c, f) iodine. The dashed lines shows the global mean height of the global tropopause and boundary layer.**

Figure 20 shows the geographical distribution of $Cl_y$, $Br_y$ and $I_y$ abundance averaged within the boundary layer for the different model configurations. Although all compsets show comparable global distributions and mean values (see Table 12), we highlight that the maximum abundance and partitioning between reactive and reservoir species within regional hot-spots present significant differences. Particularly remarkable are the $Br_y$ differences between the *FCHIST_slh* and *BWHIST_slh* compsets over the North Atlantic and North Pacific oceans, as well as over the Southern ocean. This is coherent with the larger SSA-dehalogenation sources simulated in these regions, which affect the $BrO/Br_y$ partitioning. Similarly, the higher (1×1) resolution for *FCHIST_slh* and *FCnudged_slh* display greater $Cl_y$ abundances over the oceanic outflow of polluted air masses from Europe, Eastern US and East Asia. This is associated with to the enhancement of the acid-displacement reaction occurring when $HNO_3$ air-masses are mixed with halogen rich SSA-oceanic fresh air.

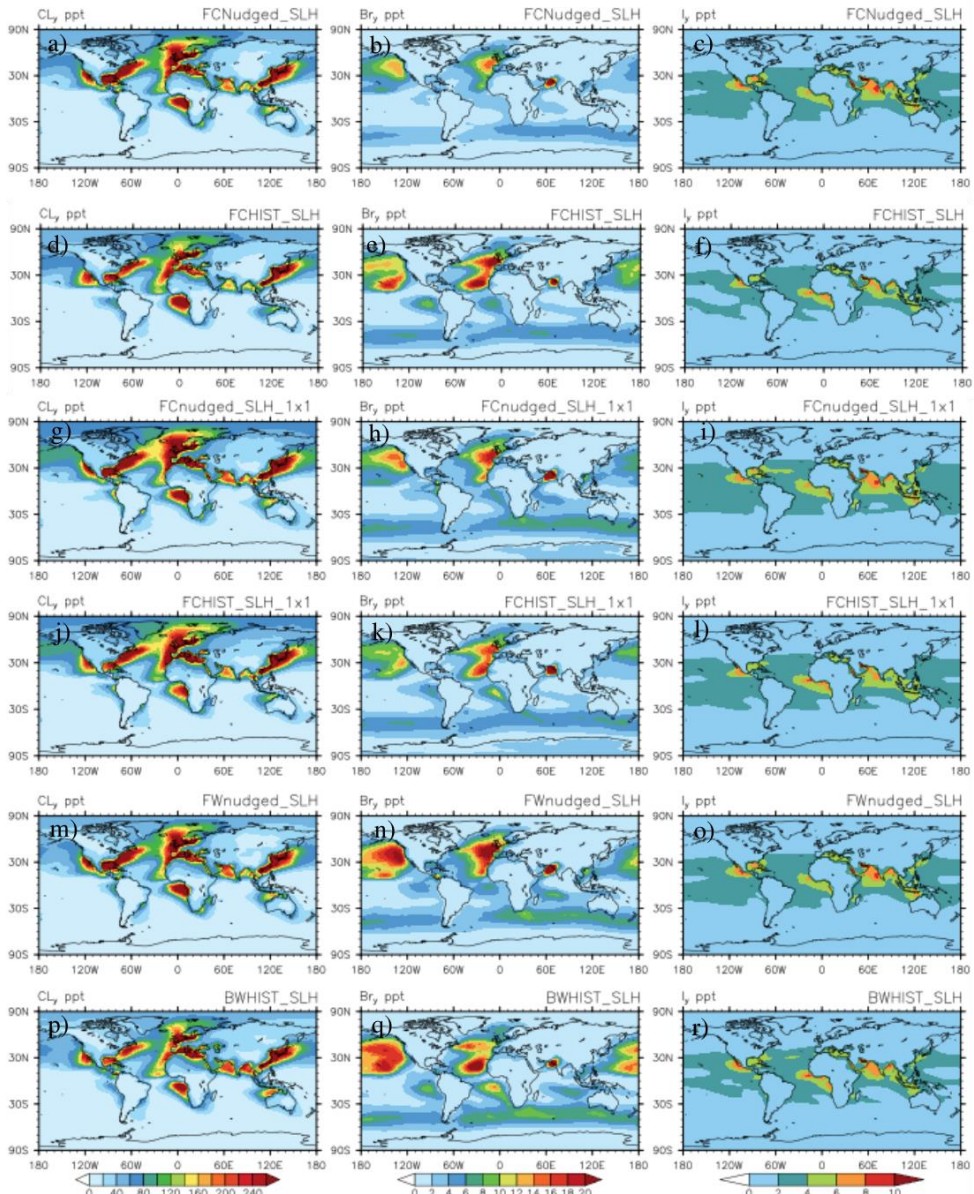

1572

**Figure 20: Comparison of the geographical distributions between different CESM2-SLH compsets and resolutions. Left, middle and right columns show the 2000-2005 annual mean distribution over the boundary layer for inorganic chlorine ($Cl_y$), bromine ($Br_y$) and iodine ($I_y$), respectively. ($VSL_{Cl}$), Each row presents results for an individual CESM2-SLH configuration: FCnudged_slh 2×2 (1st row), FCHIST_slh 2×2 (2nd row), FCnudged_slh 1×1 (3rd row), FCHIST_slh 1×1 (4th row), FWnudged_slh (5th row), and BWHIT_slh 2×2 (6th row).**

Finally, Table 12 summarize a comparison of the global mean surface and tropospheric halogen abundance of
halogenated VSLs and inorganic halogens across the different model configurations assessed, as well as the changes in $O_3$,
OH and $NO_2$ abundance between *compset_slh* and *compset_noh* sensitivities. We highlight that all values and results presented
in Section 3.4 were obtained considering the compset-individual *&slh_nl* scaling factors compiled in Table 7, which should
be taken as the starting point to adjust SLH and $X_y$ abundance in CESM2-SLH for any other model configuration besides those
that have been addressed here.





**Table 12: Global annual mean surface and tropospheric SLH loading and the corresponding changes in O₃, OH, and NO₂ abundance for different model compsets and resolutions.**

| Species | Simulation | Units | FCnudged_slh | FCHIST_slh | FCnudged_slh_1x1 | FCHIST_slh_1x1 | FWnudged_slh | BWHIST_slh |
|---|---|---|---|---|---|---|---|---|
| **Offline Sources** | | | | | | | | |
| VSL_Cl | | Tropo (Gg yr⁻¹) | 1624.0 | 1658.8 | 1587.3 | 1603.7 | 1644.1 | 1652.9 |
| VSL_Br | | Tropo (Gg yr⁻¹) | 715.5 | 715.4 | 715.3 | 715.2 | 717.7 | 716.7 |
| VSL_I | | Tropo (Gg yr⁻¹) | 602.6 | 601.6 | 602.0 | 601.3 | 602.8 | 602.3 |
| **Online Sources** | | | | | | | | |
| Oceanic Iodine | | Surface (Gg yr⁻¹) | 2391.4 | 2189.0 | 2234.4 | 2143.6 | 2258.4 | 2321.3 |
| SSA-dehal. Br | | Tropo (Gg yr⁻¹) | 2323.7 | 3886.1 | 2961.4 | 2840.7 | 4224.9 | 5208.5 |
| SSA-dehal. Cl | | Tropo (Gg yr⁻¹) | 3212.0 | 4051.7 | 4024.5 | 3730.4 | 4054.2 | 4686.6 |
| Acid-displacement | | Tropo (Gg yr⁻¹) | 20488.2 | 19036.5 | 20926.6 | 19212.2 | 20906.2 | 20936.3 |
| **Abundance** | | | | | | | | |
| **Short-Lived Halogens** | | | | | | | | |
| VSL_Cl | | Surface (pptv) | 101.6 | 98.4 | 103.0 | 101.3 | 100.3 | 98.3 |
| | | Tropo (Gg) | 419.1 | 415.2 | 437.2 | 433.6 | 413.0 | 418.6 |
| VSL_Br | | Surface (pptv) | 8.5 | 7.9 | 8.6 | 8.2 | 8.4 | 7.7 |
| | | Tropo (Gg) | 56.7 | 55.2 | 59.3 | 58.5 | 55.2 | 55.4 |
| VSL_I | | Surface (pptv) | 1.2 | 1.1 | 1.2 | 1.1 | 1.2 | 1.0 |
| | | Tropo (Gg) | 4.8 | 4.7 | 4.9 | 4.8 | 4.7 | 4.6 |
| **Inorganic Halogens** | | | | | | | | |
| Cl_y | | Surface (pptv) | 37.4 | 36.0 | 40.8 | 37.5 | 38.3 | 36.8 |
| | | Tropo (Gg) | 155.9 | 140.7 | 176.5 | 161.9 | 171.3 | 165.6 |
| Br_y | | Surface (pptv) | 2.2 | 3.3 | 2.7 | 2.7 | 3.5 | 4.1 |
| | | Tropo (Gg) | 19.8 | 19.8 | 22.7 | 20.9 | 20.5 | 22.1 |
| I_y | | Surface (pptv) | 3.1 | 2.7 | 3.0 | 2.7 | 3.0 | 2.9 |
| | | Tropo (Gg) | 17.3 | 16.8 | 15.3 | 15.5 | 16.0 | 15.7 |

Continue next page





1588

| Species Simulation | | FCnudged_slh | | FCHIST_slh | | FCnudged_slh_1x1 | | FCHIST_slh_1x1 | | FWnudged_slh | | BWHIST_slh | |
|---|---|---|---|---|---|---|---|---|---|---|---|---|---|
| | | Abs. | Perc. (%) | Abs. | Perc. (%) | Abs. | Perc. (%) | Abs. | Perc. (%) | Abs. | Perc. (%) | Abs. | Perc. (%) |
| Atmospheric Implications | | | | | | | | | | | | | |
| $\Delta O_3$ | Surface (ppbv) | -6.2 | -21.8 | -7.0 | -23.4 | -8.1 | -27.5 | -8.5 | -28.2 | -6.3 | -22.8 | -7.7 | 26.2 |
| | Tropo (Gg) | -58.9 | -18.0 | 59.6 | -18.3 | -69.6 | -21.3 | -73.0 | -22.1 | -57.2 | -17.7 | -64.0 | 19.6 |
| $\Delta OH$ | Surface (ppqv) | -1.9 | -3.0 | -3.1 | -4.6 | -3.4 | -5.4 | -3.7 | -5.7 | -2.4 | -3.7 | -3.8 | -5.7 |
| | Tropo (Mg) | -13.0 | -5.7 | 14.6 | -6.4 | -20.8 | -9.6 | -21.0 | -9.5 | -14.8 | -6.2 | -15.3 | -6.7 |
| $\Delta NO_2$ | Surface (pptv) | -0.7 | -0.1 | -3.7 | -0.6 | 1.0 | 0.2 | -0.2 | 0.0 | -1.8 | -0.3 | -5.2 | -0.9 |
| | Tropo (Gg) | -23.0 | -6.5 | 28.9 | -8.3 | -31.8 | -8.9 | -35.3 | -9.7 | -30.5 | -8.6 | -33.5 | -9.5 |

1589

1590



**4 Final Remarks**

In this work, we provide a comprehensive description of the technical implementation and operational configuration of a complete set of CESM compsets conceived to represent the baseline influence of SLH emissions and chemistry on the global atmosphere. The new CESM2-SLH version includes a detailed representation of the main organic halocarbon and natural inorganic sources, as well as the gas-phase and heterogeneous chemistry of chlorine, bromine and iodine in the marine boundary layer, free troposphere and lowermost stratosphere, highlighting the dominant processes determining the role played by SLH on atmospheric composition. In doing so, we evaluate their global budgets, chemical partitioning and spatio-temporal distributions, with a special focus on the regional and global implications of SLH on atmospheric oxidizing capacity and atmospheric chemistry. Despite during the past 15 years most of the SLH developments have already been reported using different CESM1 versions, there was still missing a complete description integrated in the context of the latest model developments made within the Community Earth System Model framework, i.e., CESM2.

The released CESM2-SLH version comprises six new model compsets and includes a complete set of configuration files and namelist updated that must be properly setup with user-defined namelist options. This include increasing the number of default emission sources and prescribed LBCs, as well as expanding the list of species that suffer dry- and wet-deposition, including the ice-uptake list. Similarly, a new set of initialization and absorption cross-section files including updated data for VSL halocarbon and inorganic halogen species are now available. Most notably, and with the intention of avoid unintended misevaluation of SLH influence on atmospheric composition for users familiarized with current CESM2 configurations typically used for climate and air quality studies on the global scale, this work provides a general but complete inter-comparison of organic VSL and inorganic halogen distributions obtained for the different model compsets. Given the large spatio-temporal variability of $Cl_y$, $Br_y$ and $I_y$ within different regions of the atmosphere, as well as its strong dependence with several model variables and parameters such as the abundance and distribution of sea-salt aerosols and other substrate surfaces where inorganic halogens recycle, we recommend new-users not familiarized with SLH chemistry to initially select the coarse (2°×2°) resolution, particularly with the *FCnudged_slh* compset. This ensures that dynamical transport is consistently represented across simulations, which is crucial for isolating and quantifying chemical changes between different model sensitivities. In case other model configuration is preferred, we strongly recommend performing an equivalent model intercomparison as the one provided here to ensure that the global halogen sources and budgets are within the ranges of the many model configurations compiled in this work.

Given the important role played by SLH in tropospheric and stratospheric chemitry, as well as their changing influence depending on many Earth's system components which in turn depends on background atmospheric composition and climate, we call the wider CESM community to consider the CESM2-SLH release to obtain a realistic representation of the background influence of natural and anthropogenic short-lived halogen sources and chemistry in air quality and Earth's climate studies. Similarly, we encourage other global chemistry-climate models and regional chemistry-transport models to implement equivalent representations of SLH sources and chemistry to improve their representation of the natural halogen baseline in the atmosphere.

**Author contributions**

A.S.-L. deigned research and lead the development of SLH chemistry in CESM. R.P.F. with the help of C.A.C., D.K. and F.V. implemented the porting of the code. R.P.F., C.A.C. and J.V. run the simulations. R.P.F., C.A.C., J.V., A.F., A.B., J.A.B., A.R., O.T. and Q.L. performed the data curation, processed the output and validated the model results. All co-authors contributed to investigation, discussion and visualization. R.P.F., C.A.C. and A.S.-L. wrote the original draft, with further review and editing from all co-authors.




**Acknowledgements**
This study has been funded by the European Research Council Executive Agency under the European Union´s Horizon 2020
Research and Innovation programme (Project "ERC-- 2016-- COG 726349 CLIMAHAL"), and supported by the Consejo
Superior de Investigaciones Científicas (CSIC) of Spain. R.P.F. would like to thank financial support from ANPCyT (PICT
2019-2187 & 2022-0474) and MinCyT (REMATE IF-2023-85161983-APN). The National Center for Atmospheric Research
(NCAR) is sponsored by NSF under grant number 1852977. Computing resources, support, and data storage are provided and
maintained by the Computational and Information System Laboratory from NCAR (https://doi.org/10.5065/D6RX99HX).
A.F. acknowledges support from Horizon Europe through a Marie Skłodowska-Curie Actions Postdoctoral Fellowship (grant
agreement 101103544, SUMAC). Q.L. is supported by National Natural Science Foundation of China (W2411028). We would
like to thanks to an endless list of scientists, postdocs. technical personnel and PhD students from many groups around the
world that contributed to the development and evaluation of SLH sources and chemistry in CESM1 and CESM2.

**Code and Data Availability**
The Community Earth System Model (CESM) code is maintained by the NSF National Center for Atmospheric Research
(NCAR). The benchmark version of CESM2 is distributed via GitHub (https://github.com/ESCOMP/CESM) and available at
the NCAR's official site (https://www.cesm.ucar.edu/models/cesm2). The complete CESM2-SLH branch used in this work
(*cesm2.2-asdbranch_slh*) can be downloaded from GitHub (https://github.com/RafaPedroFernandez/CESM/tree/cesm2.2-
asdbranch_slh), which is a fork version of the official CESM2 repository (https://github.com/ESCOMP/CESM/tree/cesm2.2-
asdbranch). NCAR recommends citing the CESM2 model as (Danabasoglu et al., 2020). All SLH code and data supporting
this work, including the complete set of building scripts, Fortran routines, individual namelist templates for *compset_slh* and
*compset_noh* configurations with the necessary input files, as well as the annual mean output used to generate all Figures and
Tables, can be obtained from Mendeley Datasets at https://doi.org/10.17632/f87hvrv25v.1 (Fernandez et al., 2025).

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
