# Peer review of "Short-Lived Halogen Sources and Chemistry in the Community Earth"

_EGUsphere, 2025_

## Referee Comment (RC2)

**Review egusphere-2025-3250**

**1 General comments**

`Overall quality of the preprint`

This preprint is clearly targeted towards the Community Earth System Model (CESM) community and presents the implementation of short-lived halogen (SLH) sources and very short-lived (VSL) halocarbons into CESM version 2.2.0. The authors demonstrate substantial expertise in this subject area.

The manuscript is organized as follows:

- Section 2 (2.1–2.4) provides a detailed account of historical developments related to SLH sources and VSL halocarbons, combined with model implementation details and scientific background.

- Section 2.5 describes shared community model configurations and setups for SLH and VSL experiments.

- Section 3 presents results from CESM-SLH experiments, comparing them to previous CESM1 work and observational benchmarks. These include global distributions, vertical species partitioning, and atmospheric burdens.

- A short summary concludes the manuscript.

This topic and the results presented are highly relevant to the SLH/VSL research community. However, the current presentation limits accessibility and clarity. Specifically, the manuscript is:

- insufficiently structured,

- repetitive, with numerous filler words and phrases (e.g. "we highlight/note that", "Here, we note/should be noted"),

- grammatically inconsistent (e.g., plural nouns combined with singular verbs),

- using lab-specific jargon or unexplained community terms, and

- affected by persistent typesetting issues, particularly with citations (e.g., parentheses instead of in-text citation).

**Equations, tables, and figures:**

- Several tables and headings do not follow the journal's formatting standards.

- Formulae are incorrectly typeset (see some examples in Section 3).

- Figures are sometimes low-resolution, with frame artifacts and inconsistent labeling.

- Colormaps draw disproportionate attention to light blue and yellow regions.

- Empty figure panels are included.

- Exponential expressions are written as `E-02` instead of $10^{-2}$.

Although submitted to *GMD*, the manuscript does not yet meet expectations for a clear and comprehensive model description. For instance, the term `compset` is never explained for readers outside the CESM community.

**Suggested Improvements**

To enhance clarity and accessibility, the following structural changes are recommended:

- Split and reorganize Sections 2 and 3.

- Following common conventions, Section 4 should be titled "Summary and Conclusions."

- Introduce an additional section dedicated to discussion, consolidating currently fragmented discussion points.

**More detailed structural advise:**

- The new *Section 2* (Model description) should begin with a concise, general overview of *CESM2* and its relevant components. For example, explain:
  - CESM structure (refer to Supplement Chart S1),
  - definition of specified dynamics (SD),
  - what a compset is,
  - differences between compsets (e.g., SD vs. free-running).

- Only after this overview should CESM2 be contrasted with CESM1.

- Follow with an overview of CESM2-SLH developments and available compsets (currently Section 2.5). Detailed implementation should then be presented in the next section 3.

- Section 2.1 (historical developments), while interesting, is unnecessary for this manuscript and could be removed, as this is not intended as a review paper.

- The new *Section 3* (Model implementation of SLH and VSL) should combine Sections 2.2–2.4, focusing on implemented schemes, mechanisms, and chemistry.

- Consider including Figure S1 here, as it illustrates key differences in sea salt properties between CESM1 and CESM2.

- Avoid historical context and developments in this section. It is enough to briefly introduce the Ordoñez inventory.

- Move "user guide" like paragraphs into an Appendix section (User guide) or rephrase them for a more general audience.

- The new *Section 4* (Results) could be subdivided into:
  - 4.1 Validation (currently Sections 3.2.2 and 3.3.1),
  - 4.2 Comparison to CESM1 results,
  - 4.3 Sensitivity studies with CESM2-SLH and additional single-experiment studies.

- First describe sensitivity experiments and summarize them in tabular form (in parts Tab. 6).

- Consolidate all discussion into a dedicated *Section 5* (Discussion).

- Rename the final section as *Section 6* (Summary and Conclusions).

Implementing these changes will significantly improve readability, accessibility, and compliance with journal standards.

The title may be expanded along the lines of "– implementations and benchmark results".

**2 Specific comments**

Individual scientific questions/issues

**2.1 Section 2**

- L259–263: *"Despite [...]"* It seems there is no variable in Eq. (1) that connects emissions to the long- or shortwave incoming solar irradiance of the atmospheric basemodel which would vary with atmospheric state, season, time of the day, and orbit. Or is this hidden behind our "diurnal emission profiles"? From the text, it is not clear how these "profiles" are implemented.

- L261–263: *"Emissions follow either [...]"* What does this mean? Can the user choose between the two?

- L263–264: *"Given that no long-term trend has been established [...]"* Has a trend been *observed* or not? Please include a recent reference (post 2007 and/or 2023) that could support your statement.

- L264–268: This paragraph is an example of "user guide" like paragraphs. As indicated above, please collect this and similar paragraphs in a dedicated Appendix section. I will not point out each of them individually.

- L286–287: *Based on these works, [...]* With respect to the preceding sentence, it is unclear whether this imposed trend is part of the inventories or an alternation in present work. If latter is the case this imposed trend needs more explanation and references. If former is the case this sentence may be dropped as it doesn't provide much additional information.

- L311: *"routine [...]* iodine_emissions.F90 [...]" Here you refer to a Fortran (module) file as routine. Please be more precise and distinguish between subroutine and module file. What is the actual name of the subroutine? Please also refer my rephrasing suggestion in Section 3.

- L325–326: *"Note that the fitting [...].* This sentence is not well written and should be rephrased along the lines: → 'Equation (2a,b) are only valid for wind speeds $w \geq 3\,\mathrm{m\,s^{-1}}$ (Inamdar et al., 2020). Below this threshold, anonymously high iodine fluxes are predicted.'

- L326–328: *"Therefore, a wind speed mask imposing [...] periods."* This paragraph is not entirely clear from an implementation point of view in the code. Rephrase: → To avoid this overprediction, $w = 3\,\mathrm{m\,s^{-1}}$ is assumed for all grid cells where $w < 3\,\mathrm{m\,s^{-1}}$.

- L350: *"[...] above a pressure threshold [...]"* Does this mean below or above $300\,\mathrm{hPa}$?

- L367–373: *"Given, the large changes [...] (see Section 3.2.2)."* Be more explicit. How large are the uncertainties and how much have you adjusted the reaction rates compared to the most recent JPL catalog?

- L379–382: *"Careful consideration [...] budget and burden."* This content of this paragraph is of more general nature and should be formulated that way. → 'The representation of aerosols in general circulation models, their resulting distribution, abundance, and properties play a key role for prediction of SLH production via SSA dehalogenation in the troposphere.'

- L384–385: *"[...] the vertical extent of sea-salt in the free-troposphere was largely reduced [...]"* Be more explicit. What does this mean? How was this achieved?

- L478–482: *"CESM-SLH uses the Modal Aerosol Model with 4 Modes [...]"* This has been implicitly mentioned earlier in that section. Either include this, when you first mention the aerosol model difference between CAM4-chem and CAM6, or in the general CESM model description section as proposed.

- L528–530: *"Regarding stratospheric heterogeneous [...]"* This sentence seems to refer to the stratospheric heterogeneous chemistry in the context of Polar Stratospheric Clouds (PSCs) that had been previously implemented in `mo_strato_rates.F90`. But it is unclear what "complemented [...] mapping" means. The expression "historical implementation" here seems inadequate. This sentence should be rephrased.

- L536–539: *"In doing so, [...]."* This sentence is unclear and should be rephrased. Do you mean that you used reaction yields measured for chlorine and bromine species for the respective iodine species instead of the actually observed iodine yields rates? What is the reason for this?

- L604–608: Putting on my code developer hat... Considering that the VSL/SLH developments in CESM1 were not directly transferable to CESM2, how sustainable are your developments based on CESM2? First, `forking` implies that the development may never be merged back into the CESM repository and is in jeopardy to end up on a stale branch (again). Second, taking a look at your github repository shows that the last pull request from the upstream repository has been more than one year ago. Third, since submitting this preprint you have added changes to the branches on your CESM2 and CAMS forks without (as far as I can tell) tagging the versions you have based this preprint on. Therefore, it would not be possible to reproduce your results, although your repository is publicly available (which is actually great!).

- L625–631: *"[...] factors are maintained identical to the default FCHIST and FCnudged parent's configuration [...]"* It remains unclear how the two compsets differ (without going through additional resources). I suggest elaborating on the difference between the two in the beginning of the results section.

- L643–645: *"Finally, a fully-coupled [...] setups, both of them at [...] resolution [...] were mapped to their corresponding WACCAM compsets."* This sentence is grammatically not correct, hard to understand, and needs rephrasing. What do you mean by "mapped"? Do you mean that you have created also compsets for WACCAM based on your *BWHIST_slh* and *FWnudged_slh* compsets that use CAM6?

- L664 and following: *"stabilization"* Expression is unusual. Usually, this is referred to as 'equilibrium'.

**2.2 Section 3**

- L766–769: *"[...] their photodecomposition constitutes the critical first step in releasing inorganic chlorine and bromine [...]"* This is a repetition of scientific background presented in Section 2. Here, and elsewhere in this section, you should focus on the results and not on repeating scientific background.

- L898: *"[...] in agreement with previous estimates [...]"* You may cite the numbers from the reference.

- L900–901: *"model surface"* What "surface" does this refer to? Is it the actual land/water surface or the lowermost model level above the surface?

- L904–907: *"This is explained by the large non-linear response of the [...]"* Can you further elaborate on this? Where do the non-linearities in Eq. (3) come from? I understand that there are non-linearities in the involved atmospheric feedback processes, e.g. radiation-temperature-photolysis.

- L952: *"[...] the $IO_x$ burden change [...] remains close to 10 %."* It is not clear, to what these 10 % refer to. In Table 8 no "burden change" is found, rather the tropospheric burdens in Gg are displayed. The percentage difference between the two SLH experiments and the given burden of $3.2\,Gg\,IO_x$ amounts to 9.4 % and 12.5 %, respectively. Please clarify and rephrase accordingly.

- L1023–1038: Given the varying height of the tropopause between $90°\,N - 90°S$ ($\approx 400 - 90\,hPa$) and that the ratio between ocean to land is roughly 2 to 3, I don't quite understand this paragraph. First, a global average over pressure/height lumps together polar stratospheric and topical tropospheric air. Therefore, you absolutely cannot draw any conclusion with respect to the UTLS region. Second,

you refer to the Marine Boundary Layer (MBL) but average globally including also grid cells over land, I suppose. Therefore, it seems impossible to draw conclusions with respect to the MBL. You would need to convert your data into vertical coordinates relative to the tropopause height and use a land-sea-mask to include ocean grid cells only to be able to draw robust conclusions.

- L1051–1082: The concerns raised above also apply to these paragraphs and analyses.

- L1132–1139: *"Please note that due to [...]"* This paragraph belongs to a discussion section.

- L1116–1165: *"Here we recall [...]"* Beside this odd expression, this paragraph belongs into a discussion section.

- L1165–1167: *"[...] should not represent a significant bias in our model implications on other atmospheric components."* Quite unclear. Do you mean that atmospheric (stratospheric) chemistry would not be affected (or only to a small degree) by the underestimation of SGI$_I$?

- L1177–1182: *"[...] over the boundary layer (i.e., mean modelled value from the surface up to $\sim$ 850 hPa)."* This should be rephrased, as the boundary layer is highly variable. $\rightarrow$ 'average over model levels $P \geq 850$ hPa and the time period $2015 - 2020$'.

- L1278–1284: *"Given [...] in the troposphere [...]."* This paragraph could be placed in a discussion section.

- L1434–1438: *"the old nudging over model levels [...] most recent nudging approach [...] the chemical influence of SLH on tropospheric and stratospheric transport are of equivalent magnitude."* The first part is basically a repetition from the model description section. Albeit, the formulations differ. Please rephrase in more general terms. Regarding the second part, you have previously discussed in this preprint, that dynamics are important for the amount of VSL ans SLH in the troposphere and stratosphere and therefore for the ozone depletion potential and ozone abundance. Please clarify, how the model results presented in this section could indicate an impact of *SLH chemistry* on atmospheric transport, especially in a SD or nudged setup where dynamics are prescribed.

- L1517–1550: Large parts of this paragraph are written as if they were meant for a *user guide* which should clearly not be the purpose of a scientific paper. Please rephrase with a more general modeling community audience in mind.

**2.3 Section 4**

For your *Summary and conclusions* section, it would also be important to mention the necessary scaling of emissions depending on compset and model resolution. Potentially, the scaling factors for the nudged case will depend on the nudging datasets (e.g. ERA5, MERRA, CRUJRA, ...) as well. As part of this section, but also in the model development section, you should clearly state, how these scaling factors should be derived.

**3 Technical corrections**

```
Purely technical corrections
```

Due to the large amount of repeated grammar and expression issues, I will not be able to give a full account. I rather list some examples with general suggestions on how to improve the style.

**3.1 Section 1**

- L82: *"X, Y = Cl, Br, I"* Maybe write this as mathematical expression? $\rightarrow X, Y \in \{\mathrm{Cl, Br, I}\}$.

- L99–118: Reactants in chemical formulae are supposed to be set upright, e.g. using \chem in LaTeX. $\rightarrow$ https://www.geoscientific-model-development.net/submission.html#math

**3.2 Section 2**

- L151: *"(e.g., (Saiz-Lopez et al., 2023)"* Missing ')'. → e.g. using LaTeXcommand `citep[e.g.][]<sl23>`, where `<sl23>` is the reference label ⇒ (e.g., Saiz-Lopez et al., 2023).

- L239 (and many others): *"[...] described in the original work of (Ordoñez et al., 2012)."* When citing a reference *in-text*, in-text citation should be used, e.g. in LaTeX `\citet{}` ⇒ Ordoñez et al. (2012).

- L239: *"[...] has been extensively described in the original work of [...]"* This expression is odd. → 'has been described in'.

- L248: *"20°-50°"* em-dash should be used according to journal house-style. Please check similar expressions! → $20° − 50°$.

- L252: Formula not set correctly. Please refer to the journal house-style, e.g. non-counting indices should be typeset upright. →https://www.geoscientific-model-development.net/submission.html#math

- L256: *"Note that, differing from previous [...]"* As indicated above, historical context is unnecessary and can be omitted.

- L259–263: *"Despite [...]"* This sentence is hard to understand like most of your sentences that start with "despite". Try to rephrase this and similar sentences! → 'The emission flux is predefined and read from file to which we referred to as offline emissions. The FORTRAN routine `mo_srf_emissions.F90`' was modified to compute a time-dependent halocarbon emission in hourly resolution accounting for the radiation dependency of photosynthesis. Emissions follow either [...]'

- L269–281: *"Finally, [...]."* Please refrain from giving historical context, rather concentrate on the important content! This paragraph should be rewritten along the lines: → 'Due to improvements in MBL dynamics and resulting different oxidative capacity, the Oroñez emission inventory for brominated and chlorinated species has to be rescaled by a factor of 1.15 (15 %) to achieve the observed stratospheric halogen loading.'

- L284–286: *"For the particular case of chlorine, [...]"* In-text citation style instead of parenthesis needed. Rephrasing of this sentence is recommended as it is hard to understand. → 'For anthropogenic SLH sources of chlorine, we follow the Hossaini et al. (2019) and Claxton et al. (2020) emission inventories. These inventories include time-dependent emission of two major contributors to the organic chlorine load, $CH_2Cl_2$ and $C_2Cl_4$.'

- L289–291: *"Given that the dominant VSL chlorine sources [...]"* This sentence is difficult to understand. Rephrase along the lines: → 'Beside significant emissions of $CHCl_3$ from natural sources, chlorinated VSL are dominated by anthropogenic emissions that display a pronounced hemispheric asymmetry.'

- L293: *"[...] globally scaled [...] globally."* Doubling. Remove one "globally".

- L297–298: *"[...] but not yet implemented [...]"* You may remove the part of the sentence as it is not important whether it has not been implemented in any CESM version.

- L298–301: *"To include [...]"* This would be part of the proposed "user guide" Appendix section.

- L308: *"[...] variables HOI and $I_2$ [...]"* HOI and $I_2$ are not "variables" but 'trace gases'. Please phrase accordingly.

- L307–310: *"To include [...] and replaced by the online computation."* This would also be part of the "user guide".

- L315–317: Formulae not set correctly, e.g. $F_{I2} \rightarrow F_{I_2}$; $ln(w) \rightarrow \ln(w)$; $1.75 \times 10^9 \rightarrow 1.75 \cdot 10^9$; →https://www.geoscientific-model-development.net/submission.html#math

- L310–313: *"The implementation of this source [...]. The online emission fluxes [...]."*. Rephrase. → 'Following Prados-Roman et al. (2015)b; MacDonald et al. (2014) the online emission fluxes of [...] are included in the Fortran module file `iodine_emissions.F90` and computed in subroutine **???** with Eq. [...].'

- L326,327: m/s → $\mathrm{m\,s^{-1}}$

- L328–329: *"Finally, we highlight [...] with ozone abundance [...]"* This sentence seems to be grammatically not correct and is one example for the extensive use of filler phrases mentioned above. *"Finally, we highlight"* should be removed and the sentence (and other sentences of similar kind, too) rephrased. → 'The first order dependency on ozone abundance of $F_{\mathrm{HOI}}$ and $F_{\mathrm{I_2}}$ has important [...].'

- L338: *"[...] following Eq. (3)"* Might be removed as the referred equation is following right after.

- L345: *"[...] (R is the universal [...])."* Rephrase and resolve the brackets. → '[...], with $R$ the universal [...].'

- L346: *3 (three)* Not clear. According to house standard this should read 'three'.

- L382–385: *"This is indeed the case, [...]"* This sentence should be rephrased and split to make it clearer. → 'The major difference for aerosols between CESM1 (CAM4-chem) and CESM2 (CAM6) is the change from a continuous bulk representation with four size bins to a modal representation with three bins.'

- L397: Wrong arrow. '→'

- L400–403: *"The CESM-SLH implementation [...]"* This sentence should be split after *"Eq. (3)"*. The second part is difficult to understand and should be rephrased along the lines: → 'The emission of chlorine from the bulk sea salt aerosols is driven by the heterogeneous reaction of oxidized nitrogen species that do not contain halogens.'

- L412–413: *"[...] (e.g. [...]"*. Missing parenthesis. It might be better to connect the half-sentence by comma. → '[...], e.g., [...].'

- L427–432: The 'checksum' of reaction types seems to be unnecessary. You may consider constructing a matrix/table of the following form:

| Family | ph | odd | org | sh | het | ss | strat | total |
|---|---|---|---|---|---|---|---|---|
| Chlorine | 11 | 6 | 20 | 2 | 7 | 11 | 10 | 62 |
| Bromine | 10 | 9 | 8 | 1 | 6 | 6 | 13 | 53 |
| Iodine | 17 | 39 | 2 | 1 | 10 | 6 | 18 | 93 |
| Total | 31 | 46 | 24 | 4 | 17 | 16 | 25 | 163 |

- L442–443: *"JPL [...] IUPAC"* Acronyms have not been previously defined.

- L452–456: *"[...], some laboratory measurements show [...]. [...] recommends neglecting the photochemical breakdown [...]. Within CESM-SLH [...] forcing to zero [...] was required [...]."* These sentences are unclear and overcomplicated. Refrain from using a term like *"forcing to zero"*. You are basically trying to say: → 'We are following the JPL recommendation, although some laboratory measurements suggest [...].'

- L453: *"JPL 19-5 handbook"* inconsistent with L442.

- L456–457: *"Not doing so [...]."* This sentence is somewhat redundant. References are missing. The sentence could be removed or should be rephrased if kept: → 'Including photolysis rates above [...] resulted in a complete decomposition of $C_2Cl_4$ down to the surface which is not in line with observations.'

- L475–477: *"Here, we highlight [...]"* Rephrase and avoid phrases like *"Here, we highlight"*. Furthermore, it is not clear which CESM version this sentence refers to.

- L477–478: *"Therefore [...]."* Missing comma after *"Therefore"*. Overall, this sentence seems to be in disorder grammatically. Please rephrase.

- L491: $rate_{ICE}$: Variable typesetting not in accordance to house style. $\rightarrow$ $rate_{ice}$

- L491–493: *"[...] assumed to be uptake [...]"* This sentence is grammatically incorrect. Please correct.

- L533: *"[...] a logical condition of humidity [...]"* This phrase is unclear. What does this mean? Please rephrase and make clearer.

- L540: *"FC"* is not defined and should be explained.

- L575: *"Henry law coefficient"* inconsistent naming. Called *"constants"* in L576.

- L630: *"maintained"* $\rightarrow$ 'kept'.

- L634–635: *Despite none of them have been used in this work, we note [...]* Odd expression. The quoted part of the sentence may be removed.

- L639: *"[...] many [...] depends [...]"* $\rightarrow$ 'depend'.

- L640–642: *"Similarly, we highlight that [...] significantly impact on [...]"* Filler-phrase and grammar. You may remove the filler-phrase. The preposition *"on"* doesn't belong to *"impact"* and should be removed.

- L642–643: *"[...] shifting [...]"* The expression might not to be correct. $\rightarrow$ 'changing'.

- L705: *"rationale"* This expression may not be correct in this context. $\rightarrow$ 'reason'.

- L719: *"(the so called AANE sensitivity in (Saiz-Lopez et al., 2023)"* Missing parentheses and formatting of the citation. Acronym AANE is not resolved. The insertion should be elaborated on and written out.

**3.3 Section3**

- L738–740: *"Results presented [...] are based on [...] sensitivities [...]."* Sentence contains lab-slang. *"sensitivities"* $\rightarrow$ 'sensitivity studies' or 'sensitivity experiments'. Please change all occurrence accordingly!

- L747–748: *"Finally [...] model inter-comparison between the different CESM2-SLH resolutions and configurations."* Usually the term *"model inter-comparison"* is reserved for the Coupled Model Intercomparison Project (CMIP) and derivatives – organized comparisons between different models in the Earth System and Climate modeling community with a clear modeling protocol. *"model inter-comparison"* $\rightarrow$ 'comparison' or 'sensitivity experiments' or 'sensitivity studies'

- L754: *"Therefore, we highlight that [...]"* As mentioned previously, these filler-phrases should be removed throughout the text to improve readability.

- L780–782: *"Here, we highlight [...]"* This sentence should be split after *"iodine"*.

- L782: *"on-line"* $\rightarrow$ 'online'

- L809: *"allow"* $\rightarrow$ 'allow for'

- L809–810: *"to provide [...] to show"* Correcting the above, these infinitive forms become unnecessary.

- L816: *"[...] presents a larger increment than in CESM1 [...]"* This sentence is unclear. Do you mean → 'Iodine volume mixing ratios continue to increase after year 2000 in the CESM2 experiments, while they flatten out in the CESM1 results'? This could be construction of your CESM1 experiments, e.g. recycling of SSTs and SICs after the end of the hindcast period...

- L864–868: *"Here we note that [...]"* Remove filler-phrase and split the sentence before "highlighting".

- L870–873: *"In addition, [...] in any case [...] in any case"* Repetitive expression. At least on "in any case" should be removed.

- L870–873: *"[...] in the lower edge [...]"* grammatically not correct. → 'at the lower margin'

- L874: *"timeline simulation"* This expression is odd. Do you mean 'transient simulation' or 'hindcast'?

- L877–880: *"The larger tropospheric [...]"* This sentence is grammatically not sound and should be rephrased. How is present-day conditions defined? → 'The larger present-day tropospheric $I_y$ burden in *FCHIST_slh* and *FCnudged_slh* ([...]) compared to estimates by Saiz-Lopez et al. (2023) and Barrera et al. (2023) ([...]) are due to substantial changes [...]'

- L885–886: *"Table 8 [...]"* This is a rather unusual way to start a section. You should rather start with a sentence of introduction.

- L887–888: *"Equivalent results highlighting [...]"* This sentence is unclear and should be rephrased or, as it references a later section, it may be removed.

- L890–893: *"Here, we recall [...]"* Odd style AND repetition. Should be removed.

- L893–896: *"Thus, VSL iodine mean global emissions are the same between [...]"* This half-sentence is grammatically not sound and difficult to understand. It should be rephrased. → 'Global mean iodine emissions are (by definition) the same in CESM1 and CESM2, while [...]'

- L902: *"online sources"* → 'online emissions'

- L919: *"[...] inspection of Table 8 shows [...]"* This expression is odd. → 'We show in Table 8 [...]' or 'In Table 8 it is shown [...]'

- L950: *"[...] remains similar compared to [...]"* This is another odd expression and should be rephrased. Please also include some numbers from the cited publication. → 'remains comparable with'

- L955: *"Fig. 3 bottom row"* Here and elsewhere, please reference the subplots with their respective subcaptions.

- L958: Wrong arrow in the in-line formula. → '→'

- L968: *"Here it is evident [...]"* Yet another filler-phrase that can be safely removed.

- L984: *"[...] the difference between both configurations [...]"* "Configurations" is not the right term. As indicated above, you should clearly distinguish between setups/configurations/compsets and model experiments → 'experiments'.

- L986–987: *"[...] from organic SGs to inorganic halogen (PGs) [...]"* The brackets around PGs seem to be misplaced.

- L989–990: *"[...] (see for example Table A-1 in the Annex of (WMO, 2022) and elsewhere) [...]".* In-text citation needed. You should refrain from using "elsewhere", either you would cite other sources or not. The sentence is too complicated and should be split here, removing the parentheses.

- L1004–1005: *"[...] which is driven by [...] (see Fig. 7)."* Either the order of figures is wrong here and you are actually referring to Figure 6 or the order of the text does not match the figure order.

- L1010: *"throughout tropical injection"* → 'through the tropical tropopause region'

- L1086–1089: *"In doing so, [...]"* This is a repetition of model description and development and does not belong here.

- L1089–1090: *"For this exercise, [...]"* This expression is odd in the context of scientific paper. Furthermore, you should give references to the indicated "standardized evaluations against observations and international assessments" and describe them briefly.

- L1086–1090: Given the comments above, you could also remove the whole paragraph and only keep the first sentence.

- L1099: *"configurations"* Do you mean 'model experiments'?

- L1105: *"[...] the more extreme contributions [...]"* Unclear expression. What do you mean by "extreme contributions"?

- L1120: *"[...] modelled range indicate the variability between compsets."* Grammatically not correct. "indicate" → 'indicates'. Furthermore, you may need to rephrase this sentence as it is not quite clear. → 'The results show that the modelled SG to PG partitioning and total abundance of VSL in the stratosphere depend on the chosen compset.' You may discuss the reason (difference between the compsets) in a dedicated discussion section.

- L1130: *"Differing [...]"* → 'Unlike'

- L1165: *"Having said this, we acknowledge [...]"* This expression is misplaced in the context of a scientific paper.

- L1169: *"online sources"* This seems to be lab-slang. → 'online emissions' or 'emission computed during runtime of the model'.

- L1183–1204: This whole paragraph should be revised. Many sentences are grammatically incorrect or use odd expressions: "Several distinct features are observed", "This species presents", "Indeed, despite maximum $Br_y$ peaks in the NH are up to".

- L1276: *"significant"* If significance is not strictly proven the term should not be used. → 'strong' or 'substantial'.

- L1293: *"Regardless of the compset [...]"* This phrase is not correct and should be removed, because the results for the CESM1 experiments do not fall within the given bounds of $-(2.7 - 3.9)\,\%$.

- L1301: *"[...] is encouraged for future studies."* Odd expression. Rephrase. → 'should be subject to future studies.'

- L1303–1304: *"[...] (top) [...] (middle) [...] (bottom row)."* Please reference the subfigures by their labels. → '(a–c) [...] (d–f) [...] (g–i)'

- L1306–1308: *"[...], although due to the high [...]."* "although" can be removed.

- L1327: *"[...] is similar to the analogous pattern [...]."* It is not clear what "analogous" refers to in this context. It can be removed.

- L1333: *"Indeed, and despite [...]"* Filler-phrase. Can be removed.

- L1348: *"60N-60S"* Wrong formatted geographic range. → '60° N − 60° S'

- L1353–1354: *"[...], where the model [...] fails to capture their seasonal cycle."* "their" probably shall refer to "stations 11 and 12". The stations do not have a "seasonal cycle", but the ozone observed at the stations does. "their" → 'the'

- L1356: *"[...] impact in the [...]"* Grammatically not sound. → 'impacts the'

- L1357–1358: *"[...] predicted by a [...] pixel size in the model."* Odd expression. A *pixel size* cannot predict anything. → 'predicted by the model in a [...] resolution.'

- L1358–1360: *"[...] demonstrate that the inclusion of SLH chemistry in the model helps to close the gap with observations"* Preposition *"with"* is not grammatically correct. → 'on' or 'between model and observations'

- L1399: *"setup"* Please clearly distinguish between 'model experiments' (in this section) and 'model configurations' in the model description section. Consistently use either 'configuration' or 'compset' or 'setup'. It is not clear if those are the same.

- L1421: *"Globally, of the total halogen [...]"* This sentence is grammatically not correct and difficult to understand. Please rephrase.

- L1513: *"on-line"* Inconsistency. → 'online'.

- L1568: *"1 × 1"* → $1° \times 1°$

- L1570: *"This is associated with to the [...]"* → Remove "to".

**3.4 Section 4**

This section accumulates many of the repeated issues with expressions and lab-slang. Please rephrase in accordance to all suggestions made. I refrain from giving a detailed account.

**3.5 Tables and figures**

**3.5.1 Tables**

- Table 1

  - Chemical reactions are not correctly set, e.g. $BrONO_2$ -> $0.65 * Br_2 + 0.35 * BrCl$
    $\Rightarrow BrONO_2 \rightarrow 0.65 \cdot Br_2 + 0.35 \cdot BrCl$

  - Exponential expressions are not correctly set.

  - Empty lines and subheadlines should be removed. The information contained in the subheadlines should be part of the table caption. → 'Table 1: [...]. Reactions het_ss_0...8 include heterogenated-reservoir species and het_ss_9...12 oxidized nitrogen-compounds.'

- Table 2: same as Table 1.

- Table 3: same as Table 1. What does "mapped to X" mean?

- Table 4: same as Table 1. There us a lot of unnecessary whitespace in one column. This could be optimized.

- Table 5: same as Table 4.

- Table 6: Mixes the overview over available compsets with performed model experiments. It should be split. The part containing a summary of the available *SLH-compsets* should remain here. A Table summarizing ALL model *experiments* (also CESM1) should be place in the next section (Model experiments and results). You should clearly distinguish between compsets and model experiments. It is rather unfortunate that model experiments at different horizontal resolutions are listed under the same name. This ambiguity has to be resolved. Each experiment should have a unique identifier. "Suggested" and "available" do not make sense in the table. The recommendation should be part of the table caption.

- Table 7: Empty rows should be removed. The subtitles "SSA-Sources" and "FRA-Sinks" should be moved into the Table caption.

- Table 8: Table is oddly formatted. Horizontal lines appear broken and not all columns are labeled correctly, e.g. "species" seems to refer to 'species families'. The caption is not correct because the table displays different emission fluxes, volume mixing ratios, sinks, and atmospheric abundances. The rows "Abundance–Long-lived halogens" have no unit assigned. You may consider splitting the table into three or four parts which would make it also easier to reference in the text.

- Table 9: As mentioned earlier: "sensitivities" → 'sensitivity experiments'. Here and in other parts of the preprint, it remains unclear whether the CESM1 results refer to previously published results, which would need explicit citations, or new results based on old/repeated model experiments.

- Table 10: Wrong font. Empty lines should be removed. *All tropospheric magnitudes [...]* This text should be part of the table caption. What are "tropospheric magnitudes"? What is a "minimum lapse rate pressure"? Do you mean the pressure at which the lapse rate minimum is reached? A dot is missing at the end of the caption.

- Table 11: A dot is missing at the end of the caption. The line below the table should be part of the caption. Wrong font.

- Table 12: This looks like two pictures rather than a continued sideways table with a lot of artifacts ("continue next page", "page", cut in half row "Simulation ..."). Horizontal lines are broken.

**3.5.2 Figures**

- Figure 1: *"NH = 20°N(S) - 90°N(S)"* Do not apply to the journal style guide (ranges and em-dash).

- Figure 2: As indicated above. The CESM1 experiments are not well characterized in this preprint and difficult to set in context. The arrangement of this figure is a bit unfortunate, putting VOC volume mixing radiations to a (random) spot in second row together with species ratios. It might be possible to split this figure into two (7:5 panels each).

- Figure 3: The caption is too long and should not repeat the figure legend. Time ranges are not in accordance to journal style. Yellow, open markers are difficult to see. Another color could improve this. *compset_non* was not defined previously. Do you mean 'compset_noh'?

- Figure 4:
  - The resolution of this figure is rather low. It should be improved.
  - Panel (d) and (f) are misaligned compared to panel (a) and (c).
  - Panel (c) does not contain any data as stated in the figure caption ("empty panel c") and should be removed.
  - The colormap is not colorblind-friendly and draws the attention towards light blue and yellow. It should be replaced. See also `https://www.geoscientific-model-development.net/submission.html#figurestables`.
  - The uppermost tick and every second tick on the pressure-axis are unlabeled. With respect to the space restrictions on the pressure-axis, ticks should be labeled where possible, else removed.
  - Subcaptions (a-f) are placed inside the axis frame, which is in contrast to the previous figures. You may consider placing them outside.
  - The brackets around units should be round like in the previous figures.
  - The colorbar labels indicate that *white* refers to values below zero. From the arrangement of the figure, I guess, *white* is supposed to indicate *invalid* values in panel (c). Usually this would be marked with a ◁, e.g., `extent='left'` in `python, matplotlib` or *gray* color. Removing panel (c) will probably resolve this issue.

- The *zonal mean tropopause* (not "mean model tropopause" as stated in the figure caption) displays a strange gap at around $30\,°S$. You may consider plotting the *zonal mean tropopause* as a line not with markers.

- Figure 5: In general, the same as Figure 4 applies. In addition:
  - Panel (f) does not contain any data and should be removed.
  - Panel (i) is identical to panel (c) as it is supposed to show the difference between (c) and (f). It should be removed.
  - It might be better to use a sequential colormap for panels (g,h) that would highlight that the difference between the two experiments is shown here and that species concentrations in FCnudged_slh are always higher.
  - In figure titles, *FCnudged_NON* is displayed. Did you mean *FCnudged_noh*?

- Figure 6:
  - Captions should not replace figure legends. You may include another legend entry displaying the assignment of each experiment to the open or filled markers.
  - The subcaptions (a–c) are inconsistent with previously used placements and style.
  - It is not clear, what a "mean height of the global tropopause" is. Do you mean 'global mean tropopause height'?

- Figure 7: The line colors used are difficult to distinguish. The legends are overlapping with the data. The x-axis labels of panels (a,c,e) are not correct, as they display a *flux*. The flux units are rather unusual. Usual flux units are $kg\,s^{-1}$ or $molec\,s^{-1}$ is used. You may fix the units. The information contained in this figure may be supplementary material.

- Figure 8: Probably the most important figure of this manuscript. Although, it can also be improved.
  - Observations are missing in the figure legend.
  - "vmr" → 'VMR' or use $X_{Cl,Br,I}$. $X$ is preferred by the measurement community.
  - Volume mixing ratio (VMR) is not defined in the manuscript and referred to as "mole fractions" in the figure caption which is inconsistent.
  - The expression "observed mean ± range mole fractions" is odd. Do you mean 'Average and range of halogen volume mixing ratios (VMR) compiled from observations in the tropical tropopause region'?
  - $\Delta X_i$ is not mentioned in the legend. It is called PGs there. Hence, legend and caption are inconsistent.
  - "mean ± range (computed as the standard deviation [...]) is shown by [...] coloured shading [...]". The "coloured shading" is usually referred to as 'error bands'. Why do you call the standard deviation "range"?
  - The caption is too long, difficult to understand, and should not replace or repeat the main text.
  - The figure caption could read: 'Vertical profiles of the tropical ($20°N - 20°S$) zonal mean, averaged over the time period $2015 - 2020$ for experiments [...] compared to observations for (a) $X_{Cl}$, (b) $X_{Br}$, and (c) $X_{I}$.

- Figure 9:
  - The caption is too long and should be rephrased.
  - Time axis label may be added.
  - Units should be placed in round brackets.
  - Shouldn't the tropopause height show a climate trend over this period?

- Figure 10:
  - The colormap is not colorblind friendly and highlight the regions early named.
  - The colorbars are not labeled.
  - Subplot captions are placed inside the axis frames and hardly visible.
  - Colorbar ticks in subplot (g) should be formatted in exponential form ($\times 10^{-4}$)

- Figure 11: The hatches are almost invisible. You may choose different hues or shades of red and blue instead. "Empty" is an odd expression for unhatched bars.

- Figure 12: Some units are not placed in parentheses. Labels are hardly readable due to the low resolution picture. You may consider plotting the zonal mean tropopause height as line not with markers. The color range of subfigure (c,f) is saturated in large parts of the Northern Hemisphere. Upper and lower bounds should be adapted.

- Figure 13: Same as Figure 12.

- Figure 14: Albeit the model experiments are only evaluated for one year (2015) a monthly mean should be accompanied by a standard deviation or standard error. It remains unclear what the error bars in the TOAR observations are. If the TOAR data represent climatologies (which years?), aka multi-annual means for each month, then these would represent the inter-annual variability. This should be clarified here and in the main text. Usually the annual averages displayed here would be separated from the monthly means by a vertical line. Is "NON" in the legend the same as model experiment *FCnudged_noh*? Should it read 'NOH'?

- Figure 15: Axis labels and tick labels are too small. The colormap is drawing the attention to the yellow region and hence highlights features that might not be there. A different (sequential) colormap should be chosen. The range of the colormap is not the same for subfigures (a–c) and (d–f).

- Figure 16: Yet another variant of units not placed on round brackets. Labels and tick labels are too small. Line colors are hardly visible or distinguishable. For subfigure (a–c), the same as for previous global averages applies – polar stratosphere and tropical troposphere are lumped together. Global averages should be made in vertical coordinate system relative to the tropopause.

- Figure 17: "°lat → Latitude (°)"

- Figure 18: Same as for similar plots.

- Figure 19: (a–c), same as for similar plots regarding the vertical coordinates and globally averaged tropopause and boundary layer.

- Figure 20: Same as for similar plots: colormap and missing colorbar labeling.

**References**

[1] Burkholder, J. B., Sander, S. P., Abbatt, J., Barker, J. R., Cappa, C., Crounse, J. D., Dibble, T. S., Huie, R. E., Kolb, C. E., and Kurylo, M. J. (2020). Chemical Kinetics and Photochemical Data for Use in Atmospheric Studies, Evaluation No. 19 (JPL Publication 19-5). Jet Propulsion Laboratory, https://www.google.com/url?sa=t&source=web&rct=j&opi=89978449&url=https://jpldataeval.jpl.nasa.gov/pdf/NASA-JPL%2520Evaluation%252019-5.pdf&ved=2ahUKEwisr4TPsLCRAxV-BdsEHXGNNWYQFnoECBAQAQ&usg=AOvVaw23Z7bVIzdNIUSCUh5CkFcG.